# A Communication Efficient Stochastic Multi-Block Alternating Direction Method of Multipliers

**Hao Yu**
Amazon
eeyuhao@gmail.com

## Abstract

The alternating direction method of multipliers (ADMM) has recently received tremendous interests for distributed large scale optimization in machine learning, statistics, multi-agent networks and related applications. In this paper, we propose a new parallel multi-block stochastic ADMM for distributed stochastic optimization, where each node is only required to perform simple stochastic gradient descent updates. The proposed ADMM is fully parallel, can solve problems with arbitrary block structures, and has a convergence rate comparable to or better than existing state-of-the-art ADMM methods for stochastic optimization. Existing stochastic (or deterministic) ADMMs require each node to exchange its updated primal variables across nodes at each iteration and hence cause significant amounts of communication overhead. Existing ADMMs require roughly the same number of inter-node communication rounds as the number of in-node computation rounds. In contrast, the number of communication rounds required by our new ADMM is only the square root of the number of computation rounds.

## 1  Introduction

Fix integer $N \geq 2$. Consider multi-block linearly constrained stochastic convex programs given by:

$$\min_{\mathbf{x}_i \in \mathcal{X}_i, \forall i} \quad f(\mathbf{x}) = \sum_{i=1}^{N} f_i(\mathbf{x}_i) \text{ s.t. } \sum_{i=1}^{N} \mathbf{A}_i \mathbf{x}_i = \mathbf{b}, \tag{1}$$

where $\mathbf{x}_i \in \mathbb{R}^{d_i}, \mathbf{A}_i \in \mathbb{R}^{m \times d_i}, \mathbf{b} \in \mathbb{R}^m, \mathcal{X}_i \subseteq \mathbb{R}^{d_i}$ are closed convex sets, and $f_i(\mathbf{x}_i) = \mathbb{E}_\xi[f_i(\mathbf{x}_i; \xi)]$ are convex functions. To have a compact representation of (1), we define $\mathbf{x} = [\mathbf{x}_1; \mathbf{x}_2; \ldots; \mathbf{x}_N] \in \mathbb{R}^{\sum_{i=1}^{N} d_i}$, $\mathcal{X} = \prod_{i=1}^{N} \mathcal{X}_i$, $f(\mathbf{x}) = \sum_{i=1}^{N} f_i(\mathbf{x}_i)$ and $\mathbf{A} = [\mathbf{A}_1, \mathbf{A}_2, \ldots, \mathbf{A}_N] \in \mathbb{R}^{m \times \sum_{i=1}^{N} d_i}$. Note that constraint $\sum_{i=1}^{N} \mathbf{A}_i \mathbf{x}_i = \mathbf{b}$ now can be written as $\mathbf{A}\mathbf{x} = \mathbf{b}$.

The problem (1) captures many important applications in machine learning, network scheduling, statistics and finance. For example, (stochastic) linear programs that are too huge to be solved over a single node can be written as (1). To solve such large scale linear programs in a distributed manner, we can save each $\mathbf{A}_i$ and $f_i(\cdot)$ at a separate node and let each node iteratively solves smaller sub-problems (with necessary inter-node communication). Another important application of formulation (1) is the distributed consensus training of a machine learning model over $N$ nodes [15, 17, 23] described as follows:

- In an online training setup, i.i.d. realizations of $f_i(\cdot; \xi)$ are sampled at each node. In an offline training setup, $f_i(\mathbf{x}_i) = \mathbb{E}_\xi[f_i(\mathbf{x}_i; \xi)]$ are approximated by $\frac{1}{N_i} \sum_{j=1}^{N_i} f_{ij}(\mathbf{x}_i)$ where $N_i$ is the number of training samples at node $i$ and each $f_{ij}(\cdot)$ represents one training sample.
- To enforce all $N$ nodes are training the same model, our constraint $\mathbf{A}\mathbf{x} = \mathbf{b}$ is given by $\mathbf{x}_i = \mathbf{x}_j$ for all $i \neq j \in \{1, 2, \ldots, N\}$. (In fact, we only need such constraints for pairs $(i, j)$ that construct a connected graph for all nodes.)

The Alternating Direction Method of Multipliers (ADMM) is an effective and popular method to solve linearly constrained convex programs, especially distributed consensus optimiation [28, 5],

since it often yields distributed implementations with low complexity [4]. Conventional ADMMs are developed for the special case of problem (1) with $N = 2$ and/or deterministic $f_i(\mathbf{x}_i)$. To solve a two-block problem (1) where $f_1$ is a stochastic function and $f_2$ is a deterministic function, previous works [21, 25, 31, 1] have developed stochastic (two-block) ADMMs to solve problem (1) with $N = 2$. It is unclear whether these methods can be extended to solve the case $N \geq 3$. In fact, even for problem (1) where all $f_i(\mathbf{x}_i)$ are deterministic, [6] proves that the classical (two-block) ADMM, on which the stochastic versions in [21, 25] are built, converges for $N = 2$ but diverges for $N \geq 3$. To solve stochastic convex program (1) with $N \geq 3$, randomized block coordinate updated ADMMs with $O(1/\epsilon^2)$ convergence are developed in [27, 11]. Due to the challenging stochastic objective functions, the convergence rate of stochastic ADMMs is fundamentally slower than deterministic ADMMs, i.e., $O(1/\epsilon^2)$ v.s. $O(1/\epsilon)$ [13, 7, 11]. The $O(1/\epsilon^2)$ convergence is optimal since it is optimal even for unconstrained stochastic convex optimization without strong convexity [20]. However, in distributive implementations of ADMMs, each node has to pass its most recent $\mathbf{x}_i$ value to its neighbors or a fusion center and then updates the dual variable $\boldsymbol{\lambda}$. Existing stochastic ADMM methods [21, 25, 11] require a communication step immediately after each $\mathbf{x}_i$ computation step. In practice, the inter-node communication over TCP/IP is much slower than in-node memory computations and often requires additional set-up time such that communication overhead is the performance bottleneck of most distributed optimization methods.

As a consequence, communication efficient optimization recently attracted a lot of research interests [29, 14, 24, 15, 17, 18, 23]. Work [17] proposes a primal-dual method that can solve problem (1) with stochastic objective functions using $O(1/\epsilon^2)$ computation iterations and $O(1/\epsilon)$ communication iterations. However, the method in [17] requires each objective function $f_i(\cdot)$ to satisfy the stringent condition that there exists $M$ such that $f_i(\mathbf{u}) \leq f_i(\mathbf{v}) + \langle \mathbf{d}, \mathbf{u} - \mathbf{v} \rangle + M\|\mathbf{u} - \mathbf{v}\|$ for any $\mathbf{u}, \mathbf{v}$ and $\mathbf{d} \in \partial f_i(\mathbf{v})$. Such a condition is more stringent than the smoothness when $\mathbf{u}$ and $\mathbf{v}$ are far apart from each other. For example, the simple scalar smooth function $f(x) = x^2$ does not satisfy this condition over $\mathcal{X} = \mathbb{R}$. Work [18] proposes a communication efficient method to solve **deterministic** convex programs based on the quadratic penalty method and can obtain an $\epsilon$-optimal solution with $O(1/\epsilon^{2+\delta})$ computation rounds ($\delta$ is a positive constant) and $O(1/\epsilon)$ communication rounds. For distributed consensus optimization over a network, which can be formulated as a special case of problem (1) where $\mathbf{A}_i$ and $\mathbf{b}$ are chosen to ensure all $\mathbf{x}_i$ are identical, mixing or local averaging based methods with fast convergence (and low communication overhead) are recently developed in [26, 22, 23, 19].

**Our Contributions:** This paper proposes a new communication efficient stochastic multi-block ADMM which has communication rounds less frequently than computation rounds. For stochastic convex programs with general convex objective functions, our algorithm can achieve an $\epsilon$-solution with $O(1/\epsilon^2)$ computation[1] rounds and $O(1/\epsilon)$ communication rounds. That is, our communication efficient ADMM has the same computation convergence rate as the ADMM in [11] but only requires the square root of communication rounds required by the method in [11]. For stochastic convex programs with strongly convex objective functions, our algorithm can achieve an $\epsilon$-accuracy solution with $\tilde{O}(1/\epsilon)$ computation rounds and $\tilde{O}(1/\sqrt{\epsilon})$ communication rounds[2]. The fast computation convergence (and even faster communication convergence) for strongly convex stochastic programs is not possessed by the ADMM in [11]. When applying our new multi-block ADMM to the special case of two-block problems, our algorithm has the same computation convergence as existing two-block stochastic ADMM methods in [21, 25, 31, 1]. However, the number of communication rounds used by our ADMM is only the squared root of these previous methods.

**Notations:** This paper uses $\|\mathbf{A}\|$ to denote the spectral norm of matrix $\mathbf{A}$; $\|\mathbf{z}\|$ to denote the Euclidean norm of vector $\mathbf{z}$; and $\langle \mathbf{y}, \mathbf{z} \rangle = \mathbf{y}^\mathsf{T}\mathbf{z}$ to denote the inner product of vectors $\mathbf{y}$ and $\mathbf{z}$. If symmetric matrix $\mathbf{Q} \succeq \mathbf{0}$ is positive semi-definite, then we define $\|\mathbf{z}\|_\mathbf{Q}^2 = \mathbf{z}^\mathsf{T}\mathbf{Q}\mathbf{z}$ for any vector $\mathbf{z}$.

## 2 Formulation and New Algorithm

Following the convention in [8], a function $h(\mathbf{x})$ is said to be *convex with modulus* $\mu$, or equivalently, $\mu$-convex, if $h(\mathbf{x}) - \frac{\mu}{2}\|\mathbf{x}\|^2$ is convex. The $\mu$-convex definition unifies the conventional definitions of convexity and strong convexity. That is, a general convex function, which is not necessarily strongly convex, is convex with modulus $\mu = 0$; and a strongly convex function is convex with modulus $\mu > 0$. Throughout this paper, convex program (1) is assumed to satisfy the following standard assumption:

**Assumption 1.** *Convex program* (1) *has a saddle point* $(\mathbf{x}^*, \boldsymbol{\lambda}^*)$. *That is,* $\mathbf{x}^*$ *is an optimal solution and* $\boldsymbol{\lambda}^* \in \mathbb{R}^m$ *is a Lagrange multiplier attaining strong duality* $q(\boldsymbol{\lambda}^*) = f(\mathbf{x}^*)$, *where* $q(\boldsymbol{\lambda}^*) \triangleq \inf_{\{\mathbf{x}_i \in \mathcal{X}_i, \forall i\}}\{f(\mathbf{x}) + \langle \boldsymbol{\lambda}^*, \mathbf{A}\mathbf{x} - \mathbf{b} \rangle\}$ *is the Lagrangian dual function.*

Note that strong duality in Assumption 1 is often stated as its equivalent "KKT conditions", e.g., in [7]. A mild sufficient condition for Assumption 1 to hold is (1) has at least one feasible point and the domain of each $f_i(\mathbf{x}_i)$ includes $\mathcal{X}_i$ as an interior [3].

Assume unbiased subgradients $G_i(\mathbf{x}_i; \xi)$ satisfying $\mathbb{E}_\xi[G_i(\mathbf{x}_i; \xi)] = \partial f_i(\mathbf{x}_i), \forall \mathbf{x}_i \in \mathcal{X}_i$ for each function $f_i(\mathbf{x}_i)$ can be sampled. Denote the stacked column vector $\mathbf{G}(\mathbf{x}; \xi) \triangleq [G_1(\mathbf{x}_1; \xi)^\mathsf{T}, \ldots, G_N(\mathbf{x}_N; \xi)^\mathsf{T}]^\mathsf{T} \in \mathbb{R}^{\sum_{i=1}^N d_i}$. We have $\mathbb{E}_\xi[\mathbf{G}(\mathbf{x}; \xi)] = \partial f(\mathbf{x})$.

Consider the communication efficient stochastic multi-block ADMM described in Algorithm 1. Since $f_i(\mathbf{x}_i)$ are stochastic, $\phi_i(\mathbf{x}_i)$ defined in (2) is fundamentally unknown. However, each $\phi_i(\mathbf{x}_i)$ is $\nu^{(t)}$-convex and its unbiased stochastic subgradient is available as long as we have unbiased stochastic subgradients of $f_i(\mathbf{x}_i)$. The sub-procedure STO-LOCAL involved in Algorithm 1 is a simple stochastic subgradient decent (SGD) procedure (with particular choices of parameters, starting points and averaging schemes) to minimize $\phi_i^{(t)}(\cdot)$ over set $\mathcal{X}_i$ and is described in Algorithm 2.

---

**Algorithm 1** Two-Layer Communication Efficient ADMM

1: **Input:** Algorithm parameters $T$, $\{\rho^{(t)}\}_{t \geq 1}$, $\{\nu^{(t)}\}_{t \geq 1}$ and $\{K^{(t)}\}_{t \geq 1}$.
2: Initialize arbitrary $\mathbf{y}_i^{(0)} \in \mathcal{X}_i, \forall i$, $\mathbf{r}^{(0)} = \sum_{i=1}^N \mathbf{A}_i \mathbf{y}_i^{(0)} - \mathbf{b}$, $\boldsymbol{\lambda}^{(0)} = \mathbf{0}$, and $t = 1$.
3: **while** $t \leq T$ **do**
4:     Each node $i$ defines

$$\phi_i^{(t)}(\mathbf{x}_i) \triangleq f_i(\mathbf{x}_i) + \rho^{(t)} \langle \mathbf{r}^{(t-1)} + \frac{1}{\rho^{(t)}} \boldsymbol{\lambda}^{(t-1)}, \mathbf{A}_i \mathbf{x}_i - \frac{\mathbf{b}}{N} \rangle + \frac{\nu^{(t)}}{2} \|\mathbf{x}_i - \mathbf{y}_i^{(t-1)}\|^2 \qquad (2)$$

and **in parallel** updates $\mathbf{x}_i^{(t)}, \mathbf{y}_i^{(t)}$ using local sub-procedure Algorithm 2 via

$$(\mathbf{x}_i^{(t)}, \mathbf{y}_i^{(t)}) = \text{STO-LOCAL}(\phi_i^{(t)}(\cdot), \mathcal{X}_i, \mathbf{y}_i^{(t-1)}, K^{(t)}) \qquad (3)$$

5:     Each node $i$ passes $\mathbf{x}_i^{(t)}$ and $\mathbf{y}_i^{(t)}$ between nodes or to a parameter server. Update $\boldsymbol{\lambda}^{(t)}$ and $\mathbf{r}^{(t)}$ via

$$\boldsymbol{\lambda}^{(t)} = \boldsymbol{\lambda}^{(t-1)} + \rho^{(t)} \Big( \sum_{i=1}^N \mathbf{A}_i \mathbf{x}_i^{(t)} - \mathbf{b} \Big) \qquad (4)$$

$$\mathbf{r}^{(t)} = \sum_{i=1}^N \mathbf{A}_i \mathbf{y}_i^{(t)} - \mathbf{b}. \qquad (5)$$

6:     Update $t \leftarrow t + 1$.
7: **end while**
8: **Output:** $\overline{\mathbf{x}}^{(T)} = \frac{1}{\sum_{t=1}^T \rho^{(t)}} \sum_{t=1}^T \rho^{(t)} \mathbf{x}^{(t)}$

---

**Algorithm 2** STO-LOCAL$(\phi(\mathbf{z}), \mathcal{Z}, \mathbf{z}^{\text{init}}, K)$

1: **Input:** $\mu$: strong convexity modulus of $\phi(\mathbf{z})$; Algorithm parameters: $k_0 > 0$; $\gamma^{(k)} = \frac{2}{\mu(k + k_0)}, \forall k \in \{1, 2, \ldots, K\}$.
2: Initialize $\mathbf{z}^{(0)} = \mathbf{z}^{\text{init}}$ and $k = 1$.
3: **while** $k \leq K$ **do**
4:     Observe an unbiased gradient $\boldsymbol{\zeta}^{(k)}$ such that $\mathbb{E}[\boldsymbol{\zeta}^{(k)}] = \partial \phi(\mathbf{z}^{(k-1)})$ and update $\mathbf{z}^{(k)}$ via

$$\mathbf{z}^{(k)} = \mathcal{P}_\mathcal{Z} \Big[ \mathbf{z}^{(k-1)} - \gamma^{(k)} \boldsymbol{\zeta}^{(k)} \Big] \qquad (6)$$

where $\mathcal{P}_\mathcal{Z}[\cdot]$ is the projection onto $\mathcal{Z}$.
5: **end while**
6: **Output:** $(\widehat{\mathbf{z}}, \mathbf{z}^{(K)})$ where $\widehat{\mathbf{z}}$ is the time average of $\{\mathbf{z}^{(0)}, \ldots, \mathbf{z}^{(K)}\}$ defined in Lemmas 1 or 2.

---

We now justify why Algorithm 1 is a two-layer ADMM method. (See Supplement 6.1 for a more detailed discussion.)

- The Lagrange multiplier update (4) is identical to that used in existing ADMM methods or other Lagrangian based methods. It is helpful to enforce the linear constraint.

- At the first sight, the primal update in Algorithm (4) is quite different from existing deterministic ADMMs in [10, 4, 7], which require to solve an "*argmin*" problem, or stochastic ADMMs in [21, 25, 11], which perform a single gradient descent step . However, with a simple manipulation, it is not difficult to show that that function $\phi_i^{(t)}(\mathbf{x}_i)$ in (2) is similar to the "*argmin*" target in the proximal Jacobi ADMM method [7] with the distinction that the proximal term $\|\mathbf{x}_i - \mathbf{y}_i^{(t-1)}\|^2$ is regarding a newly introduced variable $\mathbf{y}_i^{(t-1)}$ rather than $\mathbf{x}_i^{(t-1)}$.

Recall that the fastest stochastic ADMMs in [21, 25, 11] can solve general convex problem (1) (with $N = 2$) with $O(1/\sqrt{T})$ convergence. That is, to obtain a solution with $\epsilon$ errors for both the objective value and the constraint violation, the ADMMs in [21, 25, 11] require $O(1/\epsilon^2)$ computation steps, each of which uses a single gradient evaluation and variable update. The ADMMs in [21, 25, 11] has a single layer structure and hence are communication inefficient in the sense that each computation step involves a communication steps. Thus, the communication complexity of these stochastic ADMMs is also $O(1/\epsilon^2)$. Compared with existing ADMMs in [21, 25, 11], Algorithm 1 has a two layer structure where each outer layer step involves a single inter-node communication step given by (4)-(5) and calls the sub-procedure, i.e. Algorithm 2, STO-LOCAL($\phi_i^{(t)}(\cdot), \mathcal{X}_i, \mathbf{y}_i^{(t)}, K^{(t)}$), which is run by each node locally and in parallel and hence does not incur any inter-node communication overhead. Since each call of Algorithm 2 incurs $K^{(t)}$ SGD update, $T$ iterations of Algorithm 1 use $\sum_{t=1}^{T} K^{(t)}$ computation steps. We shall show that to achieve an $\epsilon$ solution for general convex problem (1), Algorithm 1 uses $T = O(1/\epsilon)$ communication rounds and $\sum_{t=1}^{T} K^{(t)} = O(1/\epsilon^2)$ computation steps. That is, Algorithm 1 is as fast as existing fastest stochastic ADMMs but uses only a square root of the number of communications rounds in [21, 25, 11].

Note that inter-node communication in Algoirthm 1 can be either centralized or decentralized. To use centralized communication, we can let all nodes pass their $\mathbf{x}_i^{(t)}$ to a parameter server, where (4)-(5) are executed, and then pull the updated $\boldsymbol{\lambda}^{(t)}$ and $\mathbf{r}^{(t)}$ from the server. It is possible to implement (4)-(5) using decentralized communication by exploring the structure of matrix $\mathbf{A} = [\mathbf{A}_1, \mathbf{A}_2, \ldots, \mathbf{A}_N]$. For example, consider distributed machine learning in a line network where $\mathbf{A}\mathbf{x} = \mathbf{b}$ is given by $N - 1$ equality constraints $\mathbf{x}_i - \mathbf{x}_{i+1} = 0, i \in \{1, 2, \ldots, N - 1\}$. In this case, $\boldsymbol{\lambda}_i^{(t)}$ and $\mathbf{r}_i^{(t)}$ only depend on $\mathbf{x}_i^{(t)}$ and $\mathbf{x}_{i+1}^{(t)}$ and are only used to updates $\mathbf{x}_i^{(t+1)}$ and $\mathbf{x}_{i+1}^{(t+1)}$. Thus, to implement Algorithm 1, each node only needs to send its local $\mathbf{x}_i^{(t)}$ to and pull $\boldsymbol{\lambda}_j^{(t)}$ and $\mathbf{r}_j^{(t)}$ from its neighbors in the line network.

## 2.1 Basic Facts of Algorithm 2

Since each iteration of Algorithm 1 calles Algorithm 2, which essentially applies SGD with carefully designed step size rules to newly introduced objective functions $\phi_i^{(t)}(\cdot)$. This subsection provides some useful insight of SGD for strongly convex stochastic minimization.

It is known that SGD can have $O(1/\epsilon)$ convergence for strongly convex minimization. The next two lemmas summarize the convergence of SGD Algorithm 2. When characterizing $O(1/\epsilon)$ rate, our lemmas also include a push-back term involving the last iteration solution. This term ensures when the SGD solution from Algorithm 2 is used in the outer-level ADMM dynamics, the accumulated error of our final solution does not explode. It also explains why we use $\mathbf{y}_i^{(t-1)}$, which is the last iteration solution from the SGD sub-procedure, rather than conventional $\mathbf{x}_i^{(t-1)}$ to define $\phi_i^{(t)}(\mathbf{x}_i)$.

**Lemma 1** ([16]). *Assume $\phi(\mathbf{z})$ is a $\mu$-convex function ($\mu > 0$) over set $\mathcal{Z}$ and there exists a constant $B$ such that the unbiased subgradient $\boldsymbol{\zeta}^{(k)}$ used in Algorithm 2 satisfies $\mathbb{E}[\|\boldsymbol{\zeta}^{(k)}\|^2] \leq B^2, \forall k \in \{1, 2, \ldots, K\}$. If we take $k_0 = 1$ in Algorithm 2, then for all $\mathbf{z} \in \mathcal{Z}$, we have*

$$\mathbb{E}[\phi(\widehat{\mathbf{z}})] \leq \phi(\mathbf{z}) - \underbrace{\frac{\mu}{2}\mathbb{E}[\|\mathbf{z}^{(K)} - \mathbf{z}\|^2]}_{(7)\text{-}term\ (I)} + \frac{2B^2}{\mu(K+1)}, \tag{7}$$

*where $\widehat{\mathbf{z}} = \frac{1}{\sum_{k=0}^{K-1}(k+k_0)} \sum_{k=0}^{K-1}(k + k_0)\mathbf{z}^{(k)}$.*

**Remark 1.** *It is firstly shown in [16] that Algorithm 2 with $k_0 = 1$ (vanilla SGD with a particular averaging scheme) has $O(1/\epsilon)$ convergence for non-smooth strongly convex problems. Note that (7) holds for all $\mathbf{z} \in \mathcal{Z}$ (not necessarily the minimizer of $\phi(\cdot)$). The push-back term (7)-term (I) is often ignored in convergence rate analysis for SGD but is important for our analysis of Algorithm 1.*

Recall that a function $h(\mathbf{x})$ is said to be $L$-smooth if its gradient $\nabla h(\mathbf{x})$ is Lipschitz with modulus $L$. The next lemma is new and extends Lemma 1 to smooth minimization such that the error term depends only on the variance of stochastic gradients (using a different averaging scheme).

**Lemma 2.** *Assume $\phi(\mathbf{z})$ is a $L$-smooth and $\mu$-convex function ($\mu > 0$) with conditional number $\kappa = \frac{L}{\mu}$ and there exists $\sigma > 0$ such that unbiased gradient $\boldsymbol{\zeta}^{(k)}$ (at point $\mathbf{z}^{(k-1)}$) in Algorithm 2 satisfies $\mathbb{E}[\|\boldsymbol{\zeta}^{(k)} - \nabla\phi(\mathbf{z}^{(k-1)})\|^2] \leq \sigma^2, \forall k \in \{1, 2, \ldots, K\}$. If we take integer $k_0 > 2\kappa$, then for any $\mathbf{z} \in \mathcal{Z}$, we have*

$$\mathbb{E}[\phi(\widehat{\mathbf{z}})] \leq \phi(\mathbf{z}) + \frac{\mu(k_0^2 - k_0)}{2K(K + 2k_0 - 1)}\left(\mathbb{E}[\|\mathbf{z} - \mathbf{z}^{(0)}\|^2] - \mathbb{E}[\|\mathbf{z} - \mathbf{z}^{(K)}\|^2]\right) - \frac{\mu}{2}\mathbb{E}[\|\mathbf{z} - \mathbf{z}^{(K)}\|^2] + \frac{2k_0\sigma^2}{(K + 2k_0 - 1)\mu} \quad (8)$$

*where $\widehat{\mathbf{z}} = \frac{1}{\sum_{k=1}^{K}(k+k_0-1)}\sum_{k=1}^{K}(k + k_0 - 1)\mathbf{z}^{(k)}$.*

*Proof.* See Supplement 6.6. $\qquad\qquad\square$

# 3 Performance Analysis of Algorithm 1

This section shows that Algorithm 1 can achieve an $\epsilon$-accuracy solution using $O(1/\epsilon^2)$ computation rounds and $O(1/\epsilon)$ communication rounds for general convex stochastic programs; or using $\tilde{O}(1/\epsilon)$ computation rounds and $\tilde{O}(1/\sqrt{\epsilon})$ communication rounds for strongly convex stochastic programs.

## 3.1 General objective functions (possibly non-smooth non-strongly convex)

**Theorem 1.** *Consider convex program (1) under Assumption 1. Let $(\mathbf{x}^*, \boldsymbol{\lambda}^*)$ be any saddle point defined in Assumption 1. Assume that*
- *The constraint set $\mathcal{X}$ is bounded, i.e., there exists constant $R > 0$ such that $\|\mathbf{x}\| \leq R, \forall \mathbf{x} \in \mathcal{X}$.*

- *The function $f(\mathbf{x})$ has unbiased stochastic subgradients with a bounded second order moment, i.e., there exists constant $D > 0$ such that $\mathbb{E}_\xi[\|\mathbf{G}(\mathbf{x}; \xi)\|^2] \leq D^2, \forall \mathbf{x} \in \mathcal{X}$.*

*For all $T \geq 1$, if we choose any fixed $\rho^{(t)} = \rho > 0$, $\nu^{(t)} = \nu \geq 8\rho\|\mathbf{A}\|^2$, $K^{(t)} = K \geq T$ in Algorithm 1 and the sub-procedure STO-LOCAL (Algorithm 2) uses $\widehat{\mathbf{z}}$ defined in Lemma 1 as the output, then*

$$\mathbb{E}[f(\overline{\mathbf{x}}^{(T)})] \leq f(\mathbf{x}^*) + \frac{\nu}{2T}\|\mathbf{x}^* - \mathbf{y}^{(0)}\|^2 + \frac{C}{2\nu T} \quad (9)$$

$$\mathbb{E}[\|\mathbf{A}\overline{\mathbf{x}}^{(T)} - \mathbf{b}\|] \leq \frac{1}{T}\frac{\sqrt{Q}}{\rho} \quad (10)$$

*where $\overline{\mathbf{x}}^{(T)} = \frac{1}{t}\sum_{t=1}^{T}\mathbf{x}^{(t)}$; $Q = (2\|\boldsymbol{\lambda}^*\| + \sqrt{\rho\nu\|\mathbf{x}^* - \mathbf{y}^{(0)}\|^2 + \frac{24\rho D^2}{\nu} + \frac{24(\rho)^3\|\mathbf{A}\|^2(\|\mathbf{A}\|R + \|\mathbf{b}\|)^2}{\nu} + 96\nu\rho R^2})/(1 - \sqrt{\frac{8\rho\|\mathbf{A}\|^2}{\nu}}))^2$ is an absolute constant (irrelevant to $T$); and $C \stackrel{\Delta}{=} 4\|\mathbf{A}\|^2 Q + 12D^2 + 12\rho^2\|\mathbf{A}\|^2(\|\mathbf{A}\|R + \|\mathbf{b}\|)^2 + 48\nu^2 R^2$ is also an absolute constant.*

*Proof.* See Supplement 6.7. $\qquad\qquad\square$

**Remark 2.** *After $T$ outer-level rounds, Algorithm 1 yields a solution with error $O(1/T)$. Note that the number of communication rounds is equal to the number of outer-level rounds and the number of computation rounds is $\sum_{t=1}^{T} K^{(t)} = O(T^2)$ when $K^{(t)} = T, \forall t$. Thus, to obtain an $\epsilon$-solution, Algorithm 1 uses $O(1/\epsilon)$ communication rounds and $O(1/\epsilon^2)$ computation rounds.*

**Remark 3.** *If we choose $\nu^{(t)} = \nu = 8\rho\|\mathbf{A}\|^2$ in Theorem 1 and further analyze the dependence on $\|\mathbf{A}\|$ in (9)-(10), we have $\mathbb{E}[f(\overline{\mathbf{x}}^{(T)})] \leq f(\mathbf{x}^*) + O(\frac{1}{T}\rho\|\mathbf{A}\|^2)$ and $\mathbb{E}[\|\mathbf{A}\overline{\mathbf{x}}^{(T)} - \mathbf{b}\|] \leq O(\frac{1}{T}(\frac{1}{\rho} + \|\mathbf{A}\|))$. If $\|\mathbf{A}\|$ is large, to balance the dependence on $\|\mathbf{A}\|$ in (9)-(10), we shall choose $\rho = \frac{1}{\|\mathbf{A}\|}$ such that the error terms in both (9) and (10) are order $O(\frac{1}{T}\|\mathbf{A}\|)$. In general, $\rho$ can be controlled to trade off between objective error and constraint error. For distributed consensus optimization considered in [26, 22, 23, 19] (assuming $d_i = 1$ without loss of generality), we can choose any $\mathbf{A}, \mathbf{b}$ that suffices to ensure the consistence of local solutions, e.g., Null$\{\mathbf{A}\}$ =Span$\{\mathbf{1}\}$ and $\mathbf{b} = \mathbf{0}$. Our method does not necessarily require $\mathbf{A} = \mathbf{I} - \mathbf{W}$ with a stochastic matrix $\mathbf{W}$ encoding the network topology as some methods in [26, 22, 23, 19]. Nevertheless, even when ung $\mathbf{A} = \mathbf{I} - \mathbf{W}$, our communication overhead can possibly have a better dependence on $\mathbf{W}$. Note that a stochastic matrix $\mathbf{W}$ ensures $\|\mathbf{A}\| \leq 2$. The convergence in [26, 22, 23, 19] (using a doubly stochastic or symmetric PSD $\mathbf{W}$ for mixing) further depends on $1/(1 - \max\{|\lambda_2(\mathbf{W})|, |\lambda_N(\mathbf{W})|\})$ or the eigen-gap $\lambda_1(\mathbf{W})/\lambda_{N-1}(\mathbf{W})$, which can be much larger than constant 2 when some eigenvalues are extreme.*

## 3.2 Smooth objective functions

For unconstrained stochastic smooth minimization, the constant factor in the SGD convergence rate is determined by the variance that can be significantly less than the second order moment for non-smooth stochastic minimization[20]. Such a property enable us to speed up SGD by averaging multiple i.i.d. stochastic gradients, e.g., mini-batch SGD. In this subsection, we show that Algorithm 1 has a similar property when $f(\cdot)$ in problem (1) is smooth.

**Theorem 2.** *Consider convex program (1) with $\mu$-convex (possibly $\mu = 0$) objective function under Assumption 1. Let $(\mathbf{x}^*, \boldsymbol{\lambda}^*)$ be any saddle point defined in Assumption 1. Assume that*
- *The function $f(\mathbf{x})$ is $L$-smooth.*
- *The function $f(\mathbf{x})$ has unbiased stochastic gradients with a bounded variance, i.e., there exists constant $\sigma > 0$ such that $\mathbb{E}_\xi[\|\mathbf{G}(\mathbf{x};\xi) - \nabla f(\mathbf{x})\|^2] \leq \sigma^2, \forall \mathbf{x} \in \mathcal{X}$.*

*If the sub-procedure STO-LOCAL (Algorithm 2) uses $\hat{\mathbf{z}}$ defined in Lemma 2 as the output, then Algorithm 1 ensures:*

- **General Convex** ($\mu = 0$)*: For all $T \geq 1$, if we choose any fixed $\rho^{(t)} = \rho > 0$, $\nu^{(t)} = \nu \geq \rho\|\mathbf{A}\|^2$, $K^{(t)} = K = T$ and positive integer $k_0 \geq 2\frac{L+\nu}{\nu}$, then we have*

$$\mathbb{E}[f(\bar{\mathbf{x}}^{(T)})] \leq f(\mathbf{x}^*) + \frac{1}{T}\frac{\nu(k_0+1)}{4}\|\mathbf{x}^* - \mathbf{y}^{(0)}\|^2 + \frac{1}{T}\frac{2k_0\sigma^2}{\nu} \tag{11}$$

$$\mathbb{E}[\|\mathbf{A}\bar{\mathbf{x}}^{(T)} - \mathbf{b}\|] \leq \frac{1}{T}\Big(\frac{2}{\rho}\|\boldsymbol{\lambda}^*\| + \sqrt{\frac{\nu(k_0+1)}{2\rho}}\|\mathbf{x}^* - \mathbf{y}^{(0)}\| + 2\sqrt{\frac{k_0\sigma^2}{\rho\nu}}\Big) \tag{12}$$

*where $\bar{\mathbf{x}}^{(T)} = \frac{1}{t}\sum_{t=1}^T \mathbf{x}^{(t)}$.*

- **Strongly Convex** ($\mu > 0$)*: For all $T \geq 1$, if we choose $\rho \leq \frac{\mu}{3\|\mathbf{A}\|^2}, \rho^{(t)} = t\rho$, $\nu^{(t)} = t\rho\|\mathbf{A}\|^2$, positive integer $k_0 \geq 2(1 + \frac{L}{\mu})$ and $K^{(t)} = (2k_0 - 1)t$, then we have*

$$\mathbb{E}[f(\bar{\mathbf{x}}^{(T)})] \leq f(\mathbf{x}^*) + \frac{1}{T(T+1)}\Big(c_1\|\mathbf{x}^* - \mathbf{y}^{(0)}\|^2 + \frac{c_2}{\rho}\log(T+1)\Big) \tag{13}$$

$$\mathbb{E}[\|\mathbf{A}\bar{\mathbf{x}}^{(T)} - \mathbf{b}\|] \leq \frac{2}{T(T+1)}\Big(\frac{4\|\boldsymbol{\lambda}^*\|}{\rho} + \frac{\sqrt{c_1}}{\sqrt{\rho}}\|\mathbf{x}^* - \mathbf{y}^{(0)}\| + \frac{\sqrt{c_2\log(T+1)}}{\rho}\Big) \tag{14}$$

*where $\bar{\mathbf{x}}^{(T)} = \frac{1}{\sum_{t=1}^T \rho^{(t)}}\sum_{t=1}^T \rho^{(t)}\mathbf{x}^{(t)}$; and $c_1 \triangleq \rho\|\mathbf{A}\|^2 + \frac{(\rho\|\mathbf{A}\|^2+\mu)(k_0^2-k_0)}{2(2k_0-1)^2}$ and $c_2 \triangleq \frac{4k_0\sigma^2}{(2k_0-1)\|\mathbf{A}\|^2}$ are two constants.*

*Proof.* See Supplement 6.8. □

**Remark 4.** *If $f(\mathbf{x})$ in convex program (1) is strongly convex, Algorithm 1 can obtain a solution with error $O(\frac{\log(T)}{T^2})$ after $T$ outer-level rounds. Recall the number of communication rounds is equal to the number of outer-level rounds and the number of computation rounds is equal to $\sum_{t=1}^T K^{(t)} = \frac{2k_0-1}{2}T(T+1) = O(T^2)$, Algorithm 1 requires $\tilde{O}(\frac{1}{\epsilon})$ communication rounds and $\tilde{O}(\frac{1}{\epsilon^2})$ computation rounds to obtain an $\epsilon$-solution.*

## 3.3 Non-smooth strongly convex objective functions

There is a fourth case, where the stochastic objective function $f(\mathbf{x})$ is strongly convex but possibly non-smooth, uncovered in the previous subsections. In this case, we assume the following condition (originally introduced in [17]): There exists constant $M > 0$ such that

$$f(\mathbf{x}) \leq f(\mathbf{y}) + \langle \mathbf{d}, \mathbf{x} - \mathbf{y}\rangle + M\|\mathbf{x} - \mathbf{y}\|, \tag{15}$$

for all $\mathbf{x}, \mathbf{y} \in \mathcal{X}$ and $\mathbf{d} \in \partial f(\mathbf{y})$. This condition is assumed throughout [17] to develop a different communication efficient primal-dual method. Supplement 6.9 shows this condition is almost as useful as smoothness and under this condition, our communication efficient ADMM can achieve an $\epsilon$-accuracy solution with $\tilde{O}(1/\epsilon)$ computation rounds and $\tilde{O}(1/\sqrt{\epsilon})$ communication rounds for non-smooth strongly convex stochastic optimization.

# 4 Experiments

## 4.1 Distributed Stochastic Optimization with Noisy Stochastic Gradient Information

Consider simple stochastic optimization given by

$$\min \quad \sum_{i=1}^{3} \mathbb{E}_{\mathbf{c}_i}[\|\mathbf{x}_i - \mathbf{c}_i\|_2^2] \tag{16}$$

$$\text{s.t.} \quad \mathbf{x}_1 = \mathbf{x}_2, \mathbf{x}_2 = \mathbf{x}_3 \tag{17}$$

$$\mathbf{x}_i \in [-1,1]^3, \forall i \in \{1,2,\ldots,3\} \tag{18}$$

where $\mathbf{c}_i \sim \mathcal{N}(\bar{\mathbf{c}}_i, \sigma_i^2 \mathbf{I})$ satisfy normal distributions with $\bar{\mathbf{c}}_1 = [-2.0871, -0.3702, 0.2302]^\mathsf{T}$, $\sigma_1 = 0.1$, $\bar{\mathbf{c}}_2 = [-0.5556, -0.4413, 0.2869]^\mathsf{T}$, $\sigma_2 = 0.2$, $\bar{\mathbf{c}}_3 = [-1.4991, -1.8286, -2.0477]^\mathsf{T}$ and $\sigma_3 = 0.1$. Solving this problem with Algorithm 1 only requires each node to access samples of local $\mathbf{c}_i$ and does not use the true value $\bar{\mathbf{c}}_i$ and $\sigma_i$, which are fundamentally unavailable. However, by assuming the knowledge of $\bar{\mathbf{c}}_i$ and $\sigma_i$, we can convert this stochastic optimization to a deterministic problem and use CVXPY [9] to obtain the unique solution $\mathbf{x}_1^* = \mathbf{x}_2^* = \mathbf{x}_3^* = [-1, -0.88003599, -0.51020207]^\mathsf{T}$ such that we can evaluate the performance of Algorithm 1. Since the objective function is smooth and strongly convex, by Theorem 2, using time-varying parameters in Algorithm 1 has faster convergence. We run Algorithm 1 with constant $\rho, \nu$ according to[3] Theorem 1 and with time-varying $\rho^{(t)}, \nu^{(t)}$ according to Theorem 2, respectively. Note that if an algorithm has $O(1/\epsilon^\beta)$ convergence, then its error should decay like $O(1/t^{1/\beta})$ where $t$ is the iteration index.

Figures 1 plots the distance to $\mathbf{x}^*$ versus the computation round index or the communication round index in a log-log scale. It also plots baseline curves $1/t^{\frac{1}{\beta}}$ corresponding to $O(1/\epsilon^\beta)$ convergence proven in the theorems. Note that in a log-log scale, curves $1/t^{\frac{1}{\beta}}$ become straight lines with slopes $-\frac{1}{\beta}$. That is, if our algorithm has the proven convergence rate, the error curves should be eventually parallel to corresponding baseline for large $t$. In Figures 1, we observe the numerical result is consistent with our theoretical rate proven in our theorems. This simple experiment verifies the correctness of our theorems. Our multi-core implementation of Algorithm 1 uses Python 3.7 and MPI4PY. In an experiment over a machine with a multi-core Intel Xeon Processor E5-2682 2.5GHz. Each computation round takes $0.3$ms and each communication round takes $43.7$ms. Note communication becomes more relatively expensive as more parallel nodes/cores are involved.

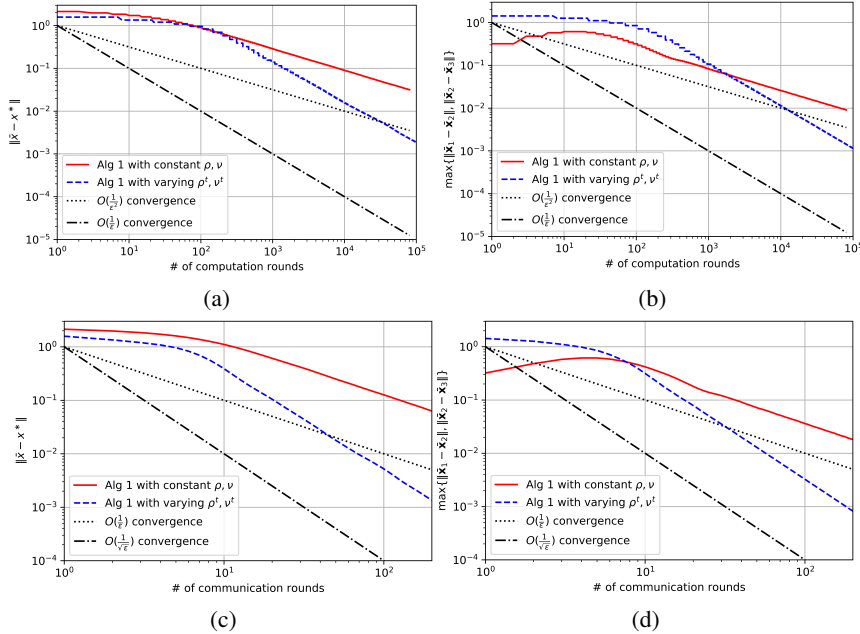

Figure 1: Performance of Algorithm 1 to solve stochastic optimization (16)-(18): (a)& (b) convergence w.r.t. # of computation rounds; (c)&(d) convergence w.r.t. # of communication rounds.

## 4.2 Distributed $l_1$ Regularized Logistic Regression

Consider a distributed $l_1$ regularized logistic regression problem (over 10 nodes) given by:

$$\min \quad \frac{1}{10}\sum_{i=1}^{10}\frac{1}{N_i}\sum_{j=1}^{N_i}\log(1+\exp(b_{ij}(\mathbf{a}_{ij}^\mathsf{T}\mathbf{x}_i)))+\mu\|\mathbf{x}_i\|_1 \tag{19}$$

with each optimization variable $\mathbf{x}_i \in \mathbb{R}^d$. Each node contains $N_i$ training pairs $(\mathbf{a}_{ij}, b_{ij})$, where $\mathbf{a}_{ij} \in \mathbb{R}^d$ is a feature vector and $b_{ij} \in \{-1, 1\}$ is the corresponding label. To ensure all nodes yield a consistent model, consensus constraints are needed to enforce all $\mathbf{x}_i$ are equal. Note that conventional two-block ADMMs must introduce a dummy block (server node) $\mathbf{z}$ and add constraints $\mathbf{x}_i = \mathbf{z}$. (See e.g., [4, 21, 25].) However, such an ADMM method requires all nodes to pass the updated $\mathbf{x}_i$ value to the (server) node corresponding to the $\mathbf{z}$ block and hence can turn $\mathbf{z}$ node into a communication bottleneck in large networks. In contrast, using a multi-block ADMM method allows arbitrary linear constraints, e.g., constraints $\mathbf{x}_i = \mathbf{x}_{i+1}, \forall i$ that ensure all $\mathbf{x}_i$ are equal, and the corresponding multi-block ADMM only uses communication between adjacent blocks. Alternatively, consider a line network where only one-hop transmission is allowed, then our ADMM naturally yields a protocol that is faithful to the network communication restriction. In general, given an arbitrary network communication topology, our multi-block ADMM can always yield an implementable distributed protocol by adding constraints $\mathbf{x}_i = \mathbf{x}_j$ for links $(i, j)$ existing in the network.

We generate a problem instance in a way similarly to [4]. Our problem instance uses $d = 100$, $N_i = 10^5$ for all $i$ and $\mu = 0.002$. Each feature vector $\mathbf{a}_{ij}$ is generated from a standard normal distribution. We choose a true weight vector $\mathbf{x}^{\text{true}} \in \mathbb{R}^d$ with 10 non-zero entries from a standard normal distribution and then generate the label $b_{ij} = \text{sign}(\mathbf{a}_{ij}^\mathsf{T}\mathbf{x}^{\text{true}} + n_i)$ where noise $n_i \sim \mathcal{N}(0, \sigma_i^2)$ with fixed constants $\sigma_i$ randomly generated from a uniform distribution $\text{Unif}[0, 1]$. Figures 2 compares Algorithm 1 with RPDBUS ADMM proposed in [11], where the number of communication rounds is the same that of computation rounds, and DCS in [17], where the number of communication rounds is the square root of that of computation rounds. We observe that Algorithm 1 has fastest convergence with respect to both computation and communication.

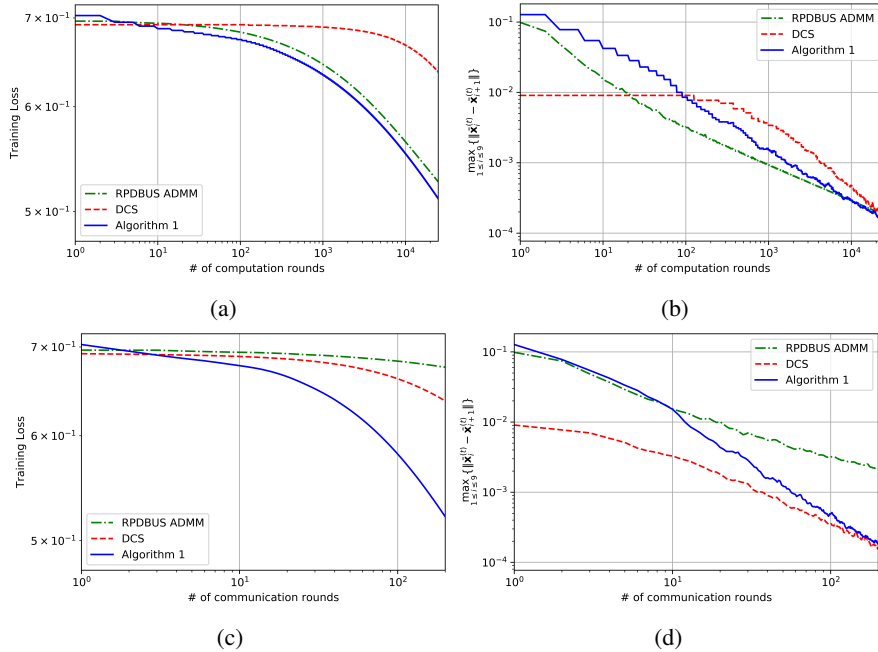

Figure 2: Distributed $l_1$ regularized logistic regression: (a)& (b) performance w.r.t. # of computation rounds; (c)&(d) performance w.r.t. # of communication rounds

## 5 Conclusions

This paper proposes a new communication efficient multi-block ADMM for linearly constrained stochastic optimization. This method is as fast as (or faster than) existing stochastic ADMMs but the associated communication overhead is only the square root of that required by existing ADMMs.

## Footnotes

[1] A computation round of our algorithm is a just a single iteration of the SGD update.

[2] A logarithm factor $\log(\frac{1}{\epsilon})$ is hidden in the notation $\tilde{O}(\cdot)$.

[3] Since $f(\mathbf{x})$ is also smooth, using constant $\rho, \nu$ according to Theorem 2 can give a similar (slightly better) performance. Theoretically, by using $K^{(t)} = t$ rather than $K^{(t)} = T$ for a fixed $T$, the rate is slightly worse, i.e. $O(\log(T)/T)$ v.s. $O(1/T)$. However, we find the performance degradation for large $T$ regions is negligible when using $K^{(t)} = t$. In contrast, using $K^{(t)} = t$ enable the algorithm converge faster for small $t$. We use $K^{(t)} = t$ when performing the numerical experiments in this paper.

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
