[Supplementary Material]

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

[4]In fact, the $\|\mathbf{x}^{(t+1)} - \mathbf{x}^{(t)}\|^2 \leq o(1/t)$ convergence is so weak that it does not even imply $\mathbf{x}^t$ converges to a fixed $\mathbf{x}^*$. For example, the scalar sequence $x^{(t)} = t^{1/4}$ satisfies $\|x^{(t+1)} - x^{(t)}\|^2 \leq o(1/t)$ but diverges to $\infty$.

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

# 6 Supplement

## 6.1 Connection between Algorithm 1 and Existing ADMMs

Note that Algorithm 1 uses the same Lagrange multiplier update as most existing ADMM methods. The Lagrange multiplier update (4) is helpful to enforce the linear constraint. (See Lemma 3 in Supplement 6.3 for a more technical justification.) However, the per-iteration $\mathbf{x}_i^{(t)}$ updates in Algorithm 1 introduce new stochastic functions $\phi_i^{(t)}(\mathbf{x}_i)$ and let each node call a SGD sub-procedure locally to minimize $\phi_i^{(t)}(\mathbf{x}_i)$. This is quite different from existing deterministic ADMMs [4, 7], which require to solve an "*argmin*" problem exactly, or existing stochastic ADMMs [21, 25, 11], which perform a single gradient descent step. The "*argmin*" update is fundamentally impossible for stochastic minimization since the stochastic objective function is fundamentally unknown and can only be sampled. Our intuition is existing stochastic ADMMs are too conservative in updating $\mathbf{x}_i$ by restricting themselves to a **single** gradient descent update and then communicate immediately for the Lagrange multiplier update. In contrast, our Algorithm 1 introduces the SGD sub-procedure (Algorithm 2) for each node to update $\mathbf{x}_i$ using $K^{(t)}$ gradient descent steps. Such SGD sub-procedures only involve local computations and do not incur any inter-node communication. This is the key reason why our Algorithm 1 requires fewer communication rounds than computation rounds. It is tempting to interpret Algorithm 1 as an ADMM variant where the "*argmin*" primal update is only approximately solved using local SGD sub-procedures. Previous work [10] considers ADMM variants with inexact "*argmin*" primal updates for **deterministic** optimization without analyzing the convergence rate. However, our Algorithm 1 is different from the method in [10] and can solve more challenging stochastic optimization with convergence rate guarantees.

It remains to see how we come up with $\phi_i^{(t)}(\mathbf{x}_i)$ in (2). To see so, we introduce Algorithm 3 that generalizes the deterministic multi-block ADMM in [7] and provide new insight and analysis.

---

**Algorithm 3** Deterministic Multi-Block Proximal Jacobi ADMM (generalized from [7])

---

1: **Input:** Algorithm parameters: $\{\mathbf{P}_i^{(t)}\}_{t\geq 1, i\in\{1,2,\ldots,N\}}$ with $\mathbf{P}_i^{(t)} \succeq 0, \forall i, \forall t$; $\{\rho^{(t)}\}_{t\geq 1}$.
2: Initialize arbitrary $\mathbf{x}_i^{(0)} \in \mathcal{X}_i, \forall i, \boldsymbol{\lambda}^{(0)} = \mathbf{0}$ and $t = 1$.
3: **while** $t \leq T$ **do**
4:    Update each $\mathbf{x}_i^{(t)}$ **in parallel** equal to

$$\operatorname*{argmin}_{\mathbf{x}_i \in \mathcal{X}_i} \left\{ f_i(\mathbf{x}_i) + \frac{\rho^{(t)}}{2}\|\mathbf{A}_i\mathbf{x}_i + \sum_{j\neq i}\mathbf{A}_j\mathbf{x}_j^{(t-1)} - \mathbf{b} + \frac{1}{\rho^{(t)}}\boldsymbol{\lambda}^{(t-1)}\|^2 + \frac{1}{2}\|\mathbf{x}_i - \mathbf{x}_i^{(t-1)}\|_{\mathbf{P}_i^{(t)}}^2 \right\}. \quad (20)$$

5:    Update $\boldsymbol{\lambda}^{(t)}$ according to (4).
6:    Update $t \leftarrow t + 1$.
7: **end while**
8: **Output:** $\overline{\mathbf{x}}^{(T)} = \frac{1}{\sum_{t=1}^{T}\rho^{(t)}}\sum_{t=1}^{T}\rho^{(t)}\mathbf{x}^{(t)}$

---

Algorithm 3 is almost identical to the original parallel multi-block ADMM proposed in [12, 7] except that it allows $\mathbf{P}_i^{(t)}$ and $\rho^{(t)}$ to be time-varying. We will show later that time-varying $\mathbf{P}_i^{(t)}$ and $\rho^{(t)}$ are useful for Algorithm 3 to achieve faster $O(1/\sqrt{\epsilon})$ convergence for problems with strongly convex objective functions. Note that if we take $\mathbf{P}_i^{(t)} = \nu^{(t)}\mathbf{I} - \rho^{(t)}\mathbf{A}_i^\mathsf{T}\mathbf{A}_i$ with scalar $\nu^{(t)} > 0$, then (20) in Algorithm 3 is equivalent to

$$\mathbf{x}_i^{(t)} = \operatorname*{argmin}_{\mathbf{x}_i \in \mathcal{X}_i} \left\{ f_i(\mathbf{x}_i) + \rho^{(t)}\big\langle \mathbf{A}_i^\mathsf{T}\Big(\sum_{i=1}^{N}\mathbf{A}_i\mathbf{x}_i^{(t-1)} - \mathbf{b} + \frac{1}{\rho^{(t)}}\boldsymbol{\lambda}^{(t-1)}\Big), \mathbf{x}_i\big\rangle + \frac{\nu^{(t)}}{2}\|\mathbf{x}_i - \mathbf{x}_i^{(t-1)}\|^2 \right\} \quad (21)$$

$$= \operatorname*{argmin}_{\mathbf{x}_i \in \mathcal{X}_i} \left\{ f_i(\mathbf{x}_i) + \rho^{(t)}\big\langle \sum_{i=1}^{N}\mathbf{A}_i\mathbf{x}_i^{(t-1)} - \mathbf{b} + \frac{1}{\rho^{(t)}}\boldsymbol{\lambda}^{(t-1)}, \mathbf{A}_i\mathbf{x}_i - \frac{\mathbf{b}}{N}\big\rangle + \frac{\nu^{(t)}}{2}\|\mathbf{x}_i - \mathbf{x}_i^{(t-1)}\|^2 \right\}$$
$$(22)$$

Since both $\langle \mathbf{c}, \mathbf{x}_i\rangle$ and $\|\mathbf{x}_i - \mathbf{x}_i^{(t-1)}\|^2$ are separable (with respect to each component of vector $\mathbf{x}_i$), the equivalent minimization step (21) or (22) can be further decomposed into $d_i$ simple scalar

minimization subproblems if $f_i(\mathbf{x}_i)$ is also separable, e.g. linear $f_i(\mathbf{x}_i)$ or $f_i(\mathbf{x}_i) = \|\mathbf{x}_i\|_1$. Thus, suitable choices of $\mathbf{P}_i^{(t)}$ can remarkably reduce the implementation complexity of Algorithm 3 and enable the parallelism of $\mathbf{x}_i^{(t)}$ updates for different $i$. See [8, 7] for more discussions on the benefit of introducing $\mathbf{P}_i^{(t)}$.

Note $\mathbf{r}^{(t-1)} = \sum_{i=1}^N \mathbf{A}_i \mathbf{y}_i^{(t-1)} - \mathbf{b}$ in Algorithm 1, it now becomes transparent how $\phi_i^{(t)}(\mathbf{x}_i)$ in (2) is developed in Algorithm 1. Each $\phi_i^{(t)}(\mathbf{x}_i)$ is obtained by replacing each $\mathbf{x}_i^{(t-1)}$ in expression (22) with a newly introduced variable $\mathbf{y}_i^{(t-1)}$. Algorithm 1 then further call a SGD sub-procedure (Algorithm 2) to minimize $\phi_i^{(t)}(\mathbf{x}_i)$. The introduction of $\mathbf{y}_i^{(t-1)}$ is to compensate the error accumulated in the SGD sub-procedures and is further justified in Section 2.1.

To further motivate the development of Algorithm 1 from Algorithm 3, the next theorem summarizes the convergence of Algorithm 3 for **deterministic** convex programs:

**Theorem 3.** *Consider convex programs in the form of* (1) *with $\mu$-convex (possibly non-smooth) deterministic $f(\mathbf{x})$. Let $(\mathbf{x}^*, \boldsymbol{\lambda}^*)$ be a saddle point in Assumption 1.*

1. **General Convex** *($\mu = 0$): If we choose any fixed $\rho^{(t)} = \rho > 0$ and $\mathbf{P}_i^{(t)} = \mathbf{P}_i = \nu \mathbf{I} - \rho \mathbf{A}_i^\mathsf{T} \mathbf{A}_i$ with $\nu \geq \rho \|\mathbf{A}\|^2$ in Algorithm 3, then we have*

$$f(\overline{\mathbf{x}}^{(T)}) \leq f(\mathbf{x}^*) + \frac{1}{2T} \|\mathbf{x}^* - \mathbf{x}^{(0)}\|_{\mathbf{Q}}^2 \tag{23}$$

$$\|\mathbf{A}\overline{\mathbf{x}}^{(T)} - \mathbf{b}\| \leq \frac{1}{T} \frac{2\|\boldsymbol{\lambda}^*\|}{\rho} + \frac{1}{T} \frac{\|\mathbf{x}^* - \mathbf{x}^{(0)}\|_{\mathbf{Q}}}{\sqrt{\rho}} \tag{24}$$

   *where $\overline{\mathbf{x}}^{(T)} = \frac{1}{\sum_{t=1}^T \rho} \sum_{t=1}^T \rho \mathbf{x}^{(t)} = \frac{1}{T} \sum_{t=1}^T \mathbf{x}^{(t)}$; $\mathbf{Q} = Diag(\mathbf{Q}_1, \ldots, \mathbf{Q}_N) = Diag(\mathbf{P}_1 + \rho \mathbf{A}_1^\mathsf{T} \mathbf{A}_1, \ldots, \mathbf{P}_N + \rho \mathbf{A}_N^\mathsf{T} \mathbf{A}_N)$.*

2. **Strongly Convex** *($\mu > 0$): If we choose $\rho \leq \frac{\mu}{3\|\mathbf{A}\|^2}$, $\rho^{(t)} = t\rho$ and $\mathbf{P}_i^{(t)} = t\rho\|\mathbf{A}\|^2 \mathbf{I} - t\rho \mathbf{A}_i^\mathsf{T} \mathbf{A}_i$ in Algorithm 3, then we have*

$$f(\overline{\mathbf{x}}^{(T)}) \leq f(\mathbf{x}^*) + \frac{\rho}{T(T+1)} \|\mathbf{A}\|^2 \|\mathbf{x}^* - \mathbf{x}^{(0)}\|^2 \tag{25}$$

$$\|\mathbf{A}\overline{\mathbf{x}}^{(T)} - \mathbf{b}\| \leq \frac{4\|\boldsymbol{\lambda}^*\|}{\rho T(T+1)} + \frac{2\|\mathbf{A}\|\|\mathbf{x}^* - \mathbf{x}^{(0)}\|}{T(T+1)} \tag{26}$$

*Proof.* See Supplement 6.5. □

**Remark 5.** *It is sufficient to use any constant $\rho$ to ensure $O(1/T)$ convergence for the $\mu = 0$ case. However, a larger $\rho$ yields larger objective error (note that $\|\cdot\|_{\mathbf{Q}}^2 = O(\rho)$) and smaller constraint error. Thus, $\rho$ can be controlled to trade off between objective error and constraint error. Similar tradeoffs also hold for the $\mu > 0$ case (as long as $\rho$ satisfies the condition ensuring the algorithm convergence) and other algorithms in this paper.*

**Remark 6.** *For the $\mu = 0$ case, Algorithm 3 with fixed algorithm parameters degrades to the proximal Jacobi ADMM considered in [7]. However, the convergence rate shown in [7] is in the weak form of $\|\mathbf{x}^{(t+1)} - \mathbf{x}^{(t)}\|^2 \leq o(1/t)$ and does not necessarily mean[4] the convergence for the objective value or feasibility shown in Theorem 3. In contrast, our Theorem 3 proves the $O(1/T)$ convergence rate of Algorithm 3 regarding the objective value and feasibility, which is the concern for math optimization.*

A similar $O(1/T)$ convergence rate, or equivalently, $O(1/\epsilon)$ convergence time, for $\mu = 0$ case is independently shown in [11] for an ADMM variant different from Algorithm 3. In Supplement 6.5, we provide a different analysis that unifies both $\mu = 0$ and $\mu > 0$ cases. To our knowledge, the $O(1/T^2)$ convergence rate of Algorithm 3 with time-varying parameters for $\mu > 0$ case (with possibly non-smooth $f(\mathbf{x})$ and arbitrary matrix $\mathbf{A}$) is new. Existing faster convergence of ADMM for strongly convex programs requires additional conditions of $f(\mathbf{x})$ and/or $\mathbf{A}$.

## 6.2 Analysis Technique in This Paper

Note that our analysis technique for the proximal Jacobi ADMM described in Algorithm 3 and the communication efficient stochastic ADMM in Algorithm 1 is different from the analysis for conventional Jacobi type ADMM as in [7]. The analysis in this paper is extended from [30] where a proximal Lagrangian based method is developed for convex programs with possibly non-linear constraints. By utilizing the simpler linear constraint structure, we obtain finer convergence rate results for Algorithm 3 and further establish the computation and communication complexity for Algorithm 1.

## 6.3 Basic Facts from Lagrange Multiplier Updates

In this section, we present two lemmas that hold for any algorithm using (4) to update $\boldsymbol{\lambda}$. These two lemmas are frequently used to analyze the feasibility violations in this paper.

**Lemma 3.** *Let* $\boldsymbol{\lambda}^{(0)} = \mathbf{0}$ *and* $\boldsymbol{\lambda}^{(t)}, t \geq 1$ *be updated according to* (4).

1. *For any* $T \geq 1$, *we have* $\sum_{t=1}^{T} \rho^{(t)} \left( \mathbf{A}\mathbf{x}^{(t)} - \mathbf{b} \right) = \boldsymbol{\lambda}^{(T)}$

2. *For all* $t \geq 1$, *we have* $\left\langle \boldsymbol{\lambda}^{(t-1)}, \mathbf{A}\mathbf{x}^{(t)} - \mathbf{b} \right\rangle = \frac{1}{2\rho^{(t)}} \left( \|\boldsymbol{\lambda}^{(t)}\|^2 - \|\boldsymbol{\lambda}^{(t-1)}\|^2 \right) - \frac{\rho^{(t)}}{2} \|\mathbf{A}\mathbf{x}^{(t)} - \mathbf{b}\|^2$.

*Proof.*

1. This follows directly from the update equation (4).

2. Fix $t \geq 1$. Taking the squared vector $l_2$ norm on both sides of (4) yields

$$\|\boldsymbol{\lambda}^{(t)}\|^2 = \|\boldsymbol{\lambda}^{(t-1)}\|^2 + (\rho^{(t)})^2 \| \sum_{i=1}^{N} \mathbf{A}_i \mathbf{x}_i^{(t)} - \mathbf{b}\|^2 + 2\rho^{(t)} \left\langle \boldsymbol{\lambda}^{(t-1)}, \mathbf{A}\mathbf{x}^{(t)} - \mathbf{b} \right\rangle.$$

This part follows by dividing by $2\rho^{(t)}$ on both sides and rearranging terms.

$\square$

Note that part (1) of lemma implies that to analyze the accumulated feasibility violations over $T$ iterations, it is sufficient to analyze the boundedness of $\boldsymbol{\lambda}^{(T)}$. The next lemma follows directly from the saddle point assumption (Assumption 1) and relates $\boldsymbol{\lambda}^T$ with the accumulated objective performance.

**Lemma 4.** *Consider convex program* (1) *under Assumption 1 such that* $(\mathbf{x}^*, \boldsymbol{\lambda}^*)$ *is any saddle point defined in Assumption 1. For any* $T \geq 1$, *if an algorithm generates* $\mathbf{x}^{(t)} \in \mathcal{X}$ *and updates* $\boldsymbol{\lambda}^{(t)}$ *according to* (4)( *with* $\boldsymbol{\lambda}^{(0)} = \mathbf{0}$) *at each iteration* $t \in \{1, 2, \ldots, T\}$, *then we have*

$$\sum_{t=1}^{T} \rho^{(t)} f(\mathbf{x}^{(t)}) \geq \sum_{t=1}^{T} \rho^{(t)} f(\mathbf{x}^*) - \|\boldsymbol{\lambda}^*\| \|\boldsymbol{\lambda}^{(T)}\|$$

*Proof.* Fix $T > 0$. For any $t \in \{1, \ldots, T\}$, by Assumption 1, we have

$$f(\mathbf{x}^*) = q(\boldsymbol{\lambda}^*) \triangleq \inf_{\mathbf{x} \in \mathcal{X}} \{f(\mathbf{x}) + \langle \boldsymbol{\lambda}^*, \mathbf{A}\mathbf{x} - \mathbf{b} \rangle\} \overset{(a)}{\leq} f(\mathbf{x}^{(t)}) + \left\langle \boldsymbol{\lambda}^*, \mathbf{A}\mathbf{x}^{(t)} - \mathbf{b} \right\rangle$$

where (a) trivially follows because $\mathbf{x}^{(t)} \in \mathcal{X}$. Multiplying $\rho^{(t)}$ on both sides and summing over $t \in \{1, 2, \ldots, T\}$ yields

$$\sum_{t=1}^{T} \rho^{(t)} f(\mathbf{x}^*) \leq \sum_{t=1}^{T} \rho^{(t)} f(\mathbf{x}^{(t)}) + \left\langle \boldsymbol{\lambda}^*, \sum_{t=1}^{T} \rho^{(t)} (\mathbf{A}\mathbf{x}^{(t)} - \mathbf{b}) \right\rangle$$

$$\overset{(a)}{=} \sum_{t=1}^{T} \rho^{(t)} f(\mathbf{x}^{(t)}) + \langle \boldsymbol{\lambda}^*, \boldsymbol{\lambda}^{(T)} \rangle$$

$$\overset{(b)}{\leq} \sum_{t=1}^{T} \rho^{(t)} f(\mathbf{x}^{(t)}) + \|\boldsymbol{\lambda}^*\| \|\boldsymbol{\lambda}^{(T)}\|$$

where (a) follows from part (1) of Lemma 3 and (b) follows from the Cauchy-Schwarz inequality. □

## 6.4 New Facts on Convex Analysis

Recall the following important fact on the minimizer of strongly convex functions:

**Lemma 5** (See e.g. Corollary 1 in [30]). *Let $h : \mathcal{X} \to \mathbb{R}$ be a strongly convex function, i.e., $\mu$-convex with $\mu > 0$, and $\mathbf{x}^{min} \in \mathcal{X}$ be a point that minimizes $h$ over set $\mathcal{X}$, then*

$$h(\mathbf{x}^{min}) \leq h(\mathbf{x}) - \frac{\mu}{2} \|\mathbf{x} - \mathbf{x}^{min}\|^2 \quad \forall \mathbf{x} \in \mathcal{X}.$$

Note that this fact holds trivially for convex functions without strong convexity ($\mu$-convex functions with $\mu = 0$). We now extend the above property for a convex function given by $h(\mathbf{x}) = g(\mathbf{x}) + \frac{1}{2}\|\mathbf{x}\|_{\mathbf{Q}}^2$ where $g(\mathbf{x})$ is a $\mu$-convex function and $\mathbf{Q} \succeq \mathbf{0}$ is a symmetric semidefinite positive matrix, in the following lemma:

**Lemma 6.** *Let $h : \mathcal{X} \to \mathbb{R}$ be defined as $h(\mathbf{x}) = g(\mathbf{x}) + \frac{1}{2}\|\mathbf{x}\|_{\mathbf{Q}}^2$ where $g(\mathbf{x})$ is a $\mu$-convex function and $\mathbf{Q} \succeq \mathbf{0}$ is a symmetric semidefinite positive matrix. If $\mathbf{x}^{min} \in \mathcal{X}$ is a point that minimizes $h$ over set $\mathcal{X}$, then*

$$h(\mathbf{x}^{min}) \leq h(\mathbf{x}) - \frac{1}{2} \|\mathbf{x} - \mathbf{x}^{min}\|_{\mathbf{Q}+\mu\mathbf{I}}^2 \quad \forall \mathbf{x} \in \mathcal{X}.$$

Since matrix $\mathbf{Q}$ can be rank deficient, the function $\frac{1}{2}\|\mathbf{x}\|_{\mathbf{Q}}^2$ is not necessarily strongly convex. Thus, $h(\mathbf{x}) = g(\mathbf{x}) + \frac{1}{2}\|\mathbf{x}\|_{\mathbf{Q}}^2$ is in general $\mu$-convex. By Lemma 5, we can only say $h(\mathbf{x}^{min}) \leq h(\mathbf{x}) - \frac{1}{2}\|\mathbf{x} - \mathbf{x}^{min}\|_{\mu\mathbf{I}}^2$ for all $\mathbf{x} \in \mathcal{X}$, which is weaker than the inequality in Lemma 6.

The following lemma will be useful to prove Lemma 6

**Lemma 7.** *Let $h : \mathcal{X} \to \mathbb{R}$ be defined as $h(\mathbf{x}) = g(\mathbf{x}) + \frac{1}{2}\|\mathbf{x}\|_{\mathbf{Q}}^2$ where $g(\mathbf{x})$ is a $\mu$-convex function and $\mathbf{Q} \succeq \mathbf{0}$ is a symmetric semidefinite positive matrix. Let $\partial h(\mathbf{x})$ be the set of all subgradients of $h$ at point $\mathbf{x}$. Then*

$$h(\mathbf{y}) \geq h(\mathbf{x}) + \langle \mathbf{d}, \mathbf{y} - \mathbf{x} \rangle + \frac{1}{2}\|\mathbf{y} - \mathbf{x}\|_{\mathbf{Q}+\mu\mathbf{I}}^2$$

*for all $\mathbf{x}, \mathbf{y} \in \mathcal{X}$ and all $\mathbf{d} \in \partial h(\mathbf{x})$.*

*Proof.* Define $\phi(\mathbf{x}) = h(\mathbf{x}) - \frac{\mu}{2}\|\mathbf{x}\|^2 - \frac{1}{2}\|\mathbf{x}\|_{\mathbf{Q}}^2 = h(\mathbf{x}) - \frac{1}{2}\|\mathbf{x}\|_{\mathbf{Q}+\mu\mathbf{I}}^2$. Since $h(\mathbf{x}) = g(\mathbf{x}) + \frac{1}{2}\|\mathbf{x}\|_{\mathbf{Q}}^2$, we known $\phi(\mathbf{x})$ is a convex function. Let $\partial\phi(\mathbf{x})$ denote the set of all subgradients of $\phi$ at point $\mathbf{x}$, then $\partial\phi(\mathbf{x}) = \partial h(\mathbf{x}) - (\mathbf{Q} + \mu\mathbf{I})\mathbf{x} = \{\mathbf{d} - (\mathbf{Q} + \mu\mathbf{I})\mathbf{x} \mid \mathbf{d} \in \partial h(\mathbf{x})\}$. By convexity of $\phi$, for all $\mathbf{d} \in \partial h(\mathbf{x})$ and all $\mathbf{x}, \mathbf{y} \in \mathcal{X}$, we have

$$\phi(\mathbf{y}) \geq \phi(\mathbf{x}) + \langle \mathbf{d} - (\mathbf{Q} + \mu\mathbf{I})\mathbf{x}, \mathbf{y} - \mathbf{x} \rangle$$
$$= \phi(\mathbf{x}) + \|\mathbf{x}\|_{\mathbf{Q}+\mu\mathbf{I}}^2 + \langle \mathbf{d}, \mathbf{y} - \mathbf{x} \rangle - \langle (\mathbf{Q} + \mu\mathbf{I})\mathbf{x}, \mathbf{y} \rangle$$

Substituting $\phi(\mathbf{x}) = h(\mathbf{x}) - \frac{1}{2}\|\mathbf{x}\|_{\mathbf{Q}+\mu\mathbf{I}}^2$ and $\phi(\mathbf{y}) = h(\mathbf{y}) - \frac{1}{2}\|\mathbf{y}\|_{\mathbf{Q}+\mu\mathbf{I}}^2$ into it and rearranging terms (noting that $\mathbf{Q} + \mu\mathbf{I}$ is symmetric) yields

$$h(\mathbf{y}) \geq h(\mathbf{x}) + \langle \mathbf{d}, \mathbf{y} - \mathbf{x} \rangle + \frac{1}{2}\|\mathbf{y} - \mathbf{x}\|_{\mathbf{Q}+\mu\mathbf{I}}^2$$

□

Now we are ready to prove Lemma 6:

**Proof of Lemma 6:** Fix $\mathbf{x} \in \mathcal{X}$. Note that $h$ is also convex. By the first order optimality condition of convex functions, e.g., Proposition B.24 (f) in [2], there exists $\mathbf{d} \in \partial h(\mathbf{x}^{min})$ such that $\langle \mathbf{d}, \mathbf{x} - \mathbf{x}^{min} \rangle \geq 0$. By Lemma 7, we also have

$$h(\mathbf{x}) \geq h(\mathbf{x}^{min}) + \langle \mathbf{d}, \mathbf{x} - \mathbf{x}^{min} \rangle + \frac{1}{2}\|\mathbf{x} - \mathbf{x}^{min}\|_{\mathbf{Q}+\mu\mathbf{I}}^2$$

$$\overset{(a)}{\geq} h(\mathbf{x}^{min}) + \frac{1}{2}\|\mathbf{x} - \mathbf{x}^{min}\|_{\mathbf{Q}+\mu\mathbf{I}}^2,$$

where $(a)$ follows from the fact that $\langle \mathbf{d}, \mathbf{x} - \mathbf{x}^{min} \rangle \geq 0$.

**Corollary 1.** *Let $\mathbf{c}$ be a fixed constant vector and $h(\mathbf{x}) = g(\mathbf{x}) + \frac{1}{2}\|\mathbf{x} - \mathbf{c}\|^2_{\mathbf{Q}}$ where $g(\mathbf{x})$ is a $\mu$-convex function and $\mathbf{Q} \succeq 0$ is a symmetric semidefinite positive matrix. If $\mathbf{x}^{min} \in \mathcal{X}$ be a point that minimizes $h$ over set $\mathcal{X}$, then*

$$h(\mathbf{x}^{min}) \leq h(\mathbf{x}) - \frac{1}{2}\|\mathbf{x} - \mathbf{x}^{min}\|^2_{\mathbf{Q}+\mu\mathbf{I}} \quad \forall \mathbf{x} \in \mathcal{X}.$$

*Proof.* Let $\tilde{g}(\mathbf{x}) = g(\mathbf{x}) + \frac{1}{2}\|\mathbf{c}\|^2_{\mathbf{Q}} + \langle \mathbf{c}, \mathbf{x} \rangle$. Note that $\tilde{g}(\mathbf{x})$ is $\mu$-convex as long as $g(\mathbf{x})$ is. We further note that $h(\mathbf{x}) = \tilde{g}(\mathbf{x}) + \frac{1}{2}\|\mathbf{x}\|^2_{\mathbf{Q}}$, which is a summation of $\mu$-convex function and $\frac{1}{2}\|\mathbf{x}\|^2_{\mathbf{Q}}$. Thus, this corollary follows directly from Lemma 6. □

### 6.5 Proof of Theorem 3

The proof is built upon Corollary 1 from Section 6.4 and a different interpretation of the $\mathbf{x}^{(t)}$ update in Algorithm 3.

**Lemma 8.** *The update in (20) (Algorithm 3) is equivalent to*

$$\mathbf{x}_i^{(t)} = \underset{\mathbf{x}_i \in \mathcal{X}_i}{argmin} \left\{ f_i(\mathbf{x}_i) + \rho^{(t)} \left\langle \sum_{i=1}^N \mathbf{A}_i \mathbf{x}_i^{(t-1)} - \mathbf{b} + \frac{1}{\rho^{(t)}} \boldsymbol{\lambda}^{(t-1)}, \mathbf{A}_i \mathbf{x}_i - \frac{\mathbf{b}}{N} \right\rangle + \frac{1}{2}\|\mathbf{x}_i - \mathbf{x}_i^{(t-1)}\|^2_{\mathbf{Q}_i^{(t)}} \right\},$$
(27)

*$\forall i \in \{1, 2, \ldots, N\}$ with $\mathbf{Q}_i^{(t)} = \mathbf{P}_i^{(t)} + \rho^{(t)}\mathbf{A}_i^\mathsf{T}\mathbf{A}_i \succeq 0$.*

*Proof.* Note that $\boldsymbol{\lambda}^{(t-1)}$ and $\mathbf{x}_i^{(t-1)}$ are given constants in (27). This lemma follows by noting that (27) is equivalent to

$$\underset{\mathbf{x}_i \in \mathcal{X}_i}{argmin} \left\{ f_i(\mathbf{x}_i) + \rho^{(t)} \left\langle \sum_{i=1}^N \mathbf{A}_i \mathbf{x}_i^{(t-1)} - \mathbf{b} + \frac{1}{\rho^{(t)}} \boldsymbol{\lambda}^{(t-1)}, \mathbf{A}_i \mathbf{x}_i - \frac{\mathbf{b}}{N} \right\rangle + \frac{\rho^{(t)}}{2}\|\mathbf{A}_i(\mathbf{x}_i - \mathbf{x}_i^{(t-1)})\|^2 \right.$$
$$\left. + \frac{1}{2}\|\mathbf{x}_i - \mathbf{x}_i^{(t-1)}\|^2_{\mathbf{P}_i^{(t)}} \right\}$$

$$\overset{(a)}{\Leftrightarrow} \underset{\mathbf{x}_i \in \mathcal{X}_i}{argmin} \left\{ f_i(\mathbf{x}_i) + \rho^{(t)} \left\langle \sum_{i=1}^N \mathbf{A}_i \mathbf{x}_i^{(t-1)} - \mathbf{b} + \frac{1}{\rho^{(t)}} \boldsymbol{\lambda}^{(t-1)}, \mathbf{A}_i(\mathbf{x}_i - \mathbf{x}_i^{(t-1)}) \right\rangle + \frac{\rho^{(t)}}{2}\|\mathbf{A}_i(\mathbf{x}_i - \mathbf{x}_i^{(t-1)})\|^2 \right.$$
$$\left. + \frac{1}{2}\|\mathbf{x}_i - \mathbf{x}_i^{(t-1)}\|^2_{\mathbf{P}_i^{(t)}} \right\}$$

$$\overset{(b)}{\Leftrightarrow} \underset{\mathbf{x}_i \in \mathcal{X}_i}{argmin} \left\{ f_i(\mathbf{x}_i) + \frac{\rho^{(t)}}{2} \left\| \mathbf{A}_i(\mathbf{x}_i - \mathbf{x}_i^{(t-1)}) + \sum_{i=1}^N \mathbf{A}_i \mathbf{x}_i^{(t-1)} - \mathbf{b} + \frac{1}{\rho^{(t)}} \boldsymbol{\lambda}^{(t-1)} \right\|^2 + \frac{1}{2}\|\mathbf{x}_i - \mathbf{x}_i^{(t-1)}\|^2_{\mathbf{P}_i^{(t)}} \right\}$$

$$\Leftrightarrow \underset{\mathbf{x}_i \in \mathcal{X}_i}{argmin} \left\{ f_i(\mathbf{x}_i) + \frac{\rho^{(t)}}{2} \left\| \mathbf{A}_i\mathbf{x}_i + \sum_{j \neq i} \mathbf{A}_j \mathbf{x}_j^{(t-1)} - \mathbf{b} + \frac{1}{\rho^{(t)}} \boldsymbol{\lambda}^{(t-1)} \right\|^2 + \frac{1}{2}\|\mathbf{x}_i - \mathbf{x}_i^{(t-1)}\|^2_{\mathbf{P}_i^{(t)}} \right\}$$

where (a) follows because an argmin solution does not change if we add constant terms to the expression to minimize and (b) follows by completing the square (and adding necessary constant terms for this). □

**Corollary 2.** *The update in (20) (Algorithm 3) is equivalent to*

$$\mathbf{x}^{(t)} = \underset{\mathbf{x} \in \mathcal{X}}{argmin} \left\{ f(\mathbf{x}) + \rho^{(t)} \left\langle \mathbf{A}\mathbf{x}^{(t-1)} - \mathbf{b} + \frac{1}{\rho^{(t)}} \boldsymbol{\lambda}^{(t-1)}, \mathbf{A}\mathbf{x} - \mathbf{b} \right\rangle + \frac{1}{2}\|\mathbf{x} - \mathbf{x}^{(t-1)}\|^2_{\mathbf{Q}^{(t)}} \right\},$$
(28)

*with*

$$\mathbf{Q}^{(t)} \triangleq Diag(\mathbf{Q}_1^{(t)}, \ldots, \mathbf{Q}_N^{(t)}) = Diag(\mathbf{P}_1^{(t)} + \rho^{(t)}\mathbf{A}_1^\mathsf{T}\mathbf{A}_1, \ldots, \mathbf{P}_N^{(t)} + \rho^{(t)}\mathbf{A}_N^\mathsf{T}\mathbf{A}_N)$$
(29)

*Proof.* Note that the update of each $\mathbf{x}_i^{(t)}$ is fully decoupled in (27). That is, $\mathbf{x}^{(t)}$ chosen by Algorithm 3 is to jointly minimize

$$\sum_{i=1}^{N}\left[f_i(\mathbf{x}_i) + \rho^{(t)}\left\langle \sum_{i=1}^{N}\mathbf{A}_i\mathbf{x}_i^{(t-1)} - \mathbf{b} + \frac{1}{\rho^{(t)}}\boldsymbol{\lambda}^{(t-1)}, \mathbf{A}_i\mathbf{x}_i - \frac{\mathbf{b}}{N}\right\rangle + \frac{1}{2}\|\mathbf{x}_i - \mathbf{x}_i^{(t-1)}\|_{\mathbf{Q}_i^{(t)}}^2\right]$$

$$=f(\mathbf{x}) + \rho^{(t)}\left\langle \mathbf{A}\mathbf{x}^{(t-1)} - \mathbf{b} + \frac{1}{\rho^{(t)}}\boldsymbol{\lambda}^{(t-1)}, \mathbf{A}\mathbf{x} - \mathbf{b}\right\rangle + \frac{1}{2}\|\mathbf{x} - \mathbf{x}^{(t-1)}\|_{\mathbf{Q}^{(t)}}^2$$

over set $\mathcal{X} = \prod_{i=1}^{N}\mathbf{x}_i$

$\square$

**Lemma 9.** *Let $\mathbf{x}^*$ be any optimal solution of problem* (1)*. Let $\mathbf{Q}^{(t)}$ be defined in* (29)*. If $\mathbf{P}_i^{(t)} \succeq 0$ and $\rho^{(t)} > 0$ in Algorithm 3 are chosen to satisfy*

$$\mathbf{Q}^{(t)} \succeq \rho^{(t)}\mathbf{A}^{\mathsf{T}}\mathbf{A} \tag{30}$$

*Then, for all $T \geq 1$, Algorithm 3 ensures that*

$$\sum_{t=1}^{T}\rho^{(t)}f(\mathbf{x}^{(t)}) \leq \sum_{t=1}^{T}\rho^{(t)}f(\mathbf{x}^*) + \frac{1}{2}\sum_{t=1}^{T}\rho^{(t)}\Theta^{(t)} - \frac{1}{2}\|\boldsymbol{\lambda}^{(T)}\|^2$$

*where $\Theta^{(t)} \overset{\Delta}{=} \|\mathbf{x}^* - \mathbf{x}^{(t-1)}\|_{\mathbf{Q}^{(t)}}^2 - \|\mathbf{x}^* - \mathbf{x}^{(t)}\|_{\mu\mathbf{I}+\mathbf{Q}^{(t)}}^2$.*

*Proof.* Fix $T \geq 1$. For any $t \in \{1, 2, \ldots, T\}$, by Corollary 2, $\mathbf{x}^{(t)}$ is chosen to minimize $f(\mathbf{x}) + \rho^{(t)}\left\langle \mathbf{A}\mathbf{x}^{(t-1)} - \mathbf{b} + \frac{1}{\rho^{(t)}}\boldsymbol{\lambda}^{(t-1)}, \mathbf{A}\mathbf{x} - \mathbf{b}\right\rangle + \frac{1}{2}\|\mathbf{x} - \mathbf{x}^{(t-1)}\|_{\mathbf{Q}^{(t)}}^2$ over $\mathbf{x} \in \mathcal{X}$. Note that $f(\mathbf{x}) + \rho^{(t)}\left\langle \mathbf{A}\mathbf{x}^{(t-1)} - \mathbf{b} + \frac{1}{\rho^{(t)}}\boldsymbol{\lambda}^{(t-1)}, \mathbf{A}\mathbf{x} - \mathbf{b}\right\rangle$ is $\mu$-convex since $f(\mathbf{x})$ is $\mu$-convex. By Corollary 1 (note that $\mathbf{x}^* \in \mathcal{X}$), we have

$$f(\mathbf{x}^{(t)}) + \rho^{(t)}\left\langle \mathbf{A}\mathbf{x}^{(t-1)} - \mathbf{b} + \frac{1}{\rho^{(t)}}\boldsymbol{\lambda}^{(t-1)}, \mathbf{A}\mathbf{x}^{(t)} - \mathbf{b}\right\rangle + \frac{1}{2}\|\mathbf{x}^{(t)} - \mathbf{x}^{(t-1)}\|_{\mathbf{Q}^{(t)}}^2$$

$$\leq f(\mathbf{x}^*) + \rho^{(t)}\left\langle \mathbf{A}\mathbf{x}^{(t-1)} - \mathbf{b} + \frac{1}{\rho^{(t)}}\boldsymbol{\lambda}^{(t-1)}, \mathbf{A}\mathbf{x}^* - \mathbf{b}\right\rangle + \frac{1}{2}\|\mathbf{x}^* - \mathbf{x}^{(t-1)}\|_{\mathbf{Q}^{(t)}}^2 \tag{31}$$

$$- \frac{1}{2}\|\mathbf{x}^* - \mathbf{x}^{(t)}\|_{\mu\mathbf{I}+\mathbf{Q}^{(t)}}^2$$

$$\overset{(a)}{=} f(\mathbf{x}^*) + \frac{1}{2}\Theta^{(t)} \tag{32}$$

where (a) follows because $\mathbf{A}\mathbf{x}^* - \mathbf{b} = \mathbf{0}$ and $\Theta^{(t)} \overset{\Delta}{=} \|\mathbf{x}^* - \mathbf{x}^{(t-1)}\|_{\mathbf{Q}^{(t)}}^2 - \|\mathbf{x}^* - \mathbf{x}^{(t)}\|_{\mu\mathbf{I}+\mathbf{Q}^{(t)}}^2$.

Recall that by part (2) of Lemma 3, we have

$$\left\langle \boldsymbol{\lambda}^{(t-1)}, \mathbf{A}\mathbf{x}^{(t)} - \mathbf{b}\right\rangle = \frac{1}{2\rho^{(t)}}\left(\|\boldsymbol{\lambda}^{(t)}\|^2 - \|\boldsymbol{\lambda}^{(t-1)}\|^2\right) - \frac{\rho^{(t)}}{2}\|\mathbf{A}\mathbf{x}^{(t)} - \mathbf{b}\|^2 \tag{33}$$

By the basic identity $\langle \mathbf{u}, \mathbf{v}\rangle = \frac{1}{2}\|\mathbf{u}\|_2^2 + \frac{1}{2}\|\mathbf{v}\|^2 - \frac{1}{2}\|\mathbf{u} - \mathbf{v}\|^2$ for any vector $\mathbf{u}, \mathbf{v}$, we have

$$\left\langle \mathbf{A}\mathbf{x}^{(t-1)} - \mathbf{b}, \mathbf{A}\mathbf{x}^{(t)} - \mathbf{b}\right\rangle = \frac{1}{2}\|\mathbf{A}\mathbf{x}^{(t-1)} - \mathbf{b}\|^2 + \frac{1}{2}\|\mathbf{A}\mathbf{x}^{(t)} - \mathbf{b}\|^2 - \frac{1}{2}\|\mathbf{A}(\mathbf{x}^{(t)} - \mathbf{x}^{(t-1)})\|^2 \tag{34}$$

Substituting (33)-(34) into (32) and rearranging terms yields

$$f(\mathbf{x}^{(t)}) \leq f(\mathbf{x}^*) + \frac{1}{2}\Theta^{(t)} + \frac{1}{2\rho^{(t)}}\left(\|\boldsymbol{\lambda}^{(t-1)}\|^2 - \|\boldsymbol{\lambda}^{(t)}\|^2\right) - \frac{\rho^{(t)}}{2}\|\mathbf{A}\mathbf{x}^{(t-1)} - \mathbf{b}\|^2$$

$$+ \frac{\rho^{(t)}}{2}\|\mathbf{A}(\mathbf{x}^{(t)} - \mathbf{x}^{(t-1)})\|^2 - \frac{1}{2}\|\mathbf{x}^{(t)} - \mathbf{x}^{(t-1)}\|_{\mathbf{Q}^{(t)}}^2$$

$$\overset{(a)}{\leq} f(\mathbf{x}^*) + \frac{1}{2}\Theta^{(t)} + \frac{1}{2\rho^{(t)}}\left(\|\boldsymbol{\lambda}^{(t-1)}\|^2 - \|\boldsymbol{\lambda}^{(t)}\|^2\right) - \frac{1}{2}\|\mathbf{x}^{(t)} - \mathbf{x}^{(t-1)}\|_{\mathbf{Q}^{(t)}-\rho^{(t)}\mathbf{A}^{\mathsf{T}}\mathbf{A}}^2$$

$$\overset{(b)}{\leq} f(\mathbf{x}^*) + \frac{1}{2}\Theta^{(t)} + \frac{1}{2\rho^{(t)}}\left(\|\boldsymbol{\lambda}^{(t-1)}\|^2 - \|\boldsymbol{\lambda}^{(t)}\|^2\right)$$

where (a) follows by ignoring the negative term $-\frac{\rho^{(t)}}{2}\|\mathbf{A}\mathbf{x}^{(t-1)} - \mathbf{b}\|^2$ and noting that $\frac{\rho^{(t)}}{2}\|\mathbf{A}(\mathbf{x}^{(t)} - \mathbf{x}^{(t-1)})\|^2 = \frac{1}{2}\|\mathbf{x}^{(t)} - \mathbf{x}^{(t-1)}\|^2_{\rho^{(t)}\mathbf{A}^\mathsf{T}\mathbf{A}}$; and (b) follows because $\mathbf{Q}^{(t)} \succeq \rho^{(t)}\mathbf{A}^\mathsf{T}\mathbf{A}$.

Multiplying $\rho^{(t)}$ on both sides and summing over $t \in \{1, \ldots, T\}$ yields

$$\sum_{t=1}^{T} \rho^{(t)} f(\mathbf{x}^{(t)}) \le \sum_{t=1}^{T} \rho^{(t)} f(\mathbf{x}^*) + \frac{1}{2}\sum_{t=1}^{T}\rho^{(t)}\Theta^{(t)} + \frac{1}{2}\sum_{t=1}^{T}\left(\|\boldsymbol{\lambda}^{(t-1)}\|^2 - \|\boldsymbol{\lambda}^{(t)}\|^2\right)$$

$$\overset{(a)}{=} \sum_{t=1}^{T}\rho^{(t)}f(\mathbf{x}^*) + \frac{1}{2}\sum_{t=1}^{T}\rho^{(t)}\Theta^{(t)} - \frac{1}{2}\|\boldsymbol{\lambda}^{(T)}\|^2$$

where (a) follows by simplifying the telescoping sums and recalling that $\boldsymbol{\lambda}^{(0)} = \mathbf{0}$. $\qquad\square$

The following lemma provides a few practical sufficient conditions that ensure (30)

**Lemma 10.** *The condition* (30) *holds if* **any** *of the following three conditions holds*

1. $\mathbf{P}_i^{(t)} = \nu^{(t)}\mathbf{I} - \rho^{(t)}\mathbf{A}_i^\mathsf{T}\mathbf{A}_i$ *with* $\nu^{(t)} \ge \rho^{(t)}\|\mathbf{A}\|^2$.

2. $\mathbf{P}_i^{(t)} = \nu^{(t)}\mathbf{I}$ *with* $\nu^{(t)} \ge \rho^{(t)}\|\mathbf{A}\|^2$.

3. $\mathbf{P}_i^{(t)} = \nu_i^{(t)}\mathbf{I}$ *with* $\nu_i^{(t)} \ge \rho^{(t)}(N-1)\|\mathbf{A}_i\|_2^2$

*Proof.* Note that (30) holds trivially when the first or the second condition holds. To see (30) holds when $\mathbf{P}_i = \nu_i^{(t)}\mathbf{I}$ with $\nu_i^{(t)} \ge \rho^{(t)}(N-1)\|\mathbf{A}_i\|_2^2$, we note that for any $\mathbf{z} = [\mathbf{z}_1; \ldots, \mathbf{z}_N] \in \mathbb{R}^{\sum_{i=1}^{N} d_i}$,

$$\|\mathbf{z}\|^2_{\rho^{(t)}\mathbf{A}^\mathsf{T}\mathbf{A} - \mathbf{Q}^{(t)}} = \rho^{(t)}\|\sum_{i=1}^{N}\mathbf{A}_i\mathbf{z}_i\|^2 - \sum_{i=1}^{N}\|\mathbf{z}_i\|^2_{\mathbf{P}_i^{(t)} + \rho^{(t)}\mathbf{A}_i^\mathsf{T}\mathbf{A}_i}$$

$$\overset{(a)}{\le} \rho^{(t)}N\sum_{i=1}^{N}\|\mathbf{z}_i\|^2_{\mathbf{A}_i^\mathsf{T}\mathbf{A}_i} - \sum_{i=1}^{N}\|\mathbf{z}_i\|^2_{\mathbf{P}_i^{(t)} + \rho^{(t)}\mathbf{A}_i^\mathsf{T}\mathbf{A}_i}$$

$$= -\sum_{i=1}^{N}\|\mathbf{z}_i\|^2_{\mathbf{P}_i^{(t)} - \rho^{(t)}(N-1)\mathbf{A}_i^\mathsf{T}\mathbf{A}_i}$$

where (a) follows from the Cauchy-Schwarz inequality. $\qquad\square$

**Remark 7.** *Note that the sufficient conditions developed in Lemma 10 are similar to the conditions from [7], under which [7] shows Algorithm 3 with constant $\rho$ and $\mathbf{P}_i$ can ensure $\mathbf{x}^{(t)}$ eventually converge to an optimal solution $\mathbf{x}^*$ and has an $o(1/t)$ non-ergodic convergence rate in the sense $\|\mathbf{x}^{t+1} - \mathbf{x}^{(t)}\|^2 = o(1/t)$. However, work [7] does not establish the convergence rate of objective violations and feasibility violations shown in our Theorem 3. Furthermore, the fast $O(1/T^2)$ convergence for strongly convex case is not considered in [7].*

Now we are ready to prove both parts of the theorem.

1. **Proof of case $\mu = 0$:** Note that $\rho^{(t)} = \rho$ and $\mathbf{P}_i^{(t)} = \mathbf{P}_i$ are chosen to satisfy (30) by Lemma 10. By Lemma 9 (with $\mu = 0$), we have

$$\rho\sum_{t=1}^{T}f(\mathbf{x}^{(t)}) \le \rho T f(\mathbf{x}^*) + \frac{1}{2}\sum_{t=1}^{T}\rho\left(\|\mathbf{x}^* - \mathbf{x}^{(t-1)}\|^2_{\mathbf{Q}} - \|\mathbf{x}^* - \mathbf{x}^{(t)}\|^2_{\mathbf{Q}}\right) - \frac{1}{2}\|\boldsymbol{\lambda}^{(T)}\|^2$$

$$\le \rho T f(\mathbf{x}^*) + \frac{\rho}{2}\|\mathbf{x}^* - \mathbf{x}^{(0)}\|^2_{\mathbf{Q}} - \frac{1}{2}\|\boldsymbol{\lambda}^{(T)}\|^2 \qquad (35)$$

Ignoring the (negative term) $-\frac{1}{2}\|\boldsymbol{\lambda}^{(T)}\|^2$, dividing both sides by $\rho T$, applying Jensen's inequality yields

$$f(\overline{\mathbf{x}}^T) \le f(\mathbf{x}^*) + \frac{1}{2T}\|\mathbf{x}^* - \mathbf{x}^{(0)}\|^2_{\mathbf{Q}},$$

which is (23) of our theorem.

By Lemma 4 (with $\rho^{(t)} = \rho$), we have

$$\rho \sum_{t=1}^{T} f(\mathbf{x}^{(t)}) \geq \rho T f(\mathbf{x}^*) - \|\boldsymbol{\lambda}^*\| \|\boldsymbol{\lambda}^{(T)}\|$$

Combining this with (35) and cancelling the common term yields

$$\|\boldsymbol{\lambda}^{(T)}\|^2 - 2\|\boldsymbol{\lambda}^*\| \|\boldsymbol{\lambda}^{(T)}\| \leq \rho \|\mathbf{x}^* - \mathbf{x}^{(0)}\|_{\mathbf{Q}}^2$$

This quadratic inequality can be further be rewritten as

$$\left( \|\boldsymbol{\lambda}^{(T)}\| - \|\boldsymbol{\lambda}^*\| \right)^2 \leq \|\boldsymbol{\lambda}^*\|^2 + \rho \|\mathbf{x}^* - \mathbf{x}^{(0)}\|_{\mathbf{Q}}^2$$

Thus, we have

$$\|\boldsymbol{\lambda}^{(T)}\| \leq \|\boldsymbol{\lambda}^*\| + \sqrt{\|\boldsymbol{\lambda}^*\|^2 + \rho \|\mathbf{x}^* - \mathbf{x}^{(0)}\|_{\mathbf{Q}}^2}$$

$$\overset{(a)}{\leq} 2\|\boldsymbol{\lambda}^*\| + \sqrt{\rho} \|\mathbf{x}^* - \mathbf{x}^{(0)}\|_{\mathbf{Q}} \tag{36}$$

where (a) follows from the basic inequality $\sqrt{a+b} \leq \sqrt{a} + \sqrt{b}$ for all $a, b \geq 0$.

By part (1) of Lemma 3 (with $\rho^{(t)} = \rho$), we have

$$\rho \sum_{t=1}^{T} \left( \mathbf{A}\mathbf{x}^{(t)} - \mathbf{b} \right) = \boldsymbol{\lambda}^{(T)}$$

Dividing both sides by $\rho T$ and taking the vector $l_2$ norm on both sides yields

$$\|\mathbf{A}\overline{\mathbf{x}}^{(T)} - \mathbf{b}\| \leq \frac{1}{\rho T} \|\boldsymbol{\lambda}^{(T)}\|$$

$$\overset{(a)}{\leq} \frac{1}{T} \frac{2\|\boldsymbol{\lambda}^*\|}{\rho} + \frac{1}{T} \frac{\|\mathbf{x}^* - \mathbf{x}^{(0)}\|_{\mathbf{Q}}}{\sqrt{\rho}}$$

where (a) follows from (36). Note this is (24) of our theorem.

2. **Proof of case** $\mu > 0$: Note that $\rho^{(t)} = \rho t$ and $\mathbf{P}_i^{(t)} = t\rho \|\mathbf{A}\|^2 - t\rho \mathbf{A}_i^{\mathsf{T}} \mathbf{A}_i$ are chosen to satisfy (30) by Lemma 10. By Lemma 9, we have

$$\sum_{t=1}^{T} \rho^{(t)} f(\mathbf{x}^{(t)})$$

$$\leq \sum_{t=1}^{T} \rho^{(t)} f(\mathbf{x}^*) + \frac{1}{2} \sum_{t=1}^{T} \rho^{(t)} \left( \|\mathbf{x}^* - \mathbf{x}^{(t-1)}\|_{\mathbf{Q}^{(t)}}^2 - \|\mathbf{x}^* - \mathbf{x}^{(t)}\|_{\mu\mathbf{I}+\mathbf{Q}^{(t)}}^2 \right) - \frac{1}{2} \|\boldsymbol{\lambda}^{(T)}\|^2 \tag{37}$$

where $\mathbf{Q}^{(t)} = t\rho \|\mathbf{A}\|^2 \mathbf{I}$.

Note that

$$\sum_{t=1}^{T} \rho^{(t)} \left( \|\mathbf{x}^* - \mathbf{x}^{(t-1)}\|_{\mathbf{Q}^{(t)}}^2 - \|\mathbf{x}^* - \mathbf{x}^{(t)}\|_{\mu\mathbf{I}+\mathbf{Q}^{(t)}}^2 \right)$$

$$= \rho \|\mathbf{x}^* - \mathbf{x}^{(0)}\|_{\rho\|\mathbf{A}\|^2\mathbf{I}}^2 - \sum_{t=1}^{T-1} \left( \rho^{(t)} \|\mathbf{x}^* - \mathbf{x}^{(t)}\|_{\mu\mathbf{I}+\mathbf{Q}^{(t)}}^2 - \rho^{(t+1)} \|\mathbf{x}^* - \mathbf{x}^{(t)}\|_{\mathbf{Q}^{(t+1)}}^2 \right) \tag{38}$$

$$- \rho^T \|\mathbf{x}^* - \mathbf{x}^{(t)}\|_{\mu\mathbf{I}+\mathbf{Q}^{(T)}}^2$$

$$= \rho^2 \|\mathbf{A}\|^2 \|\mathbf{x}^* - \mathbf{x}^{(0)}\|^2 - \sum_{t=1}^{T-1} \|\mathbf{x}^* - \mathbf{x}^{(t)}\|_{\rho^{(t)}(\mu\mathbf{I}+\mathbf{Q}^{(t)})-\rho^{(t+1)}\mathbf{Q}^{(t+1)}}^2 - \rho^T \|\mathbf{x}^* - \mathbf{x}^{(t)}\|_{\mu\mathbf{I}+\mathbf{Q}^{(T)}}^2$$

$$\overset{(a)}{\leq} \rho^2 \|\mathbf{A}\|^2 \|\mathbf{x}^* - \mathbf{x}^{(0)}\|^2 \tag{39}$$

where (a) follows by ignoring the negative term $-\rho^T\|\mathbf{x}^* - \mathbf{x}^{(t)}\|^2_{\mu\mathbf{I}+\mathbf{Q}^{(T)}}$ and noting that $\rho^{(t)}(\mu\mathbf{I}+\mathbf{Q}^{(t)}) - \rho^{(t+1)}\mathbf{Q}^{(t+1)} = (\rho t\mu + \rho^2 t^2\|\mathbf{A}\|^2 - \rho^2(t+1)^2\|\mathbf{A}\|^2)\mathbf{I} = \rho(t\mu - \rho(2t+1)\|\mathbf{A}\|^2)\mathbf{I} \succeq \rho\mu(t - \frac{2t+1}{3})\mathbf{I} \succeq 0$ where the first $\succeq$ follows because $\rho \leq \frac{\mu}{3\|\mathbf{A}\|^2}$ by our algorithm parameter selection and the second $\succeq$ follows because $t \geq 1$.

Substituting (39) into (37) yields

$$\sum_{t=1}^{T}\rho^{(t)}f(\mathbf{x}^{(t)}) \leq \sum_{t=1}^{T}\rho^{(t)}f(\mathbf{x}^*) + \frac{\rho^2}{2}\|\mathbf{A}\|^2\|\mathbf{x}^* - \mathbf{x}^{(0)}\|^2 - \frac{1}{2}\|\boldsymbol{\lambda}^{(T)}\|^2 \qquad (40)$$

Ignoring the (negative term) $-\frac{1}{2}\|\boldsymbol{\lambda}^{(T)}\|^2_2$, dividing both sides by $\sum_{t=1}^{T}\rho^{(t)}$, applying Jensen's inequality yields

$$
\begin{aligned}
f(\overline{\mathbf{x}}^T) \leq &f(\mathbf{x}^*) + \frac{\rho^2}{2\sum_{t=1}^{T}\rho^{(t)}}\|\mathbf{A}\|^2\|\mathbf{x}^* - \mathbf{x}^{(0)}\|^2 \\
\overset{(a)}{=} &f(\mathbf{x}^*) + \frac{\rho}{T(T+1)}\|\mathbf{A}\|^2\|\mathbf{x}^* - \mathbf{x}^{(0)}\|^2
\end{aligned}
$$

where (a) follows because $\sum_{t=1}^{T}\rho^{(t)} = \rho\sum_{t=1}^{T}t = \rho\frac{T(T+1)}{2}$. Note this is (25) of our theorem. By Lemma 4, we have

$$\sum_{t=1}^{T}\rho^{(t)}f(\mathbf{x}^{(t)}) \geq \sum_{t=1}^{T}\rho^{(t)}f(\mathbf{x}^*) - \|\boldsymbol{\lambda}^*\|\|\boldsymbol{\lambda}^{(T)}\|$$

Combining this with (40) and cancelling the common term yields

$$\|\boldsymbol{\lambda}^{(T)}\|^2 - 2\|\boldsymbol{\lambda}^*\|\|\boldsymbol{\lambda}^{(T)}\| \leq \rho^2\|\mathbf{A}\|^2\|\mathbf{x}^* - \mathbf{x}^{(0)}\|^2$$

This quadratic inequality can be further be rewritten as

$$\left(\|\boldsymbol{\lambda}^{(T)}\| - \|\boldsymbol{\lambda}^*\|\right)^2 \leq \|\boldsymbol{\lambda}^*\|^2 + \rho^2\|\mathbf{A}\|^2\|\mathbf{x}^* - \mathbf{x}^{(0)}\|^2$$

Thus, we have

$$
\begin{aligned}
\|\boldsymbol{\lambda}^{(T)}\| \leq &\|\boldsymbol{\lambda}^*\| + \sqrt{\|\boldsymbol{\lambda}^*\|^2 + \rho^2\|\mathbf{A}\|^2\|\mathbf{x}^* - \mathbf{x}^{(0)}\|^2} \\
\overset{(a)}{\leq} &2\|\boldsymbol{\lambda}^*\| + \rho\|\mathbf{A}\|\|\mathbf{x}^* - \mathbf{x}^{(0)}\|
\end{aligned}
\qquad (41)
$$

where (a) follows from the basic inequality $\sqrt{a+b} \leq \sqrt{a} + \sqrt{b}$ for all $a, b \geq 0$.

By part (1) of Lemma 3, we have

$$\sum_{t=1}^{T}\rho^{(t)}\left(\mathbf{A}\mathbf{x}^{(t)} - \mathbf{b}\right) = \boldsymbol{\lambda}^{(T)}$$

Dividing both sides by $\sum_{t=1}^{T}\rho^{(t)}$ and taking the vector $l_2$ norm on both sides yields

$$
\begin{aligned}
\|\mathbf{A}\overline{\mathbf{x}}^{(T)} - \mathbf{b}\| = &\frac{1}{\sum_{t=1}^{T}\rho^{(t)}}\|\boldsymbol{\lambda}^{(T)}\| \\
\overset{(a)}{\leq} &\frac{4\|\boldsymbol{\lambda}^*\|}{\rho T(T+1)} + \frac{2\|\mathbf{A}\|\|\mathbf{x}^* - \mathbf{x}^{(0)}\|}{T(T+1)}
\end{aligned}
$$

where (a) follows from (41) and the fact that $\sum_{t=1}^{T}\rho^{(t)} = \rho\sum_{t=1}^{T}t = \rho\frac{T(T+1)}{2}$. Note this is (26) of our theorem.

## 6.6 Proof of Lemma 2

Fix $\mathbf{z} \in \mathcal{Z}$. At each iteration $k$, the projected gradient update (6) in Algorithm 2 can be rewritten as

$$\mathbf{z}^{(k)} = \underset{\mathbf{z} \in \mathcal{Z}}{\operatorname{argmin}} \ \{\langle \boldsymbol{\zeta}^{(k)}, \mathbf{z} - \mathbf{z}^{(k-1)} \rangle + \frac{1}{2\gamma^{(k)}} \|\mathbf{z} - \mathbf{z}^{(k-1)}\|^2 \}.$$

Since the objective function is $\frac{1}{\gamma^{(k)}}$-convex, by Lemma 5, we have

$$\langle \boldsymbol{\zeta}^{(k)}, \mathbf{z}^{(k)} - \mathbf{z}^{(k-1)} \rangle + \frac{1}{2\gamma^{(k)}} \|\mathbf{z}^{(k)} - \mathbf{z}^{(k-1)}\|^2 \leq \langle \boldsymbol{\zeta}^{(k)}, \mathbf{z} - \mathbf{z}^{(k-1)} \rangle + \frac{1}{2\gamma^{(k)}} \|\mathbf{z} - \mathbf{z}^{(k-1)}\|^2$$

$$- \frac{1}{2\gamma^{(k)}} \|\mathbf{z} - \mathbf{z}^{(k)}\|^2$$

Adding $\phi(\mathbf{z}^{(k-1)}) + \langle \nabla\phi(\mathbf{z}^{(k-1)}) - \boldsymbol{\zeta}^{(k)}, \mathbf{z}^{(k)} - \mathbf{z}^{(k-1)} \rangle + \frac{L}{2} \|\mathbf{z}^{(k)} - \mathbf{z}^{(k-1)}\|^2$ on both sides and rearranging terms yields

$$\phi(\mathbf{z}^{(k-1)}) + \langle \nabla\phi(\mathbf{z}^{(k-1)}), \mathbf{z}^{(k)} - \mathbf{z}^{(k-1)} \rangle + \frac{L}{2} \|\mathbf{z}^{(k)} - \mathbf{z}^{(k-1)}\|^2$$

$$\leq \phi(\mathbf{z}^{(k-1)}) + \langle \boldsymbol{\zeta}^{(k)}, \mathbf{z} - \mathbf{z}^{(k-1)} \rangle + \frac{1}{2\gamma^{(k)}} \|\mathbf{z} - \mathbf{z}^{(k-1)}\|^2 - \frac{1}{2\gamma^{(k)}} \|\mathbf{z} - \mathbf{z}^{(k)}\|^2$$

$$- \frac{1}{2}(\frac{1}{\gamma^{(k)}} - L)\|\mathbf{z}^{(k)} - \mathbf{z}^{(k-1)}\|^2 + \langle \nabla\phi(\mathbf{z}^{(k-1)}) - \boldsymbol{\zeta}^{(k)}, \mathbf{z}^{(k)} - \mathbf{z}^{(k-1)} \rangle \tag{42}$$

Since $\phi(\cdot)$ is $L$-smooth, by the descent lemma, e.g., Proposition A.24 in [2], we have

$$\phi(\mathbf{z}^{(k)}) \leq \phi(\mathbf{z}^{(k-1)}) + \langle \nabla\phi(\mathbf{z}^{(k-1)}), \mathbf{z}^{(k)} - \mathbf{z}^{(k-1)} \rangle + \frac{L}{2} \|\mathbf{z}^{(k)} - \mathbf{z}^{(k-1)}\|^2 \tag{43}$$

By Young's inequality, for any $\eta^{(k)} > 0$, we have

$$\langle \nabla\phi(\mathbf{z}^{(k-1)}) - \boldsymbol{\zeta}^{(k)}, \mathbf{z}^{(k)} - \mathbf{z}^{(k-1)} \rangle \leq \frac{1}{2\eta^{(k)}} \|\nabla\phi(\mathbf{x}^{k-1}) - \boldsymbol{\zeta}^{(k)}\|^2 + \frac{\eta^{(k)}}{2} \|\mathbf{z}^{(k)} - \mathbf{z}^{(k-1)}\|^2 \tag{44}$$

Substituting (43) and (44) into (42) yields

$$\phi(\mathbf{z}^{(k)}) \leq \phi(\mathbf{z}^{(k-1)}) + \langle \boldsymbol{\zeta}^{(k)}, \mathbf{z} - \mathbf{z}^{(k-1)} \rangle + \frac{1}{2\gamma^{(k)}} \|\mathbf{z} - \mathbf{z}^{(k-1)}\|^2 - \frac{1}{2\gamma^{(k)}} \|\mathbf{z} - \mathbf{z}^{(k)}\|^2$$

$$- \frac{1}{2}(\frac{1}{\gamma^{(k)}} - L - \eta^{(k)})\|\mathbf{z}^{(k)} - \mathbf{z}^{(k-1)}\|^2 + \frac{1}{2\eta^{(k)}} \|\nabla\phi(\mathbf{x}^{k-1}) - \boldsymbol{\zeta}^{(k)}\|^2 \tag{45}$$

For any fixed $\mathbf{z}$, since $\boldsymbol{\zeta}^{(k)}$ is an unbiased i.i.d. stochastic gradient and $\mathbf{z}^{(t-1)}$ is determined by $\boldsymbol{\zeta}^{(0)}, \ldots, \boldsymbol{\zeta}^{(k-1)}$, we have

$$\mathbb{E}[\langle \boldsymbol{\zeta}^{(k)}, \mathbf{z} - \mathbf{z}^{(k-1)} \rangle] = \langle \nabla\phi(\mathbf{z}^{(k-1)}), \mathbf{z} - \mathbf{z}^{(k-1)} \rangle \tag{46}$$

By the bounded variance assumption, we have

$$\mathbb{E}[\|\nabla\phi(\mathbf{x}^{k-1}) - \boldsymbol{\zeta}^{(k)}\|^2] \leq \sigma^2 \tag{47}$$

By the $\mu$-strong convexity of $\phi(\cdot)$, we have

$$\phi(\mathbf{z}^{(k-1)}) + \langle \nabla\phi(\mathbf{z}^{(k-1)}), \mathbf{z} - \mathbf{z}^{(k-1)} \rangle \leq \phi(\mathbf{z}) - \frac{\mu}{2} \|\mathbf{z} - \mathbf{z}^{(k-1)}\|^2 \tag{48}$$

Taking expectations on both sides of (45) and substituting (46)-(48) into it yields

$$\mathbb{E}[\phi(\mathbf{z}^{(k)})] \leq \phi(\mathbf{z}) + \frac{1}{2}(\frac{1}{\gamma^{(k)}} - \mu)\mathbb{E}[\|\mathbf{z} - \mathbf{z}^{(k-1)}\|^2] - \frac{1}{2}\frac{1}{\gamma^{(k)}}\mathbb{E}[\|\mathbf{z} - \mathbf{z}^{(k)}\|^2]$$

$$- \frac{1}{2}(\frac{1}{\gamma^{(k)}} - L - \eta^{(k)})\mathbb{E}[\|\mathbf{z}^{(k)} - \mathbf{z}^{(k-1)}\|^2] + \frac{1}{2\eta^{(k)}}\sigma^2 \tag{49}$$

Note that if we take $\gamma^{(k)} = \frac{2}{\mu(k+k_0)}$, $\eta^{(k)} = \frac{\mu}{2}k$, then $\frac{1}{\gamma^{(k)}} - L - \eta^{(k)} \geq 0$ since $k_0 \geq 2\kappa = 2\frac{L}{\mu}$. Thus, under the current choice of $\gamma^{(k)}$ and $\eta^{(k)}$, (49) implies that

$$\mathbb{E}[\phi(\mathbf{z}^{(k)})] \leq \phi(\mathbf{z}) + \frac{\mu}{4}(k + k_0 - 2)\mathbb{E}[\|\mathbf{z} - \mathbf{z}^{(k-1)}\|^2] - \frac{\mu}{4}(k + k_0)\mathbb{E}[\|\mathbf{z} - \mathbf{z}^{(k)}\|^2] + \frac{1}{k\mu}\sigma^2$$

Multiplying both sides by $k + k_0 - 1$ yields

$$(k + k_0 - 1)\mathbb{E}[\phi(\mathbf{z}^{(k)})] \leq (k + k_0 - 1)\phi(\mathbf{z}) + \frac{\mu}{4}(k + k_0 - 2)(k + k_0 - 1)\mathbb{E}[\|\mathbf{z} - \mathbf{z}^{(k-1)}\|^2]$$
$$- \frac{\mu}{4}(k + k_0 - 1)(k + k_0)\mathbb{E}[\|\mathbf{z} - \mathbf{z}^{(k)}\|^2] + \frac{k + k_0 - 1}{k\mu}\sigma^2$$
$$\leq (k + k_0 - 1)\phi(\mathbf{z}) + \frac{\mu}{4}(k + k_0 - 2)(k + k_0 - 1)\mathbb{E}[\|\mathbf{z} - \mathbf{z}^{(k-1)}\|^2]$$
$$- \frac{\mu}{4}(k + k_0 - 1)(k + k_0)\mathbb{E}[\|\mathbf{z} - \mathbf{z}^{(k)}\|^2] + \frac{k_0}{\mu}\sigma^2$$

Summing over $k \in \{1, 2, \ldots, K\}$ and dividing both sides by $\sum_{k=1}^{K}(k + k_0 - 1)$ yields

$$\mathbb{E}\Big[\frac{1}{\sum_{k=1}^{K}(k + k_0 - 1)}\sum_{k=1}^{K}(k + k_0 - 1)\phi(\mathbf{z}^{(k)})\Big]$$
$$\leq \phi(\mathbf{z}) + \frac{\mu(k_0^2 - k_0)}{2K(K + 2k_0 - 1)}\mathbb{E}[\|\mathbf{z} - \mathbf{z}^{(0)}\|^2] - \frac{\mu(k_0^2 - k_0)}{2K(K + 2k_0 - 1)}\mathbb{E}[\|\mathbf{z} - \mathbf{z}^{(K)}\|^2] - \frac{\mu}{2}\mathbb{E}[\|\mathbf{z} - \mathbf{z}^{(K)}\|^2]$$
$$+ \frac{2k_0\sigma^2}{(K + 2k_0 - 1)\mu}$$

Define $\hat{\mathbf{z}} \triangleq \frac{1}{\sum_{k=1}^{K}(k + k_0 - 1)}(k + k_0 - 1)\mathbf{z}^{(k)}$. By Jensen's inequality, we have

$$\mathbb{E}[\phi(\hat{\mathbf{z}})] \leq \phi(\mathbf{z}) + \frac{\mu(k_0^2 - k_0)}{2K(K + 2k_0 - 1)}\mathbb{E}[\|\mathbf{z} - \mathbf{z}^{(0)}\|^2] - \frac{\mu(k_0^2 - k_0)}{2K(K + 2k_0 - 1)}\mathbb{E}[\|\mathbf{z} - \mathbf{z}^{(K)}\|^2]$$
$$- \frac{\mu}{2}\mathbb{E}[\|\mathbf{z} - \mathbf{z}^{(K)}\|^2] + \frac{2k_0\sigma^2}{(K + 2k_0 - 1)\mu}$$

## 6.7  Proof of Theorem 1

For convenience of our presentation, we extract the assumption in Theorem 1 and call it Assumption 2:

**Assumption 2.** *Convex program 1 satisfies the following:*

1. *The constraint set $\mathcal{X}$ is bounded, i.e., there exists constant $R > 0$ such that $\|\mathbf{x}\| \leq R, \forall \mathbf{x} \in \mathcal{X}$.*

2. *The function $f(\mathbf{x})$ has unbiased stochastic subgradients with a bounded second order moment, i.e., there exists constant $D > 0$ such that $\mathbb{E}_\xi[\|\mathbf{G}(\mathbf{x}; \xi)\|^2] \leq D^2, \forall \mathbf{x} \in \mathcal{X}$.*

**Lemma 11.** *Consider convex program (1) under Assumption 2. Let $\mathbf{x}^*$ be any optimal solution. If $\nu^{(t)} > 0$ and $\rho^{(t)} > 0$ in Algorithm 1 are chosen to satisfy*

$$\nu^{(t)} \geq \rho^{(t)}\|\mathbf{A}\|^2, \forall t,$$

*and the sub-procedure STO-LOCAL (Algorithm 2) uses $\hat{\mathbf{z}}$ defined in Lemma 1 as the output then, for all $T \geq 1$, Algorithm 1 ensures*

$$\sum_{t=1}^{T}\mathbb{E}[\rho^{(t)}f(\mathbf{x}^{(t)})] \leq \sum_{t=1}^{T}\mathbb{E}[\rho^{(t)}f(\mathbf{x}^*)] + \frac{1}{2}\sum_{t=1}^{T}\mathbb{E}[\rho^{(t)}\Lambda^{(t)}] - \frac{1}{2}\mathbb{E}[\|\boldsymbol{\lambda}^{(T)}\|^2] + \sum_{t=1}^{T}\frac{2\rho^{(t)}(B^{(t)})^2}{\nu^{(t)}(K^{(t)} + 1)}$$

*with*

$$\Gamma^{(t)} \triangleq \nu^{(t)}\|\mathbf{x}^* - \mathbf{y}^{(t-1)}\|^2 - \nu^{(t)}\|\mathbf{x}^* - \mathbf{y}^{(t)}\|^2, \tag{50}$$

*and*

$$(B^{(t)})^2 \triangleq 2\|\mathbf{A}\|^2\mathbb{E}[\|\boldsymbol{\lambda}^{(t-1)}\|^2] + 6D^2 + 6(\rho^{(t)})^2\|\mathbf{A}\|^2(\|\mathbf{A}\|R + \|\mathbf{b}\|)^2 + 24(\nu^{(t)})^2R^2 \tag{51}$$

*where $D$ and $R$ are constants defined in Assumption 2.*

*Proof.* Fix $t \in \{1, 2, \ldots, T\}$. Define $\phi^{(t)}(\mathbf{x}) = \sum_{i=1}^{N} \phi_i^{(t)}(\mathbf{x}_i)$. Since $\phi^{(t)}(\mathbf{x})$ is separable with respect to each $\mathbf{x}_i$, the fact that Algorithm 1 updates each $\mathbf{x}_i$ and $\mathbf{y}_i$ locally and in parallel by calling sub-procedure $(\mathbf{x}_i^{(t)}, \mathbf{y}_i^{(t)}) = \text{STO-LOCAL}(\phi_i^{(t)}(\cdot), \mathcal{X}_i, \mathbf{y}_i^{(t-1)}, K^{(t)})$ can be interpreted as all $N$ nodes jointly update $\mathbf{x}$ and $\mathbf{y}$ via calling sub-procedure $(\mathbf{x}^{(t)}, \mathbf{y}^{(t)}) = \text{STO-LOCAL}(\phi^{(t)}(\cdot), \mathcal{X}, \mathbf{y}^{(t-1)}, K^{(t)})$. (Note that the synchronization of parallel sub-procedures $\text{STO-LOCAL}(\phi_i^{(t)}(\cdot), \mathcal{X}_i, \mathbf{y}_i^{(t-1)}, K^{(t)})$ is not needed since these sub-procedures are fully decoupled. We just need to aggregate the variables with the same index together and write it into the above compact form.)

For each $i \in \{1, 2, \ldots, N\}$, the unbiased stochastic subgradient used in each iteration $k \in \{1, 2, \ldots, K^{(t)}\}$ of Algorithm 2 is given by

$$\boldsymbol{\zeta}_i^{(k)} = \mathbf{G}_i^{(k)} + \mathbf{A}_i^{\mathsf{T}}(\rho^{(t)}(\mathbf{A}\mathbf{y}^{(t-1)} - \mathbf{b}) + \boldsymbol{\lambda}^{(t-1)}) + \nu^{(t)}(\mathbf{z}_i^{(k)} - \mathbf{y}_i^{(t-1)})$$

where $\mathbf{G}_i^{(k)}$ is an unbiased stochastic subgradient for $f_i(\mathbf{x}_i)$ at point $\mathbf{x}_i = \mathbf{z}_i^{(k)}$.

Define $\boldsymbol{\zeta}^{(k)} = [\boldsymbol{\zeta}_1^{(k)}; \ldots; \boldsymbol{\zeta}_N^{(k)}] = \mathbf{G}^{(k)} + \mathbf{A}^{\mathsf{T}}(\rho^{(t)}(\mathbf{A}\mathbf{y}^{(t-1)} - \mathbf{b}) + \boldsymbol{\lambda}^{(t-1)}) + \nu^{(t)}(\mathbf{z}^{(k)} - \mathbf{y}^{(t-1)}), \forall k \in \{1, 2, \ldots, K^{(t)}\}$. Then, $\boldsymbol{\zeta}^{(k)}$ is the unbiased stochastic subgradient used in each iteration of the joint sub-procedure $(\mathbf{x}^{(t)}, \mathbf{y}^{(t)}) = \text{STO-LOCAL}(\phi^{(t)}(\cdot), \mathcal{X}, \mathbf{y}^{(t-1)}, K^{(t)})$.

Note that

$$\mathbb{E}[\|\boldsymbol{\zeta}^{(k)}\|^2] \overset{(a)}{\leq} 2\mathbb{E}[\|\mathbf{A}^{\mathsf{T}}\boldsymbol{\lambda}^{(t-1)}\|^2] + 6(\mathbb{E}[\|\mathbf{G}^{(k)}\|^2] + 6\mathbb{E}[\|\rho^{(t)}\mathbf{A}^{\mathsf{T}}(\mathbf{A}\mathbf{y}^{(t-1)} - \mathbf{b})\|^2]$$
$$+ 6\mathbb{E}[\|\nu^{(t)}(\mathbf{z}^{(k)} - \mathbf{y}^{(t-1)})\|^2])$$
$$\overset{(b)}{\leq} 2\|\mathbf{A}\|^2 \mathbb{E}[\|\boldsymbol{\lambda}^{(t-1)}\|^2] + 6D^2 + 6(\rho^{(t)})^2\|\mathbf{A}\|^2(\|\mathbf{A}\|R + \|\mathbf{b}\|)^2 + 24(\nu^{(t)})^2 R^2$$
$$\overset{(c)}{=} (B^{(t)})^2$$

where (a) follows from the basic inequality $\|\mathbf{v}_1 + \mathbf{v}_2 + \mathbf{v}_3 + \mathbf{v}_4\|^2 \leq 2\|\mathbf{v}_1\|^2 + 6\|\mathbf{v}_2\|^2 + 6\|\mathbf{v}_3\|^2 + 6\|\mathbf{v}_4\|^2$, which can be easily shown by noting that $\|\mathbf{v}_1 + \mathbf{v}_2 + \mathbf{v}_3 + \mathbf{v}_4\|^2 \leq 2\|\mathbf{v}_1\|^2 + 2\|\mathbf{v}_2 + \mathbf{v}_3 + \mathbf{v}_4\|^2 \leq 2\|\mathbf{v}_1\|^2 + 2 \cdot (3\|\mathbf{v}_2\|^2 + 3\|\mathbf{v}_3\|^2 + 3\|\mathbf{v}_4\|^2)$; (b) follows from Assumption 2 and basic matrix norm inequalities; and (c) follows from the definition of $B^{(t)}$ in (51).

Since $\phi^{(t)}(\cdot)$ is $\nu^{(t)}$-convex (with $\nu^{(t)} > 0$), by Lemma 1, we have

$$\mathbb{E}[\phi^{(t)}(\mathbf{x}^{(t)})] \leq \mathbb{E}[\phi^{(t)}(\mathbf{x}^*)] - \frac{\nu^{(t)}}{2}\mathbb{E}[\|\mathbf{y}^{(t)} - \mathbf{x}^*\|^2] + \frac{2(B^{(t)})^2}{\nu^{(t)}(K^{(t)} + 1)}.$$

Substituting the expression of $\phi_i^{(t)}(\cdot)$ (defined in (2)) into the above equation yields

$$\mathbb{E}[f(\mathbf{x}^{(t)})] + \rho^{(t)}\mathbb{E}[\langle \mathbf{A}\mathbf{y}^{(t-1)} - \mathbf{b}, \mathbf{A}\mathbf{x}^{(t)} - \mathbf{b}\rangle] + \mathbb{E}[\langle \boldsymbol{\lambda}^{(t-1)}, \mathbf{A}\mathbf{x}^{(t)} - \mathbf{b}\rangle]$$
$$+ \frac{\nu^{(t)}}{2}\mathbb{E}[\|\mathbf{x}^{(t)} - \mathbf{y}^{(t-1)}\|^2]$$
$$\leq \mathbb{E}[f(\mathbf{x}^*)] + \rho^{(t)}\mathbb{E}[\langle \mathbf{A}\mathbf{y}^{(t-1)} - \mathbf{b}, \mathbf{A}\mathbf{x}^* - \mathbf{b}\rangle] + \mathbb{E}[\langle \boldsymbol{\lambda}^{(t-1)}, \mathbf{A}\mathbf{x}^* - \mathbf{b}\rangle] + \frac{\nu^{(t)}}{2}\mathbb{E}[\|\mathbf{x}^* - \mathbf{y}^{(t-1)}\|^2]$$
$$- \frac{\nu^{(t)}}{2}\mathbb{E}[\|\mathbf{x}^* - \mathbf{y}^{(t)}\|^2] + \frac{2(B^{(t)})^2}{\nu^{(t)}(K^{(t)} + 1)}$$
$$\overset{(a)}{=} \mathbb{E}[f(\mathbf{x}^*)] + \frac{1}{2}\mathbb{E}[\Gamma^{(t)}] + \frac{2(B^{(t)})^2}{\nu^{(t)}(K^{(t)} + 1)} \tag{52}$$

where (a) follows because $\mathbf{A}\mathbf{x}^* - \mathbf{b} = \mathbf{0}$ and $\Gamma^{(t)} = \nu^{(t)}\|\mathbf{x}^* - \mathbf{y}^{(t-1)}\|^2 - \nu^{(t)}\|\mathbf{x}^* - \mathbf{y}^{(t)}\|^2$.

Recall that by part (2) of Lemma 3, we have

$$\left\langle \boldsymbol{\lambda}^{(t-1)}, \mathbf{A}\mathbf{x}^{(t)} - \mathbf{b}\right\rangle = \frac{1}{2\rho^{(t)}}\left(\|\boldsymbol{\lambda}^{(t)}\|^2 - \|\boldsymbol{\lambda}^{(t-1)}\|^2\right) - \frac{\rho^{(t)}}{2}\|\mathbf{A}\mathbf{x}^{(t)} - \mathbf{b}\|^2 \tag{53}$$

By the basic identity $\langle \mathbf{u}, \mathbf{v} \rangle = \frac{1}{2}\|\mathbf{u}\|_2^2 + \frac{1}{2}\|\mathbf{v}\|^2 - \frac{1}{2}\|\mathbf{u} - \mathbf{v}\|^2$ for any vector $\mathbf{u}, \mathbf{v}$, we have

$$\left\langle \mathbf{A}\mathbf{y}^{(t-1)} - \mathbf{b}, \mathbf{A}\mathbf{x}^{(t)} - \mathbf{b} \right\rangle = \frac{1}{2}\|\mathbf{A}\mathbf{y}^{(t-1)} - \mathbf{b}\|^2 + \frac{1}{2}\|\mathbf{A}\mathbf{x}^{(t)} - \mathbf{b}\|^2 - \frac{1}{2}\|\mathbf{A}(\mathbf{x}^{(t)} - \mathbf{y}^{(t-1)})\|^2$$
(54)

Substituting (53) and (54) into (52) and rearranging terms yields

$$\mathbb{E}[f(\mathbf{x}^{(t)})] \leq \mathbb{E}[f(\mathbf{x}^*)] + \frac{1}{2}\mathbb{E}[\Lambda^{(t)}] + \frac{1}{2\rho^{(t)}}\mathbb{E}[\|\boldsymbol{\lambda}^{(t-1)}\|^2 - \|\boldsymbol{\lambda}^{(t)}\|^2] - \frac{\rho^{(t)}}{2}\mathbb{E}[\|\mathbf{A}\mathbf{y}^{(t-1)} - \mathbf{b}\|^2]$$

$$+ \frac{\rho^{(t)}}{2}\mathbb{E}[\|\mathbf{A}(\mathbf{x}^{(t)} - \mathbf{y}^{(t-1)})\|^2] - \frac{\nu^{(t)}}{2}\mathbb{E}[\|\mathbf{x}^{(t)} - \mathbf{y}^{(t-1)}\|^2] + \frac{2(B^{(t)})^2}{\nu^{(t)}(K^{(t)} + 1)}$$

$$\overset{(a)}{\leq} \mathbb{E}[f(\mathbf{x}^*)] + \frac{1}{2}\mathbb{E}[\Gamma^{(t)}] + \frac{1}{2\rho^{(t)}}\mathbb{E}[\|\boldsymbol{\lambda}^{(t-1)}\|^2 - \|\boldsymbol{\lambda}^{(t)}\|^2] + \frac{2(B^{(t)})^2}{\nu^{(t)}(K^{(t)} + 1)}$$

where (a) follows by ignoring the negative term $-\frac{\rho^{(t)}}{2}\mathbb{E}[\|\mathbf{A}\mathbf{y}^{(t-1)} - \mathbf{b}\|^2]$ and noting that $\rho^{(t)}\|\mathbf{A}(\mathbf{x}^{(t)} - \mathbf{y}^{(t-1)})\|^2 - \nu^{(t)}\|\mathbf{x}^{(t)} - \mathbf{y}^{(t-1)}\|^2 \leq 0$ for any $\mathbf{x}^{(t)}$ and $\mathbf{y}^{(t-1)}$ as long as $\nu^{(t)} \geq \rho^{(t)}\|\mathbf{A}\|^2$.

Multiplying both sides by $\rho^{(t)}$ and summing over $t \in \{1, 2, \ldots, T\}$ yields

$$\sum_{t=1}^{T} \mathbb{E}[\rho^{(t)}f(\mathbf{x}^{(t)})] \leq \sum_{t=1}^{T} \mathbb{E}[\rho^{(t)}f(\mathbf{x}^*)] + \frac{1}{2}\sum_{t=1}^{T} \mathbb{E}[\rho^{(t)}\Gamma^{(t)}] + \frac{1}{2}\sum_{t=1}^{T} \mathbb{E}[\|\boldsymbol{\lambda}^{(t-1)}\|^2 - \|\boldsymbol{\lambda}^{(t)}\|^2]$$

$$+ \sum_{t=1}^{T} \frac{2\rho^{(t)}(B^{(t)})^2}{\nu^{(t)}(K^{(t)} + 1)}$$

$$\overset{(a)}{=} \sum_{t=1}^{T} \mathbb{E}[\rho^{(t)}f(\mathbf{x}^*)] + \frac{1}{2}\sum_{t=1}^{T} \mathbb{E}[\rho^{(t)}\Gamma^{(t)}] - \frac{1}{2}\mathbb{E}[\|\boldsymbol{\lambda}^{(T)}\|^2] + \sum_{t=1}^{T} \frac{2\rho^{(t)}(B^{(t)})^2}{\nu^{(t)}(K^{(t)} + 1)}$$

where (a) follows by simplifying the telescoping sum and recalling that $\boldsymbol{\lambda}^{(0)} = \mathbf{0}$. $\qquad \square$

**Lemma 12.** *Consider convex program (1) under Assumption 1-2. Let $(\mathbf{x}^*, \boldsymbol{\lambda}^*)$ be any saddle point defined in Assumption 1. For all $T \geq 1$, if we choose any fixed $\rho^{(t)} = \rho > 0$, $\nu^{(t)} = \nu > 8\rho\|\mathbf{A}\|^2$, $K^t = K \geq T$ in Algorithm 1 and the sub-procedure STO-LOCAL (Algorithm 2) uses $\widehat{\mathbf{z}}$ defined in Lemma 1 as the output, then we have*

$$\mathbb{E}[\|\boldsymbol{\lambda}^{(t)}\|^2] \leq Q, \forall t \in \{0, 1, \ldots, T\}$$

*where*

$$Q \triangleq \left( \frac{2\|\boldsymbol{\lambda}^*\| + \sqrt{\rho\nu\|\mathbf{x}^* - \mathbf{y}^{(0)}\|^2 + \frac{24\rho D^2}{\nu} + \frac{24(\rho)^3\|\mathbf{A}\|^2(\|\mathbf{A}\|R + \|\mathbf{b}\|)^2}{\nu} + 96\nu\rho R^2}}{1 - \sqrt{\frac{8\rho\|\mathbf{A}\|^2}{\nu}}} \right)^2$$

*is an absolute constant (independent of $T$) with constants $R, D$ defined in Assumption 2.*

*Proof.* We prove this lemma by inductions. Note that $\boldsymbol{\lambda}^{(0)} = \mathbf{0}$ trivially satisfies $\mathbb{E}[\|\boldsymbol{\lambda}^{(0)}\|^2] \leq Q$. Assume $\mathbb{E}[\|\boldsymbol{\lambda}^{(t)}\|^2] \leq Q$ holds for all $t \leq t_0$ with $0 \leq t_0 \leq T - 1$ and consider $t = t_0 + 1$.

By Lemma 4, we have

$$\sum_{t=1}^{t_0+1} \rho f(\mathbf{x}^*) - \sum_{t=1}^{t_0+1} \rho f(\mathbf{x}^{(t)}) \leq \|\boldsymbol{\lambda}^*\|\|\boldsymbol{\lambda}^{(t_0+1)}\|$$

Taking expectations on both sides yields

$$\sum_{t=1}^{t_0+1} \mathbb{E}[\rho f(\mathbf{x}^*)] - \sum_{t=1}^{t_0+1} \mathbb{E}[\rho f(\mathbf{x}^{(t)})] \leq \|\boldsymbol{\lambda}^*\|\mathbb{E}[\|\boldsymbol{\lambda}^{(t_0+1)}\|] \overset{(a)}{\leq} \|\boldsymbol{\lambda}^*\|\sqrt{\mathbb{E}[\|\boldsymbol{\lambda}^{(t_0+1)}\|^2]} \qquad (55)$$

where (a) follows because $\mathbb{E}[X^2] \geq (\mathbb{E}[X])^2$ for any random variable $X$.

Note that our selection of $\rho^{(t)} = \rho$ and $\nu^{(t)} = \nu > 8\rho\|\mathbf{A}\|^2$ satisfies the condition in Lemma 11. By Lemma 11, we have

$$\sum_{t=1}^{t_0+1} \mathbb{E}[\rho f(\mathbf{x}^{(t)})]$$

$$\leq \sum_{t=1}^{t_0+1} \mathbb{E}[\rho f(\mathbf{x}^*)] + \frac{\rho\nu}{2} \sum_{t=1}^{t_0+1} \mathbb{E}[\|\mathbf{x}^* - \mathbf{y}^{(t-1)}\|^2 - \|\mathbf{x}^* - \mathbf{y}^{(t)}\|^2] - \frac{1}{2}\mathbb{E}[\|\boldsymbol{\lambda}^{(t_0+1)}\|^2] + \sum_{t=1}^{t_0+1} \frac{2\rho(B^{(t)})^2}{\nu(K+1)}$$

$$\leq \sum_{t=1}^{t_0+1} \mathbb{E}[\rho f(\mathbf{x}^*)] + \frac{\rho\nu}{2}\|\mathbf{x}^* - \mathbf{y}^{(0)}\|^2 - \frac{1}{2}\mathbb{E}[\|\boldsymbol{\lambda}^{(t_0+1)}\|^2] + \sum_{t=1}^{t_0+1} \frac{2\rho(B^{(t)})^2}{\nu(K+1)}$$

Rearranging terms yields

$$\mathbb{E}[\|\boldsymbol{\lambda}^{(t_0+1)}\|^2] \leq 2\left(\sum_{t=1}^{t_0+1} \mathbb{E}[\rho f(\mathbf{x}^*)] - \sum_{t=1}^{t_0+1} \mathbb{E}[\rho f(\mathbf{x}^{(t)})]\right) + \rho\nu\|\mathbf{x}^* - \mathbf{y}^{(0)}\|^2 + \sum_{t=1}^{t_0+1} \frac{4\rho(B^{(t)})^2}{\nu(K+1)}$$

$$\overset{(a)}{\leq} 2\|\boldsymbol{\lambda}^*\|\sqrt{\mathbb{E}[\|\boldsymbol{\lambda}^{(t_0+1)}\|^2]} + \rho\nu\|\mathbf{x}^* - \mathbf{y}^{(0)}\|^2 + \sum_{t=1}^{t_0+1} \frac{4\rho(B^{(t)})^2}{\nu(K+1)} \tag{56}$$

where (a) follows by using (55).

Recalling the definition of $(B^{(t)})^2$ in (51) (with $\rho^{(t)} = \rho$ and $\nu^{(t)} = \nu$), we have

$$\sum_{t=1}^{t_0+1} \frac{4\rho(B^{(t)})^2}{\nu(K+1)}$$

$$= \frac{4\rho}{\nu(K+1)} \sum_{t=1}^{t_0+1} \left(2\|\mathbf{A}\|^2\mathbb{E}[\|\boldsymbol{\lambda}^{(t-1)}\|^2] + 6D^2 + 6(\rho)^2\|\mathbf{A}\|^2(\|\mathbf{A}\|R + \|\mathbf{b}\|)^2 + 24(\nu)^2R^2\right)$$

$$\overset{(a)}{\leq} \frac{24\rho D^2}{\nu} + \frac{24(\rho)^3\|\mathbf{A}\|^2(\|\mathbf{A}\|R + \|\mathbf{b}\|)^2}{\nu} + 96\nu\rho R^2 + \frac{8\rho\|\mathbf{A}\|^2}{\nu}Q \tag{57}$$

where (a) follows because $t_0 + 1 \leq (K+1)$ by our selection of $K$ and $\mathbb{E}[\|\boldsymbol{\lambda}^{(t)}\|^2] \leq Q, \forall 0 \leq t \leq t_0$ by induction hypothesis.

Denote $c \triangleq \rho\nu\|\mathbf{x}^* - \mathbf{y}^{(0)}\|^2 + \frac{24\rho D^2}{\nu} + \frac{24(\rho)^3\|\mathbf{A}\|^2(\|\mathbf{A}\|R+\|\mathbf{b}\|)^2}{\nu} + 96\nu\rho R^2$. Note that $Q = \left(\frac{2\|\boldsymbol{\lambda}^*\| + \sqrt{c}}{1 - \sqrt{\frac{8\rho\|\mathbf{A}\|^2}{\nu}}}\right)^2$. Substituting (57) into (56) yields

$$\mathbb{E}[\|\boldsymbol{\lambda}^{(t_0+1)}\|^2]] \leq 2\|\boldsymbol{\lambda}^*\|\sqrt{\mathbb{E}[\|\boldsymbol{\lambda}^{(t_0+1)}\|^2]} + c + \frac{8\rho\|\mathbf{A}\|^2}{\nu}Q.$$

This can be rewritten as

$$\left(\sqrt{\mathbb{E}[\|\boldsymbol{\lambda}^{(t_0+1)}\|^2]} - \|\boldsymbol{\lambda}^*\|\right)^2 \leq \|\boldsymbol{\lambda}^*\|^2 + c + \frac{8\rho\|\mathbf{A}\|^2}{\nu}Q,$$

which further implies that

$$\sqrt{\mathbb{E}[\|\boldsymbol{\lambda}^{(t_0+1)}\|^2]} \leq \|\boldsymbol{\lambda}^*\| + \sqrt{\|\boldsymbol{\lambda}^*\|^2 + c + \frac{8\rho\|\mathbf{A}\|^2}{\nu}Q}$$

$$\overset{(a)}{\leq} 2\|\boldsymbol{\lambda}^*\| + \sqrt{c} + \sqrt{\frac{8\rho\|\mathbf{A}\|^2}{\nu}}\sqrt{Q}$$

$$= \frac{2\|\boldsymbol{\lambda}^*\| + \sqrt{c}}{1 - \sqrt{\frac{8\rho\|\mathbf{A}\|^2}{\nu}}}$$

where (a) follows from the basic inequality $\sqrt{a_1 + a_2 + a_3} \leq \sqrt{a_1} + \sqrt{a_2} + \sqrt{a_3}$ for all $a_1, a_2, a_3 \geq 0$; and (b) follows by substituting $Q = \left( \frac{2\|\boldsymbol{\lambda}^*\| + \sqrt{c}}{1 - \sqrt{\frac{8\rho\|\mathbf{A}\|^2}{\nu}}} \right)^2$.

Squaring both sides yields

$$\mathbb{E}[\|\boldsymbol{\lambda}^{(t_0+1)}\|^2] \leq \left( \frac{2\|\boldsymbol{\lambda}^*\| + \sqrt{c}}{1 - \sqrt{\frac{8\rho\|\mathbf{A}\|^2}{\nu}}} \right)^2 = Q$$

By far, we have shown $\mathbb{E}[\|\boldsymbol{\lambda}^{(t)}\|^2] \leq Q$ for $t = t_0 + 1$. Thus, this lemma follows by inductions. $\square$

**Main proof of Theorem 1:** Now, we are ready to prove the theorem. Fix $T \geq 1$. By Lemma 11,

$$\sum_{t=1}^{T} \mathbb{E}[\rho f(\mathbf{x}^{(t)})]$$

$$\leq \sum_{t=1}^{Ts} \mathbb{E}[\rho f(\mathbf{x}^*)] + \frac{\rho\nu}{2} \sum_{t=1}^{T} \mathbb{E}[\|\mathbf{x}^* - \mathbf{y}^{(t-1)}\|^2 - \|\mathbf{x}^* - \mathbf{y}^{(t)}\|^2] - \frac{1}{2}\mathbb{E}[\|\boldsymbol{\lambda}^{(T)}\|^2] + \sum_{t=1}^{T} \frac{2\rho(B^{(t)})^2}{\nu(K+1)}$$

$$\leq \sum_{t=1}^{t_0+1} \mathbb{E}[\rho f(\mathbf{x}^*)] + \frac{\rho\nu}{2}\|\mathbf{x}^* - \mathbf{y}^{(0)}\|^2 + \sum_{t=1}^{T} \frac{2\rho(B^{(t)})^2}{\nu(K+1)} \tag{58}$$

Dividing both sides by $\rho T$ and applying Jensen's inequality yields

$$\mathbb{E}f(\overline{\mathbf{x}}^T)] \leq f(\mathbf{x}^*)] + \frac{\nu}{2T}\|\mathbf{x}^* - \mathbf{y}^{(0)}\|^2 + \frac{2}{\nu T}\sum_{t=1}^{T} \frac{(B^{(t)})^2}{K+1} \tag{59}$$

Note that for all $t \in \{1, 2, \ldots, T\}$, we have

$$\frac{(B^{(t)})^2}{K+1} = \frac{2\|\mathbf{A}\|^2 \mathbb{E}[\|\boldsymbol{\lambda}^{(t-1)}\|^2] + 6D^2 + 6\rho^2\|\mathbf{A}\|^2(\|\mathbf{A}\|R + \|\mathbf{b}\|)^2 + 24\nu^2 R^2}{K+1}$$

$$\overset{(a)}{\leq} \frac{2\|\mathbf{A}\|^2 Q + 6D^2 + 6\rho^2\|\mathbf{A}\|^2(\|\mathbf{A}\|R + \|\mathbf{b}\|)^2 + 24\nu^2 R^2}{T} \tag{60}$$

where (a) follows because $\mathbb{E}[\|\boldsymbol{\lambda}^{(t)}\|^2] \leq Q, \forall t \in \{0, 1, \ldots, T\}$ by Lemma 12 and $K \geq T$. Substituting (60) into (59) yields

$$\mathbb{E}f(\overline{\mathbf{x}}^T)] \leq f(\mathbf{x}^*) + \frac{\nu}{2T}\|\mathbf{x}^* - \mathbf{y}^{(0)}\|^2 + \frac{C}{\nu T}$$

with $C \overset{\Delta}{=} 4\|\mathbf{A}\|^2 Q + 12D^2 + 12\rho^2\|\mathbf{A}\|^2(\|\mathbf{A}\|R + \|\mathbf{b}\|)^2 + 48\nu^2 R^2$. This is (9) of our theorem.

By part (1) of Lemma 3 (with $\rho^{(t)} = \rho$), we have $\sum_{t=1}^{T} \rho\left(\mathbf{A}\mathbf{x}^{(t)} - \mathbf{b}\right) = \boldsymbol{\lambda}^{(T)}$. Dividing both sides by $\rho T$, taking the vector $l_2$ norm and then taking expectations on both sides yields

$$\mathbb{E}[\|\mathbf{A}\overline{\mathbf{x}}^{(T)} - \mathbf{b}\|] = \frac{1}{\rho T}\mathbb{E}[\|\boldsymbol{\lambda}^{(T)}\|]$$

$$\leq \frac{1}{\rho T}\sqrt{\mathbb{E}[\|\boldsymbol{\lambda}^{(T)}\|^2]}$$

$$\overset{(a)}{\leq} \frac{\sqrt{Q}}{\rho T}$$

where (a) follows from Lemma 12. This is (10) of our theorem.

## 6.8 Proof of Theorem 2

For convenience of our presentation, we extract the assumptions in Theorem 2 and call it Assumption 3:

**Assumption 3.** *Convex program* (1) *satisfies the following:*

- *The function $f(\mathbf{x})$ is L-smooth.*

- *The function $f(\mathbf{x})$ has unbiased stochastic gradients with a bounded variance, i.e., there exists constant $\sigma > 0$ such that $\mathbb{E}_\xi[\|\mathbf{G}(\mathbf{x};\xi) - \nabla f(\mathbf{x})\|^2] \leq \sigma^2, \forall \mathbf{x} \in \mathcal{X}.$*

**Lemma 13.** *Consider convex program* (1) *with $\mu$-convex stochastic objective functions under Assumption 3. Let $\mathbf{x}^*$ be any optimal solution. If $\nu^{(t)} > 0$ and $\rho^{(t)} > 0$ in Algorithm 1 are chosen to satisfy*

$$\nu^{(t)} \geq \rho^{(t)} \|\mathbf{A}\|^2, \forall t,$$

*and the sub-procedure STO-LOCAL (Algorithm 2) uses $k_0 \geq 2\frac{\nu^{(t)}+L}{\nu^{(t)}+\mu}, \forall t$ and $\widehat{\mathbf{z}}$ defined in Lemma 2 as the output, then, for all $T \geq 1$, Algorithm 1 ensures*

$$\sum_{t=1}^{T} \mathbb{E}[\rho^{(t)} f(\mathbf{x}^{(t)})]$$

$$\leq \sum_{t=1}^{T} \mathbb{E}[\rho^{(t)} f(\mathbf{x}^*)] + \frac{1}{2}\sum_{t=1}^{T} \mathbb{E}[\rho^{(t)} \Lambda^{(t)}] - \frac{1}{2}\mathbb{E}[\|\boldsymbol{\lambda}^{(T)}\|^2] + \sum_{t=1}^{T} \frac{2\rho^{(t)} k_0 \sigma^2}{(\nu^{(t)} + \mu)(K^{(t)} + 2k_0 - 1)}$$

$$\Lambda^{(t)} \triangleq \left(\nu^{(t)} + \frac{(\nu^{(t)}+\mu)(k_0^2 - k_0)}{K^t(K^t + 2k_0 - 1)}\right)\|\mathbf{x}^* - \mathbf{y}^{(t-1)}\|^2 - \left(\nu^{(t)} + \mu + \frac{(\nu^{(t)}+\mu)(k_0^2 - k_0)}{K^t(K^t + 2k_0 - 1)}\right)\|\mathbf{x}^* - \mathbf{y}^{(t)}\|^2 \tag{61}$$

*and $\sigma^2$ is the constant defined in Assumption 3.*

*Proof.* Fix $t \in \{1, 2, \ldots, T\}$. Define $\phi^{(t)}(\mathbf{x}) = \sum_{i=1}^{N} \phi_i^{(t)}(\mathbf{x}_i)$. Note that $\phi^{(t)}(\mathbf{x})$ is $(L + \nu^{(t)})$-smooth and $(\mu + \nu^{(t)})$-convex. Similarly to the observation in the proof of Lemma 11, each iteration of Algorithm 1 is to jointly update $\mathbf{x}$ and $\mathbf{y}$ via the sub-procedure $(\mathbf{x}^{(t)}, \mathbf{y}^{(t)}) = $ STO-LOCAL$(\phi^{(t)}(\cdot), \mathcal{X}, \mathbf{y}^{(t-1)}, K^{(t)})$. Note that the stochastic gradient used in each iteration of the sub-procedure is given by

$$\boldsymbol{\zeta}^{(k)} = \mathbf{G}^{(k)} + \mathbf{A}^{\mathsf{T}}(\rho^{(t)}(\mathbf{A}\mathbf{y}^{(t-1)} - \mathbf{b}) + \nu^{(t)}(\mathbf{z}^{(k)} - \mathbf{y}^{(t-1)}))$$

and has the same variance bound $\sigma^2$ as the stochastic gradient of $f(\mathbf{x})$.

By Lemma 2, we have

$$\mathbb{E}[\phi^{(t)}(\mathbf{x}^{(t)})]$$

$$\leq \mathbb{E}[\phi^{(t)}(\mathbf{x}^*)] + \frac{(\nu^{(t)}+\mu)(k_0^2 - k_0)}{2K^{(t)}(K^{(t)} + 2k_0 - 1)}\mathbb{E}[\|\mathbf{x}^* - \mathbf{y}^{(t-1)}\|^2] - \frac{(\nu^{(t)}+\mu)(k_0^2 - k_0)}{2K^{(t)}(K^{(t)} + 2k_0 - 1)}\mathbb{E}[\|\mathbf{x}^* - \mathbf{y}^{(t)}\|^2]$$

$$- \frac{\nu^{(t)}+\mu}{2}\mathbb{E}[\|\mathbf{x}^* - \mathbf{y}^{(t)}\|^2] + \frac{2k_0\sigma^2}{(K^{(t)} + 2k_0 - 1)(\nu^{(t)}+\mu)} \tag{62}$$

Substituting the expression of $\phi_i^{(t)}(\cdot)$ (defined in (2)) into the above equation (and recalling $\mathbf{r}^{(t)} = \mathbf{A}\mathbf{y}^{(t-1)} - \mathbf{b}$) yields

$$\mathbb{E}[f(\mathbf{x}^{(t)})] + \rho^{(t)}\mathbb{E}[\langle \mathbf{A}\mathbf{y}^{(t-1)} - \mathbf{b}, \mathbf{A}\mathbf{x}^{(t)} - \mathbf{b}\rangle] + \mathbb{E}[\langle \boldsymbol{\lambda}^{(t-1)}, \mathbf{A}\mathbf{x}^{(t)} - \mathbf{b}\rangle]$$

$$+ \frac{\nu^{(t)}}{2}\mathbb{E}[\|\mathbf{x}^{(t)} - \mathbf{y}^{(t-1)}\|^2]$$

$$\leq \mathbb{E}[f(\mathbf{x}^*)] + \rho^{(t)}\mathbb{E}[\langle \mathbf{A}\mathbf{y}^{(t-1)} - \mathbf{b}, \mathbf{A}\mathbf{x}^* - \mathbf{b}\rangle] + \mathbb{E}[\langle \boldsymbol{\lambda}^{(t-1)}, \mathbf{A}\mathbf{x}^* - \mathbf{b}\rangle] + \frac{\nu^{(t)}}{2}\mathbb{E}[\|\mathbf{x}^* - \mathbf{y}^{(t-1)}\|^2]$$

$$+ \frac{(\nu^{(t)} + \mu)(k_0^2 - k_0)}{2K^{(t)}(K^{(t)} + 2k_0 - 1)}\mathbb{E}[\|\mathbf{x}^* - \mathbf{y}^{(t-1)}\|^2] - \frac{(\nu^{(t)} + \mu)(k_0^2 - k_0)}{2K^{(t)}(K^{(t)} + 2k_0 - 1)}\mathbb{E}[\|\mathbf{x}^* - \mathbf{y}^{(t)}\|^2]$$

$$- \frac{(\nu^{(t)} + \mu)}{2}\mathbb{E}[\|\mathbf{x}^* - \mathbf{y}^{(t)}\|^2] + \frac{2k_0\sigma^2}{(K^{(t)} + 2k_0 - 1)(\nu^{(t)} + \mu)}$$

$$\overset{(a)}{=} \mathbb{E}[f(\mathbf{x}^*)] + \frac{1}{2}\mathbb{E}[\Lambda^{(t)}] + \frac{2k_0\sigma^2}{(K^{(t)} + 2k_0 - 1)(\nu^{(t)} + \mu)} \tag{63}$$

where (a) follows from $\mathbf{A}\mathbf{x}^* - \mathbf{b} = \mathbf{0}$ and the definition of $\Lambda^{(t)}$ in (61).

The remaining part of our proof is almost identical to the proof of Lemma 11. Recall that by part (2) of Lemma 3, we have

$$\left\langle \boldsymbol{\lambda}^{(t-1)}, \mathbf{A}\mathbf{x}^{(t)} - \mathbf{b}\right\rangle = \frac{1}{2\rho^{(t)}}\left(\|\boldsymbol{\lambda}^{(t)}\|^2 - \|\boldsymbol{\lambda}^{(t-1)}\|^2\right) - \frac{\rho^{(t)}}{2}\|\mathbf{A}\mathbf{x}^{(t)} - \mathbf{b}\|^2 \tag{64}$$

By the basic identity $\langle \mathbf{u}, \mathbf{v}\rangle = \frac{1}{2}\|\mathbf{u}\|^2 + \frac{1}{2}\|\mathbf{v}\|^2 - \frac{1}{2}\|\mathbf{u} - \mathbf{v}\|^2$ for any vector $\mathbf{u}, \mathbf{v}$, we have

$$\left\langle \mathbf{A}\mathbf{y}^{(t-1)} - \mathbf{b}, \mathbf{A}\mathbf{x}^{(t)} - \mathbf{b}\right\rangle = \frac{1}{2}\|\mathbf{A}\mathbf{y}^{(t-1)} - \mathbf{b}\|^2 + \frac{1}{2}\|\mathbf{A}\mathbf{x}^{(t)} - \mathbf{b}\|^2 - \frac{1}{2}\|\mathbf{A}(\mathbf{x}^{(t)} - \mathbf{y}^{(t-1)})\|^2 \tag{65}$$

Substituting (64) and (65) into (63) and rearranging terms yields

$$\mathbb{E}[f(\mathbf{x}^{(t)})]$$

$$\leq \mathbb{E}[f(\mathbf{x}^*)] + \frac{1}{2}\mathbb{E}[\Lambda^{(t)}] + \frac{1}{2\rho^{(t)}}\mathbb{E}[\|\boldsymbol{\lambda}^{(t-1)}\|^2 - \|\boldsymbol{\lambda}^{(t)}\|^2] - \frac{\rho^{(t)}}{2}\|\mathbf{A}\mathbf{y}^{(t-1)} - \mathbf{b}\|^2$$

$$+ \frac{\rho^{(t)}}{2}\mathbb{E}[\|\mathbf{A}(\mathbf{x}^{(t)} - \mathbf{y}^{(t-1)})\|^2] - \frac{\nu^{(t)}}{2}\mathbb{E}[\|\mathbf{x}^{(t)} - \mathbf{y}^{(t-1)}\|^2] + \frac{2k_0\sigma^2}{(K^{(t)} + 2k_0 - 1)(\nu^{(t)} + \mu)}$$

$$\overset{(a)}{\leq} \mathbb{E}[f(\mathbf{x}^*)] + \frac{1}{2}\mathbb{E}[\Lambda^{(t)}] + \frac{1}{2\rho^{(t)}}\mathbb{E}[\|\boldsymbol{\lambda}^{(t-1)}\|^2 - \|\boldsymbol{\lambda}^{(t)}\|^2] + \frac{2k_0\sigma^2}{(K^{(t)} + 2k_0 - 1)(\nu^{(t)} + \mu)}$$

where (a) follows by ignoring the negative term $-\frac{\rho^{(t)}}{2}\|\mathbf{A}\mathbf{y}^{(t-1)} - \mathbf{b}\|^2$ and noting that $\rho^{(t)}\|\mathbf{A}(\mathbf{x}^{(t)} - \mathbf{y}^{(t-1)})\|^2 - \nu^{(t)}\|\mathbf{x}^{(t)} - \mathbf{y}^{(t-1)}\|^2 \leq 0$ for any $\mathbf{x}^{(t)}$ and $\mathbf{y}^{(t-1)}$ as long as $\nu^{(t)} \geq \rho^{(t)}\|\mathbf{A}\|^2$.

Multiplying both sides by $\rho^{(t)}$ and summing over $t \in \{1, 2, \ldots, T\}$ yields

$$\sum_{t=1}^{T}\mathbb{E}[\rho^{(t)}f(\mathbf{x}^{(t)})]$$

$$\leq \sum_{t=1}^{T}\mathbb{E}[\rho^{(t)}f(\mathbf{x}^*)] + \frac{1}{2}\sum_{t=1}^{T}\mathbb{E}[\rho^{(t)}\Lambda^{(t)}] + \frac{1}{2}\sum_{t=1}^{T}\mathbb{E}[\|\boldsymbol{\lambda}^{(t-1)}\|^2 - \|\boldsymbol{\lambda}^{(t)}\|^2]$$

$$+ \sum_{t=1}^{T}\frac{2\rho^{(t)}k_0\sigma^2}{(K^{(t)} + 2k_0 - 1)(\nu^{(t)} + \mu)}$$

$$\overset{(a)}{=} \sum_{t=1}^{T}\mathbb{E}[\rho^{(t)}f(\mathbf{x}^*)] + \frac{1}{2}\sum_{t=1}^{T}\mathbb{E}[\rho^{(t)}\Lambda^{(t)}] - \frac{1}{2}\mathbb{E}[\|\boldsymbol{\lambda}^{(T)}\|^2] + \sum_{t=1}^{T}\frac{2\rho^{(t)}k_0\sigma^2}{(K^{(t)} + 2k_0 - 1)(\nu^{(t)} + \mu)}$$

where (a) follows by simplifying the telescoping sums and recalling that $\boldsymbol{\lambda}^{(0)} = \mathbf{0}$. $\qquad\square$

Now we are ready to prove both parts of the theorem.

1. **Proof of case $\mu = 0$:** Fix $T \geq 1$. Note that our selection of $\rho^{(t)} = \rho$, $\nu^{(t)} = \nu \geq \rho\|\mathbf{A}\|^2$ and positive integer $k_0 \geq 2\frac{L+\nu}{\nu}$ satisfies the condition in Lemma 13. By Lemma 13 (with $\mu = 0$), we have

$$\sum_{t=1}^{T} \mathbb{E}[\rho f(\mathbf{x}^{(t)})] \leq \sum_{t=1}^{T} \mathbb{E}[\rho f(\mathbf{x}^*)] + \frac{\rho}{2}\sum_{t=1}^{T}\mathbb{E}[\Lambda^{(t)}] - \frac{1}{2}\mathbb{E}[\|\boldsymbol{\lambda}^{(T)}\|^2] + \sum_{t=1}^{T}\frac{2\rho k_0 \sigma^2}{(T+2k_0-1)\nu}$$
(66)

Note that

$$\begin{aligned}
\sum_{t=1}^{T}\Lambda^{(t)} &= \nu\left(1 + \frac{k_0^2 - k_0}{T(T+2k_0-1)}\right)\sum_{t=1}^{T}\left(\|\mathbf{x}^* - \mathbf{y}^{(t-1)}\|^2 - \|\mathbf{x}^* - \mathbf{y}^{(t)}\|^2\right) \\
&= \nu\left(1 + \frac{k_0^2 - k_0}{T(T+2k_0-1)}\right)\left(\|\mathbf{x}^* - \mathbf{y}^{(0)}\|^2 - \|\mathbf{x}^* - \mathbf{y}^T\|^2\right) \\
&\leq \nu\left(1 + \frac{k_0^2 - k_0}{T(T+2k_0-1)}\right)\|\mathbf{x}^* - \mathbf{y}^{(0)}\|^2
\end{aligned}$$
(67)

and

$$\sum_{t=1}^{T}\frac{2\rho k_0 \sigma^2}{(T+2k_0-1)\nu} = \frac{2\rho k_0 \sigma^2}{\nu}\frac{T}{T+2k_0-1} \leq \frac{2\rho k_0 \sigma^2}{\nu}$$
(68)

Substituting (67)-(68) into (66) yields

$$\begin{aligned}
&\sum_{t=1}^{T}\mathbb{E}[\rho f(\mathbf{x}^{(t)})] \\
&\leq \sum_{t=1}^{T}\mathbb{E}[\rho f(\mathbf{x}^*)] + \frac{\rho\nu}{2}\left(1 + \frac{k_0^2 - k_0}{T(T+2k_0-1)}\right)\|\mathbf{x}^* - \mathbf{y}^{(0)}\|^2 + \frac{2\rho k_0 \sigma^2}{\nu} - \frac{1}{2}\mathbb{E}[\|\boldsymbol{\lambda}^{(T)}\|^2] \\
&\overset{(a)}{\leq} \sum_{t=1}^{T}\mathbb{E}[\rho f(\mathbf{x}^*)] + \frac{\rho\nu(k_0+1)}{4}\|\mathbf{x}^* - \mathbf{y}^{(0)}\|^2 + \frac{2\rho k_0 \sigma^2}{\nu} - \frac{1}{2}\mathbb{E}[\|\boldsymbol{\lambda}^{(T)}\|^2]
\end{aligned}$$
(69)

where (a) follows because $\frac{k_0^2 - k_0}{T(T+2k_0-1)} \leq \frac{k_0-1}{2}$ when $T \geq 1$.

Ignoring the negative term $-\frac{1}{2}\mathbb{E}[\|\boldsymbol{\lambda}^{(T)}\|^2]$, dividing both sides by $\rho T$ and applying Jensen's inequality yields

$$\mathbb{E}[f(\overline{\mathbf{x}}^{(T)})] \leq f(\mathbf{x}^*) + \frac{1}{T}\frac{\nu(k_0+1)}{4}\|\mathbf{x}^* - \mathbf{y}^{(0)}\|^2 + \frac{1}{T}\frac{2k_0\sigma^2}{\nu}$$

This is (11) of our theorem.

By Lemma 4 (after taking expectations on both sides), we have

$$\mathbb{E}[\sum_{t=1}^{T}\rho f(\mathbf{x}^{(t)})] \geq \sum_{t=1}^{T}\mathbb{E}[\rho f(\mathbf{x}^*)] - \|\boldsymbol{\lambda}^*\|\mathbb{E}[\|\boldsymbol{\lambda}^{(T)}\|]$$

Combining this inequality with (69) and cancelling the common terms yields

$$\mathbb{E}[\|\boldsymbol{\lambda}^{(T)}\|^2] \leq 2\|\boldsymbol{\lambda}^*\|\mathbb{E}[\|\boldsymbol{\lambda}^{(T)}\|] + \frac{\rho\nu(k_0+1)}{2}\|\mathbf{x}^* - \mathbf{y}^{(0)}\|^2 + \frac{4\rho k_0 \sigma^2}{\nu}$$

Since $\left(\mathbb{E}[\|\boldsymbol{\lambda}^{(T)}\|]\right)^2 \leq \mathbb{E}[\|\boldsymbol{\lambda}^{(T)}\|^2]$, we further have

$$\left(\mathbb{E}[\|\boldsymbol{\lambda}^{(T)}\|]\right)^2 \leq 2\|\boldsymbol{\lambda}^*\|\mathbb{E}[\|\boldsymbol{\lambda}^{(T)}\|] + \frac{\rho\nu(k_0+1)}{2}\|\mathbf{x}^* - \mathbf{y}^{(0)}\|^2 + \frac{4\rho k_0 \sigma^2}{\nu}$$

This quadratic inequality can be rewritten as

$$\left(\mathbb{E}[\|\boldsymbol{\lambda}^{(T)}\|] - \|\boldsymbol{\lambda}^*\|\right)^2 \leq \|\boldsymbol{\lambda}^*\|^2 + \frac{\rho\nu(k_0+1)}{2}\|\mathbf{x}^* - \mathbf{y}^{(0)}\|^2 + \frac{4\rho k_0\sigma^2}{\nu}$$

Thus, we have

$$\mathbb{E}[\|\boldsymbol{\lambda}^{(T)}\|] \leq \|\boldsymbol{\lambda}^*\| + \sqrt{\|\boldsymbol{\lambda}^*\|^2 + \frac{\rho\nu(k_0+1)}{2}\|\mathbf{x}^* - \mathbf{y}^{(0)}\|^2 + \frac{4\rho k_0\sigma^2}{\nu}}$$

$$\leq 2\|\boldsymbol{\lambda}^*\| + \sqrt{\frac{\rho\nu(k_0+1)}{2}\|\mathbf{x}^* - \mathbf{y}^{(0)}\|^2 + \frac{4\rho k_0\sigma^2}{\nu}} \tag{70}$$

$$\leq 2\|\boldsymbol{\lambda}^*\| + \sqrt{\frac{\rho\nu(k_0+1)}{2}}\|\mathbf{x}^* - \mathbf{y}^{(0)}\| + \sqrt{\frac{4\rho k_0\sigma^2}{\nu}} \tag{71}$$

By part (1) of Lemma 3 (with $\rho^{(t)} = \rho$), we have

$$\sum_{t=1}^{T} \rho\left(\mathbf{A}\mathbf{x}^{(t)} - \mathbf{b}\right) = \boldsymbol{\lambda}^{(T)}$$

Dividing both sides by $\rho T$, taking the vector $l_2$ norm and then taking expectations on both sides yields

$$\mathbb{E}[\|\mathbf{A}\overline{\mathbf{x}}^{(T)} - \mathbf{b}\|] = \frac{1}{\rho T}\mathbb{E}[\|\boldsymbol{\lambda}^{(T)}\|]$$

$$\overset{(a)}{\leq} \frac{1}{T}\left(\frac{2}{\rho}\|\boldsymbol{\lambda}^*\| + \sqrt{\frac{\nu(k_0+1)}{2\rho}}\|\mathbf{x}^* - \mathbf{y}^{(0)}\| + 2\sqrt{\frac{k_0\sigma^2}{\rho\nu}}\right)$$

where (a) follows from (71). This is (78) of our theorem.

2. **Proof of Case $\mu > 0$:** Note that our selection of $\rho^{(t)} = t\rho$, $\nu^{(t)} = t\rho\|\mathbf{A}\|^2$ and $k_0 \geq 2(1 + \frac{L}{\mu})$ satisfies the conditions in Lemma 13. By Lemma 13, we have

$$\sum_{t=1}^{T}\mathbb{E}[\rho^{(t)}f(\mathbf{x}^{(t)})] \leq \sum_{t=1}^{T}\mathbb{E}[\rho^{(t)}f(\mathbf{x}^*)] + \frac{1}{2}\sum_{t=1}^{T}\mathbb{E}[\rho^{(t)}\Lambda^{(t)}] - \frac{1}{2}\mathbb{E}[\|\boldsymbol{\lambda}^{(T)}\|^2]$$

$$+ \sum_{t=1}^{T}\frac{2\rho^{(t)}k_0\sigma^2}{(\nu^{(t)} + \mu)(K^{(t)} + 2k_0 - 1)} \tag{72}$$

Recalling the definition of $\Lambda^t$ in (61), we have

$$\sum_{t=1}^{T}\rho^{(t)}\Lambda^{(t)}$$

$$= \rho\left(\rho\|\mathbf{A}\|^2 + \frac{(\rho\|\mathbf{A}\|^2 + \mu)(k_0^2 - k_0)}{2(2k_0-1)^2}\right)\|\mathbf{x}^* - \mathbf{y}^{(0)}\|^2$$

$$- \sum_{t=1}^{T-1}\left(\rho\left(\rho t^2\|\mathbf{A}\|^2 + t\mu - \rho(t+1)^2\|\mathbf{A}\|^2\right) + \rho\left(\frac{\rho t\|\mathbf{A}\|^2 + \mu}{t+1} - \frac{\rho(t+1)\|\mathbf{A}\|^2 + \mu}{t+2}\right)\frac{k_0^2 - k_0}{(2k_0-1)^2}\right)\|\mathbf{x}^* - \mathbf{y}^{(t)}\|^2$$

$$- T\rho\left(T\rho\|\mathbf{A}\|^2 + \mu + \left(\frac{\rho T\|\mathbf{A}\|^2 + \mu}{T(T+1)}\right)\frac{k_0^2 - k_0}{(2k_0-1)^2}\right)\|\mathbf{x}^* - \mathbf{y}^{(T)}\|^2$$

$$\overset{(a)}{\leq} \rho\left(\rho\|\mathbf{A}\|^2 + \frac{(\rho\|\mathbf{A}\|^2 + \mu)(k_0^2 - k_0)}{2(2k_0-1)^2}\right)\|\mathbf{x}^* - \mathbf{y}^{(0)}\|^2 \tag{73}$$

where (a) follows by ignoring the last negative term and noting that $\rho t^2\|\mathbf{A}\|^2 + t\mu - \rho(t+1)^2\|\mathbf{A}\|^2 = t\mu - \rho(2t+1)\|\mathbf{A}\|^2 \geq \mu(t - \frac{2t+1}{3}) \geq 0$ for all $t \geq 1$, where the first inequality uses $\rho \leq \frac{\mu}{3\|\mathbf{A}\|^2}$; and $\frac{\rho t\|\mathbf{A}\|^2 + \mu}{t+1} - \frac{\rho(t+1)\|\mathbf{A}\|^2 + \mu}{t+2} = \frac{\mu - \rho\|\mathbf{A}\|^2}{(t+1)(t+2)} \geq 0$, where the inequality also uses $\rho \leq \frac{\mu}{3\|\mathbf{A}\|^2}$.

We also note that

$$\sum_{t=1}^{T} \frac{2\rho^{(t)} k_0 \sigma^2}{(\nu^{(t)} + \mu)(K^{(t)} + 2k_0 - 1)} = \frac{2k_0 \sigma^2 \rho}{2k_0 - 1} \sum_{t=1}^{T} \frac{t}{(\rho t \|\mathbf{A}\|^2 + \mu)(t+1)}$$

$$\overset{(a)}{\leq} \frac{2k_0 \sigma^2}{2k_0 - 1} \sum_{t=1}^{T} \frac{t}{(t+1)(t+3)\|\mathbf{A}\|^2}$$

$$\leq \frac{2k_0 \sigma^2}{(2k_0 - 1)\|\mathbf{A}\|^2} \sum_{t=1}^{T} \frac{1}{t+1}$$

$$\leq \frac{2k_0 \sigma^2}{(2k_0 - 1)\|\mathbf{A}\|^2} \log(T+1) \qquad (74)$$

where (a) follows because $\mu \geq 3\rho\|\mathbf{A}\|^2$ by $\rho \leq \frac{\mu}{3\|\mathbf{A}\|^2}$.

Substituting (73) and (74) into (72) yields

$$\sum_{t=1}^{T} \mathbb{E}[\rho^{(t)} f(\mathbf{x}^{(t)})]$$

$$\leq \sum_{t=1}^{T} \mathbb{E}[\rho^{(t)} f(\mathbf{x}^*)] + \frac{1}{2}\rho \left( \rho\|\mathbf{A}\|^2 + \frac{(\rho\|\mathbf{A}\|^2 + \mu)(k_0^2 - k_0)}{2(2k_0 - 1)^2} \right) \|\mathbf{x}^* - \mathbf{y}^{(0)}\|^2 - \frac{1}{2}\mathbb{E}[\|\boldsymbol{\lambda}^{(T)}\|^2]$$

$$+ \frac{2k_0 \sigma^2}{(2k_0 - 1)\|\mathbf{A}\|^2} \log(T+1)$$

$$\overset{(a)}{\leq} \sum_{t=1}^{T} \mathbb{E}[\rho^{(t)} f(\mathbf{x}^*)] + \frac{1}{2}\rho c_1 \|\mathbf{x}^* - \mathbf{y}^{(0)}\|^2 - \frac{1}{2}\mathbb{E}[\|\boldsymbol{\lambda}^{(T)}\|^2] + \frac{1}{2}c_2 \log(T+1) \qquad (75)$$

where (a) follows because $c_1 = \rho\|\mathbf{A}\|^2 + \frac{(\rho\|\mathbf{A}\|^2 + \mu)(k_0^2 - k_0)}{2(2k_0 - 1)^2}$ and $c_2 = \frac{4k_0 \sigma^2}{(2k_0 - 1)\|\mathbf{A}\|^2}$.

Ignoring the negative term $-\frac{1}{2}\mathbb{E}[\|\boldsymbol{\lambda}^{(T)}\|^2]$, dividing both sides by $\sum_{t=1}^{T} \rho^{(t)}$ and applying Jensen's inequality yields

$$\mathbb{E}[f(\overline{\mathbf{x}}^{(T)})] \leq f(\mathbf{x}^*) + \frac{1}{2\sum_{t=1}^{T} \rho^{(t)}} \left( \rho c_1 \|\mathbf{x}^* - \mathbf{y}^{(0)}\|^2 + c_2 \log(T+1) \right)$$

$$\overset{(a)}{=} f(\mathbf{x}^*) + \frac{1}{T(T+1)} \left( c_1 \|\mathbf{x}^* - \mathbf{y}^{(0)}\|^2 + \frac{c_2}{\rho} \log(T+1) \right)$$

where (a) follows because $\sum_{t=1}^{T} \rho^{(t)} = \rho \sum_{t=1}^{T} t = \frac{\rho T(T+1)}{2}$. This is (77) of our theorem.

By Lemma 4 (after taking expectations on both sides), we have

$$\mathbb{E}[\sum_{t=1}^{T} \rho^{(t)} f(\mathbf{x}^{(t)})] \geq \sum_{t=1}^{T} \mathbb{E}[\rho^{(t)} f(\mathbf{x}^*)] - \|\boldsymbol{\lambda}^*\| \mathbb{E}[\|\boldsymbol{\lambda}^{(T)}\|]$$

Combining this inequality with (75) and cancelling the common terms yields

$$\mathbb{E}[\|\boldsymbol{\lambda}^{(T)}\|^2] \leq 2\|\boldsymbol{\lambda}^*\| \mathbb{E}[\|\boldsymbol{\lambda}^{(T)}\|] + \rho c_1 \|\mathbf{x}^* - \mathbf{y}^{(0)}\|^2 + c_2 \log(T+1)$$

Since $\left( \mathbb{E}[\|\boldsymbol{\lambda}^{(T)}\|] \right)^2 \leq \mathbb{E}[\|\boldsymbol{\lambda}^{(T)}\|^2]$, we further have

$$\left( \mathbb{E}[\|\boldsymbol{\lambda}^{(T)}\|] \right)^2 \leq 2\|\boldsymbol{\lambda}^*\| \mathbb{E}[\|\boldsymbol{\lambda}^{(T)}\|] + \rho c_1 \|\mathbf{x}^* - \mathbf{y}^{(0)}\|^2 + c_2 \log(T+1)$$

This quadratic inequality can be rewritten as

$$\left( \mathbb{E}[\|\boldsymbol{\lambda}^{(T)}\|] - \|\boldsymbol{\lambda}^*\| \right)^2 \leq \|\boldsymbol{\lambda}^*\|^2 + \rho c_1 \|\mathbf{x}^* - \mathbf{y}^{(0)}\|^2 + c_2 \log(T+1)$$

Thus, we have

$$\mathbb{E}[\|\boldsymbol{\lambda}^{(T)}\||] \leq \|\boldsymbol{\lambda}^*\| + \sqrt{\|\boldsymbol{\lambda}^*\|^2 + \rho c_1\|\mathbf{x}^* - \mathbf{y}^{(0)}\|^2 + c_2\log(T+1)}$$

$$\leq 2\|\boldsymbol{\lambda}^*\| + \sqrt{\rho c_1}\|\mathbf{x}^* - \mathbf{y}^{(0)}\| + \sqrt{c_2\log(T+1)} \tag{76}$$

By part (1) of Lemma 3 (with $\rho^{(t)} = \rho$), we have

$$\sum_{t=1}^{T} \rho\left(\mathbf{A}\mathbf{x}^{(t)} - \mathbf{b}\right) = \boldsymbol{\lambda}^{(T)}$$

Dividing both sides by $\sum_{t=1}^{T} \rho^{(t)} = \rho\frac{T(T+1)}{2}$, taking the vector $l_2$ norm and then taking expectations on both sides yields

$$\mathbb{E}[\|\mathbf{A}\overline{\mathbf{x}}^{(T)} - \mathbf{b}\|] = \frac{2}{T(T+1)}\mathbb{E}[\|\boldsymbol{\lambda}^{(T)}\||]$$

$$\overset{(a)}{\leq} \frac{2}{T(T+1)}\left(\frac{4\|\boldsymbol{\lambda}^*\|}{\rho} + \frac{\sqrt{c_1}}{\sqrt{\rho}}\|\mathbf{x}^* - \mathbf{y}^{(0)}\| + \frac{\sqrt{c_2\log(T+1)}}{\rho}\right)$$

where (a) follows from (76). This is (78) of our theorem.

## 6.9 Performance of Algorithm 1 for strongly convex non-smooth problems

In this subsection, we consider stochastic convex program (1) under the the following assumption.

**Assumption 4.** *Convex program* (1) *satisfies the following:*

- *The function $f(\mathbf{x})$ satisfies* (15).

- *The function $f(\mathbf{x})$ has unbiased stochastic subgradients with a bounded second order moment, i.e., there exists constant $D > 0$ such that $\mathbb{E}_\xi[\|\mathbf{G}(\mathbf{x};\xi)\|^2] \leq D^2, \forall\mathbf{x} \in \mathcal{X}$.*

**Theorem 4.** *Consider convex program* (1) *with $\mu$-convex ($\mu > 0$) possibly non-smooth function under Assumption 1 and Assumption 4. Let $(\mathbf{x}^*, \boldsymbol{\lambda}^*)$ be any saddle point defined in Assumption 1.*

*If the sub-procedure STO-LOCAL (Algorithm 2) uses $\widehat{\mathbf{z}} \triangleq \frac{1}{\sum_{k=1}^{K}(k+k_0-1)}(k+k_0-1)\mathbf{z}^{(k)}$ as the output and $\rho \leq \frac{\mu}{3\|\mathbf{A}\|^2}, \rho^{(t)} = t\rho, \nu^{(t)} = t\rho\|\mathbf{A}\|^2, k_0 = 2$ and $K^t = 3t$ in Algorithm 1, then for all $T \geq 1$,*

$$\mathbb{E}[f(\overline{\mathbf{x}}^T)] \leq f(\mathbf{x}^*) + \frac{1}{T(T+1)}\left(c_1\|\mathbf{x}^* - \mathbf{y}^{(0)}\|^2 + \frac{c_2}{\rho}\log(T+1)\right) \tag{77}$$

$$\mathbb{E}[\|\mathbf{A}\overline{\mathbf{x}}^{(T)} - \mathbf{b}\|] \leq \frac{2}{T(T+1)}\left(\frac{4\|\boldsymbol{\lambda}^*\|}{\rho} + \frac{\sqrt{c_1}}{\sqrt{\rho}}\|\mathbf{x}^* - \mathbf{y}^{(0)}\| + \frac{\sqrt{c_2\log(T+1)}}{\rho}\right) \tag{78}$$

*where $\overline{\mathbf{x}}^{(T)} = \frac{1}{\sum_{t=1}^{T}\rho^{(t)}}\sum_{t=1}^{T}\rho^{(t)}\mathbf{x}^{(t)}$; and $c_1 \triangleq \rho\|\mathbf{A}\|^2 + \frac{2(\rho\|\mathbf{A}\|^2+\mu)}{18}, c_2 \triangleq \frac{16(B^2+M^2)}{3\|\mathbf{A}\|^2}$ are two constants.*

We first develop a lemma that summarizes that Algorithm 2 behaves well as a sub-procedure under Assumption 4. This lemma essentially says Algorithm 2 has good performance when used to minimize a stochastic function that is the sum of a smooth part and a part that satisfying (15).

**Lemma 14.** *Assume $\phi(\mathbf{z}) = \dot{\phi}(\mathbf{z}) + \ddot{\phi}(\mathbf{z})$ where $\dot{\phi}(\mathbf{z})$ is $\mu_1$-convex and satisfies that the assumption that there exists a constant $M > 0$ such that*

$$\dot{\phi}(\mathbf{z}_1) \leq \dot{\phi}(\mathbf{z}_2) + \langle\mathbf{d}, \mathbf{z}_1 - \mathbf{z}_2\rangle + M\|\mathbf{z}_1 - \mathbf{z}_2\|, \tag{79}$$

*for all $\mathbf{z}_1, \mathbf{z}_2 \in \mathcal{Z}$ and $\mathbf{d} \in \partial\dot{\phi}(\mathbf{z}_1)$; and $\ddot{\phi}(\mathbf{z})$ is a $L$-smooth and $\mu_2$-convex deterministic function ($\mu_2 > 0$) over set $\mathcal{Z}$ with conditional number $\kappa = \frac{L}{\mu_2} = 1$. Assume there exists a constant $B$ such that the unbiased subgradient $\boldsymbol{\zeta}^{(k)}$ used in Algorithm 2 satisfy*

$$\mathbb{E}[\|\boldsymbol{\zeta}^{(k)}\|^2] \leq B^2, \forall k \in \{1, 2, \ldots, K\}$$

*If we take $k_0 = 2$ in Algorithm 2, then we have*

$$\mathbb{E}[\phi(\widehat{\mathbf{z}})] \leq \phi(\mathbf{z}) + \frac{\mu}{K(K+3)}\mathbb{E}[\|\mathbf{z} - \mathbf{z}^{(0)}\|^2] - \frac{\mu}{K(K+3)}\mathbb{E}[\|\mathbf{z} - \mathbf{z}^{(K)}\|^2] - \frac{\mu}{2}\mathbb{E}[\|\mathbf{z} - \mathbf{z}^{(K)}\|^2]$$

$$+ \frac{8(B^2 + M^2)}{(K+3)\mu} \tag{80}$$

*where $\widehat{\mathbf{z}} \triangleq \frac{1}{\sum_{k=1}^{K}(k+k_0-1)}(k+k_0-1)\mathbf{z}^{(k)}$ and $\mu \triangleq \mu_1 + \mu_2$*

*Proof.* Fix $\mathbf{z} \in \mathcal{Z}$. At each iteration $k$, the projected gradient update (6) in Algorithm 2 can be rewritten as

$$\mathbf{z}^{(k)} = \underset{\mathbf{z} \in \mathcal{Z}}{\operatorname{argmin}} \{\langle \boldsymbol{\zeta}^{(k)}, \mathbf{z} - \mathbf{z}^{(k-1)} \rangle + \frac{1}{2\gamma^{(k)}}\|\mathbf{z} - \mathbf{z}^{(k-1)}\|^2\}.$$

Since the objective function is $\frac{1}{\gamma^{(k)}}$-convex, by Lemma 5, we have

$$\langle \boldsymbol{\zeta}^{(k)}, \mathbf{z}^{(k)} - \mathbf{z}^{(k-1)} \rangle + \frac{1}{2\gamma^{(k)}}\|\mathbf{z}^{(k)} - \mathbf{z}^{(k-1)}\|^2$$

$$\leq \langle \boldsymbol{\zeta}^{(k)}, \mathbf{z} - \mathbf{z}^{(k-1)} \rangle + \frac{1}{2\gamma^{(k)}}\|\mathbf{z} - \mathbf{z}^{(k-1)}\|^2 - \frac{1}{2\gamma^{(k)}}\|\mathbf{z} - \mathbf{z}^{(k)}\|^2$$

Since $\boldsymbol{\zeta}^{(k)}$ is an unbiased stochastic subgradient of $\phi(\mathbf{z})$ at $\mathbf{z} = \mathbf{z}^{(k-1)}$ and $\ddot{\phi}(\mathbf{z})$ is a deterministic function, we have $\mathbb{E}[\boldsymbol{\zeta}^{(k)}] = \mathbf{d} + \nabla\ddot{\phi}(\mathbf{z}^{(k-1)})$ for some $\mathbf{d} \in \partial\dot{\phi}(\mathbf{z}^{(k-1)})$.

Adding $\phi(\mathbf{z}^{(k-1)}) + \langle \mathbf{d} + \nabla\ddot{\phi}(\mathbf{z}^{(k-1)}) - \boldsymbol{\zeta}^{(k)}, \mathbf{z}^{(k)} - \mathbf{z}^{(k-1)} \rangle + \frac{L}{2}\|\mathbf{z}^{(k)} - \mathbf{z}^{(k-1)}\|^2 + M\|\mathbf{z}^{(k)} - \mathbf{z}^{(k-1)}\|$ on both sides and rearranging terms yields

$$\phi(\mathbf{z}^{(k-1)}) + \langle \mathbf{d} + \nabla\ddot{\phi}(\mathbf{z}^{(k-1)}), \mathbf{z}^{(k)} - \mathbf{z}^{(k-1)} \rangle + \frac{L}{2}\|\mathbf{z}^{(k)} - \mathbf{z}^{(k-1)}\|^2 + M\|\mathbf{z}^{(k)} - \mathbf{z}^{(k-1)}\|$$

$$\leq \phi(\mathbf{z}^{(k-1)}) + \langle \boldsymbol{\zeta}^{(k)}, \mathbf{z} - \mathbf{z}^{(k-1)} \rangle + \frac{1}{2\gamma^{(k)}}\|\mathbf{z} - \mathbf{z}^{(k-1)}\|^2 - \frac{1}{2\gamma^{(k)}}\|\mathbf{z} - \mathbf{z}^{(k)}\|^2$$

$$- \frac{1}{2}(\frac{1}{\gamma^{(k)}} - L)\|\mathbf{z}^{(k)} - \mathbf{z}^{(k-1)}\|^2 + M\|\mathbf{z}^{(k)} - \mathbf{z}^{(k-1)}\| + \langle \mathbb{E}[\boldsymbol{\zeta}^{(k)}] - \boldsymbol{\zeta}^{(k)}, \mathbf{z}^{(k)} - \mathbf{z}^{(k-1)} \rangle \tag{81}$$

Since $\ddot{\phi}(\cdot)$ is $L$-smooth, by the descent lemma, e.g., Proposition A.24 in [2], we have

$$\ddot{\phi}(\mathbf{z}^{(k)}) \leq \ddot{\phi}(\mathbf{z}^{(k-1)}) + \langle \nabla\ddot{\phi}(\mathbf{z}^{(k-1)}), \mathbf{z}^{(k)} - \mathbf{z}^{(k-1)} \rangle + \frac{L}{2}\|\mathbf{z}^{(k)} - \mathbf{z}^{(k-1)}\|^2 \tag{82}$$

By Young's inequality, for any $\eta^{(k)} > 0$, we have

$$\langle \mathbb{E}[\boldsymbol{\zeta}^{(k)}] - \boldsymbol{\zeta}^{(k)}, \mathbf{z}^{(k)} - \mathbf{z}^{(k-1)} \rangle \leq \frac{1}{2\eta^{(k)}}\|\mathbb{E}[\boldsymbol{\zeta}^{(k)}] - \boldsymbol{\zeta}^{(k)}\|^2 + \frac{\eta^{(k)}}{2}\|\mathbf{z}^{(k)} - \mathbf{z}^{(k-1)}\|^2 \tag{83}$$

Substituting (79), (82) and (83) into (81) and recalling that $\phi(\mathbf{z}^{(k)}) = \dot{\phi}(\mathbf{z}^{(k)}) + \ddot{\phi}(\mathbf{z}^{(k)})$ yields

$$\phi(\mathbf{z}^{(k)}) \leq \phi(\mathbf{z}^{(k-1)}) + \langle \boldsymbol{\zeta}^{(k)}, \mathbf{z} - \mathbf{z}^{(k-1)} \rangle + \frac{1}{2\gamma^{(k)}}\|\mathbf{z} - \mathbf{z}^{(k-1)}\|^2 - \frac{1}{2\gamma^{(k)}}\|\mathbf{z} - \mathbf{z}^{(k)}\|^2$$

$$- \frac{1}{2}(\frac{1}{\gamma^{(k)}} - L - \eta^{(k)})\|\mathbf{z}^{(k)} - \mathbf{z}^{(k-1)}\|^2 + M\|\mathbf{z}^{(k)} - \mathbf{z}^{(k-1)}\| + \frac{1}{2\eta^{(k)}}\|\mathbb{E}[\boldsymbol{\zeta}^{(k)}] - \boldsymbol{\zeta}^{(k)}\|^2 \tag{84}$$

Recall that $L = \mu_2$ and $\mu = \mu_1 + \mu_2$. If we take $\gamma^{(k)} = \frac{2}{\mu(k+k_0)}$ with $k_0 = 2$, $\eta^{(k)} = \frac{\mu}{4}k$, then $\frac{1}{\gamma^{(k)}} - L - \eta^{(k)} = \frac{\mu}{4}k + \mu_1 \geq \frac{\mu}{4}k$. Thus, we have

$$- \frac{1}{2}(\frac{1}{\gamma^{(k)}} - L - \eta^{(k)})\|\mathbf{z}^{(k)} - \mathbf{z}^{(k-1)}\|^2 + M\|\mathbf{z}^{(k)} - \mathbf{z}^{(k-1)}\|$$

$$= -\frac{\mu}{8}k\|\mathbf{z}^{(k)} - \mathbf{z}^{(k-1)}\|^2 + M\|\mathbf{z}^{(k)} - \mathbf{z}^{(k-1)}\|$$

$$\overset{(a)}{\leq} \frac{2M^2}{k\mu} \tag{85}$$

where (a) follows from the basic inequality $-au^2 + bu \leq b^2/(4a)$ for any $a < 0$ and $u \in \mathbb{R}$.

Substituting (85) into (84) yields

$$\phi(\mathbf{z}^{(k)}) \leq \phi(\mathbf{z}^{(k-1)}) + \langle \boldsymbol{\zeta}^{(k)}, \mathbf{z} - \mathbf{z}^{(k-1)} \rangle + \frac{1}{2\gamma^{(k)}} \|\mathbf{z} - \mathbf{z}^{(k-1)}\|^2 - \frac{1}{2\gamma^{(k)}} \|\mathbf{z} - \mathbf{z}^{(k)}\|^2 + \frac{2M^2}{k\mu}$$
$$+ \frac{1}{2\eta^{(k)}} \|\mathbb{E}[\boldsymbol{\zeta}^{(k)}] - \boldsymbol{\zeta}^{(k)}\|^2 \tag{86}$$

For any fixed $\mathbf{z}$, since $\boldsymbol{\zeta}^{(k)}$ is an unbiased i.i.d. stochastic subgradient and $\mathbf{z}^{(k-1)}$ is determined by $\boldsymbol{\zeta}^{(0)}, \ldots, \boldsymbol{\zeta}^{(k-1)}$, we have

$$\mathbb{E}[\langle \boldsymbol{\zeta}^{(k)}, \mathbf{z} - \mathbf{z}^{(k-1)} \rangle] = \mathbb{E}[\mathbb{E}[\langle \boldsymbol{\zeta}^{(k)}, \mathbf{z} - \mathbf{z}^{(k-1)} \rangle | \boldsymbol{\zeta}^{[0:k-1]}]] = \mathbb{E}[\langle \mathbf{d} + \nabla \ddot{\phi}(\mathbf{z}^{(k-1)}), \mathbf{z} - \mathbf{z}^{(k-1)} \rangle] \tag{87}$$

By the basic probability fact, we have

$$\mathbb{E}[\|\mathbb{E}[\boldsymbol{\zeta}^{(k)}] - \boldsymbol{\zeta}^{(k)}\|^2] \leq \mathbb{E}[\|\boldsymbol{\zeta}^{(k)}\|^2] \leq B^2 \tag{88}$$

By the $\mu$-convexity of $\phi(\cdot)$, we have

$$\phi(\mathbf{z}^{(k-1)}) + \langle \mathbf{d} + \nabla \ddot{\phi}(\mathbf{z}^{(k-1)}), \mathbf{z} - \mathbf{z}^{(k-1)} \rangle \leq \phi(\mathbf{z}) - \frac{\mu}{2} \|\mathbf{z} - \mathbf{z}^{(k-1)}\|^2 \tag{89}$$

Taking expectations on both sides of (86) and substituting (87)-(89) into it yields

$$\mathbb{E}[\phi(\mathbf{z}^{(k)})] \leq \phi(\mathbf{z}) + \frac{1}{2}\left(\frac{1}{\gamma^{(k)}} - \mu\right)\mathbb{E}[\|\mathbf{z} - \mathbf{z}^{(k-1)}\|^2] - \frac{1}{2}\frac{1}{\gamma^{(k)}}\mathbb{E}[\|\mathbf{z} - \mathbf{z}^{(k)}\|^2] + \frac{2M^2}{k\mu} + \frac{1}{2\eta^{(k)}}B^2$$
$$= \phi(\mathbf{z}) + \frac{\mu}{4}(k + k_0 - 2)\mathbb{E}[\|\mathbf{z} - \mathbf{z}^{(k-1)}\|^2] - \frac{\mu}{4}(k + k_0)\mathbb{E}[\|\mathbf{z} - \mathbf{z}^{(k)}\|^2] + \frac{2(B^2 + M^2)}{k\mu} \tag{90}$$

Multiplying both sides by $k + k_0 - 1$ yields

$$(k + k_0 - 1)\mathbb{E}[\phi(\mathbf{z}^{(k)})]$$
$$\leq (k + k_0 - 1)\phi(\mathbf{z}) + \frac{\mu}{4}(k + k_0 - 2)(k + k_0 - 1)\mathbb{E}[\|\mathbf{z} - \mathbf{z}^{(k-1)}\|^2]$$
$$- \frac{\mu}{4}(k + k_0 - 1)(k + k_0)\mathbb{E}[\|\mathbf{z} - \mathbf{z}^{(k)}\|^2] + \frac{k + k_0 - 1}{k\mu}(2(B^2 + M^2))$$
$$\overset{(a)}{\leq} (k + k_0 - 1)\phi(\mathbf{z}) + \frac{\mu}{4}(k + k_0 - 2)(k + k_0 - 1)\mathbb{E}[\|\mathbf{z} - \mathbf{z}^{(k-1)}\|^2]$$
$$- \frac{\mu}{4}(k + k_0 - 1)(k + k_0)\mathbb{E}[\|\mathbf{z} - \mathbf{z}^{(k)}\|^2] + \frac{4(B^2 + M^2)}{\mu}$$

where (a) follows by recalling $k_0 = 2$. Summing over $k \in \{1, 2, \ldots, K\}$ and dividing both sides by $\sum_{k=1}^{K}(k + k_0 - 1)$ yields

$$\mathbb{E}\left[\frac{1}{\sum_{k=1}^{K}(k + k_0 - 1)} \sum_{k=1}^{K}(k + k_0 - 1)\phi(\mathbf{z}^{(k)})\right]$$
$$\leq \phi(\mathbf{z}) + \frac{\mu(k_0^2 - k_0)}{2K(K + 2k_0 - 1)}\mathbb{E}[\|\mathbf{z} - \mathbf{z}^{(0)}\|^2] - \frac{\mu(k_0^2 - k_0)}{2K(K + 2k_0 - 1)}\mathbb{E}[\|\mathbf{z} - \mathbf{z}^{(K)}\|^2] - \frac{\mu}{2}\mathbb{E}[\|\mathbf{z} - \mathbf{z}^{(K)}\|^2]$$
$$+ \frac{8(B^2 + M^2)}{(K + 2k_0 - 1)\mu}$$

Define $\widehat{\mathbf{z}} \overset{\Delta}{=} \frac{1}{\sum_{k=1}^{K}(k + k_0 - 1)}(k + k_0 - 1)\mathbf{z}^{(k)}$. By Jensen's inequality and recalling $k_0 = 2$, we have

$$\mathbb{E}[\phi(\widehat{\mathbf{z}})]$$
$$\leq \phi(\mathbf{z}) + \frac{\mu}{K(K + 3)}\mathbb{E}[\|\mathbf{z} - \mathbf{z}^{(0)}\|^2] - \frac{\mu}{K(K + 3)}\mathbb{E}[\|\mathbf{z} - \mathbf{z}^{(K)}\|^2] - \frac{\mu}{2}\mathbb{E}[\|\mathbf{z} - \mathbf{z}^{(K)}\|^2] + \frac{8(B^2 + M^2)}{(K + 3)\mu}$$

$\square$

**Lemma 15.** *Consider convex program* (1) *with $\mu$-convex stochastic objective functions under Assumption 4. Let $\mathbf{x}^*$ be any optimal solution. If $\nu^{(t)} > 0$ and $\rho^{(t)} > 0$ in Algorithm 1 are chosen to satisfy*

$$\nu^{(t)} \geq \rho^{(t)} \|\mathbf{A}\|^2, \forall t,$$

*and the sub-procedure STO-LOCAL (Algorithm 2) uses $k_0 = 2$ and $\widehat{\mathbf{z}}$ defined in Lemma 14 as the output, then, for all $T \geq 1$, Algorithm 1 ensures*

$$\sum_{t=1}^{T} \mathbb{E}[\rho^{(t)} f(\mathbf{x}^{(t)})] \leq \sum_{t=1}^{T} \mathbb{E}[\rho^{(t)} f(\mathbf{x}^*)] + \frac{1}{2} \sum_{t=1}^{T} \mathbb{E}[\rho^{(t)} \Phi^{(t)}] - \frac{1}{2} \mathbb{E}[\|\boldsymbol{\lambda}^{(T)}\|^2] + \sum_{t=1}^{T} \frac{8\rho^{(t)}(B^2 + M^2)}{(\nu^{(t)} + \mu)(K^{(t)} + 3)}$$

*where*

$$\Phi^{(t)} \triangleq \left( \nu^{(t)} + \frac{2(\nu^{(t)} + \mu)}{K^t(K^t + 3)} \right) \|\mathbf{x}^* - \mathbf{y}^{(t-1)}\|^2 - \left( \nu^{(t)} + \mu + \frac{2(\nu^{(t)} + \mu)}{K^t(K^t + 3)} \right) \|\mathbf{x}^* - \mathbf{y}^{(t)}\|^2 \tag{91}$$

*and $B$ and $M$ are constants defined in Assumption 4.*

*Proof.* Fix $t \in \{1, 2, \ldots, T\}$. Define $\phi^{(t)}(\mathbf{x}) = \sum_{i=1}^{N} \phi_i^{(t)}(\mathbf{x}_i)$. Note that if we define $\dot{\phi}^{(t)}(\mathbf{x}) = f(\mathbf{x})$ and $\ddot{\phi}^t(\mathbf{x}) = \rho^{(t)} \langle \mathbf{r}^{t-1} + \frac{1}{\rho^{(t)}} \boldsymbol{\lambda}^{(t-1)}, \mathbf{A}\mathbf{x} - \mathbf{b} \rangle + \frac{\nu^{(t)}}{2} \|\mathbf{x} - \mathbf{y}^{(t-1)}\|^2$, then $\phi^{(t)}(\mathbf{x}) = \dot{\phi}^{(t)}(\mathbf{x}) + \ddot{\phi}^t(\mathbf{x})$ where $\dot{\phi}^{(t)}(\mathbf{x})$ is $\mu$-convex and satisfies (79) by Assumption 4 and $\ddot{\phi}^t(\mathbf{x})$ is $\nu^{(t)}$-convex and $\nu^{(t)}$-smooth. As in the observation in the proof of Lemma 11 or Lemma 13, each iteration of Algorithm 1 is to jointly update $\mathbf{x}$ and $\mathbf{y}$ via the sub-procedure $(\mathbf{x}^{(t)}, \mathbf{y}^{(t)}) = \text{STO-LOCAL}(\phi^{(t)}(\cdot), \mathcal{X}, \mathbf{y}^{(t-1)}, K^{(t)})$.

Thus, by Lemma 14, we have

$$\mathbb{E}[\phi^{(t)}(\mathbf{x}^{(t)})] \leq \phi^{(t)}(\mathbf{x}^*) + \frac{(\nu^{(t)} + \mu)}{K(K+3)} \mathbb{E}[\|\mathbf{z} - \mathbf{z}^{(0)}\|^2] - \frac{(\nu^{(t)} + \mu)}{K(K+3)} \mathbb{E}[\|\mathbf{z} - \mathbf{z}^{(K)}\|^2]$$
$$- \frac{(\nu^{(t)} + \mu)}{2} \mathbb{E}[\|\mathbf{z} - \mathbf{z}^{(K)}\|^2] + \frac{8(B^2 + M^2)}{(K+3)(\nu^{(t)} + \mu)}$$

This is almost identical to (62) (with $k_0 = 2$) in Lemma 13 except the constant $\sigma^2$ is replaced by $2(B^2 + M^2)$.

Following the same lines (after (62)) in the proof of Lemma 13, we can show

$$\sum_{t=1}^{T} \mathbb{E}[\rho^{(t)} f(\mathbf{x}^{(t)})] \leq \sum_{t=1}^{T} \mathbb{E}[\rho^{(t)} f(\mathbf{x}^*)] + \frac{1}{2} \sum_{t=1}^{T} \mathbb{E}[\rho^{(t)} \Phi^{(t)}] - \frac{1}{2} \mathbb{E}[\|\boldsymbol{\lambda}^{(T)}\|^2] + \sum_{t=1}^{T} \frac{8\rho^{(t)}(B^2 + M^2)}{(\nu^{(t)} + \mu)(K^{(t)} + 3)}$$

where $\Phi^{(t)} \triangleq \left( \nu^{(t)} + \frac{2(\nu^{(t)} + \mu)}{K^t(K^t + 3)} \right) \|\mathbf{x}^* - \mathbf{y}^{(t-1)}\|^2 - \left( \nu^{(t)} + \mu + \frac{2(\nu^{(t)} + \mu)}{K^t(K^t + 3)} \right) \|\mathbf{x}^* - \mathbf{y}^{(t)}\|^2$.

Since the conclusion from Lemma 15 is quite similar to that from Lemma 13 (with $k_0 = 2$) with the minor distinction that constant $\sigma^2$ is replaced by $2(B^2 + M^2)$, the main proof of Theorem 4 is similar to the proof of $\mu > 0$ case of Theorem 2.

**Main Proof of Theorem 4**: Note that our selection of $\rho^{(t)} = t\rho$, $\nu^{(t)} = t\rho\|\mathbf{A}\|^2$ and $k_0 = 2$ satisfies the conditions in Lemma 15. By Lemma 15, we have

$$\sum_{t=1}^{T} \mathbb{E}[\rho^{(t)} f(\mathbf{x}^{(t)})] \leq \sum_{t=1}^{T} \mathbb{E}[\rho^{(t)} f(\mathbf{x}^*)] + \frac{1}{2} \sum_{t=1}^{T} \mathbb{E}[\rho^{(t)} \Phi^{(t)}] - \frac{1}{2} \mathbb{E}[\|\boldsymbol{\lambda}^{(T)}\|^2] + \sum_{t=1}^{T} \frac{8\rho^{(t)}(B^2 + M^2)}{(\nu^{(t)} + \mu)(K^{(t)} + 3)} \tag{92}$$

Recalling the definition of $\Phi^t$ in (91) and $K^{(t)} = 3t$, we have

$$\sum_{t=1}^{T} \rho^{(t)} \Phi^{(t)}$$

$$= \rho \left( \rho \|\mathbf{A}\|^2 + \frac{(\rho\|\mathbf{A}\|^2 + \mu)}{18} \right) \|\mathbf{x}^* - \mathbf{y}^{(0)}\|^2$$

$$- \sum_{t=1}^{T-1} \left( \rho \left( \rho t^2 \|\mathbf{A}\|^2 + t\mu - \rho(t+1)^2 \|\mathbf{A}\|^2 \right) + \rho \left( \frac{\rho t \|\mathbf{A}\|^2 + \mu}{t+1} - \frac{\rho(t+1)\|\mathbf{A}\|^2 + \mu}{t+2} \right) \frac{2}{9} \right) \|\mathbf{x}^* - \mathbf{y}^{(t)}\|^2$$

$$- T\rho \left( T\rho\|\mathbf{A}\|^2 + \mu + \left( \frac{\rho T\|\mathbf{A}\|^2 + \mu}{T(T+1)} \right) \frac{2}{9} \right) \|\mathbf{x}^* - \mathbf{y}^T\|^2$$

$$\overset{(a)}{\leq} \rho \left( \rho\|\mathbf{A}\|^2 + \frac{2(\rho\|\mathbf{A}\|^2 + \mu)}{18} \right) \|\mathbf{x}^* - \mathbf{y}^{(0)}\|^2 \tag{93}$$

where (a) follows by ignoring the last negative term and noting that $\rho t^2 \|\mathbf{A}\|^2 + t\mu - \rho(t+1)^2\|\mathbf{A}\|^2 = t\mu - \rho(2t+1)\|\mathbf{A}\|^2 \geq \mu(t - \frac{2t+1}{3}) \geq 0$ for all $t \geq 1$, where the first inequality uses $\rho \leq \frac{\mu}{3\|\mathbf{A}\|^2}$; and $\frac{\rho t\|\mathbf{A}\|^2 + \mu}{t+1} - \frac{\rho(t+1)\|\mathbf{A}\|^2 + \mu}{t+2} = \frac{\mu - \rho\|\mathbf{A}\|^2}{(t+1)(t+2)} \geq 0$, where the inequality also uses $\rho \leq \frac{\mu}{3\|\mathbf{A}\|^2}$.

We also note that

$$\sum_{t=1}^{T} \frac{8\rho^{(t)}(B^2 + M^2)}{(\nu^{(t)} + \mu)(K^{(t)} + 3)} = \frac{8\rho(B^2 + M^2)}{3} \sum_{t=1}^{T} \frac{t}{(\rho t \|\mathbf{A}\|^2 + \mu)(t+1)}$$

$$\overset{(a)}{\leq} \frac{8\rho(B^2 + M^2)}{3} \sum_{t=1}^{T} \frac{t}{(t+1)(t+3)\|\mathbf{A}\|^2}$$

$$\leq \frac{8\rho(B^2 + M^2)}{3\|\mathbf{A}\|^2} \sum_{t=1}^{T} \frac{1}{t+1}$$

$$\leq \frac{8\rho(B^2 + M^2)}{3\|\mathbf{A}\|^2} \log(T+1) \tag{94}$$

where (a) follows because $\mu \geq 3\rho\|\mathbf{A}\|^2$ by $\rho \leq \frac{\mu}{3\|\mathbf{A}\|^2}$.

Substituting (93) and (94) into (92) yields

$$\sum_{t=1}^{T} \mathbb{E}[\rho^{(t)} f(\mathbf{x}^{(t)})]$$

$$\leq \sum_{t=1}^{T} \mathbb{E}[\rho^{(t)} f(\mathbf{x}^*)] + \frac{1}{2}\rho \left( \rho\|\mathbf{A}\|^2 + \frac{2(\rho\|\mathbf{A}\|^2 + \mu)}{18} \right) \|\mathbf{x}^* - \mathbf{y}^{(0)}\|^2 - \frac{1}{2}\mathbb{E}[\|\boldsymbol{\lambda}^{(T)}\|^2]$$

$$+ \frac{8\rho(B^2 + M^2)}{\|\mathbf{A}\|^2} \log(T+1)$$

$$\overset{(a)}{\leq} \sum_{t=1}^{T} \mathbb{E}[\rho^{(t)} f(\mathbf{x}^*)] + \frac{1}{2}\rho c_1 \|\mathbf{x}^* - \mathbf{y}^{(0)}\|^2 - \frac{1}{2}\mathbb{E}[\|\boldsymbol{\lambda}^{(T)}\|^2] + \frac{1}{2} c_2 \log(T+1) \tag{95}$$

where (a) follows because $c_1 = \rho\|\mathbf{A}\|^2 + \frac{2(\rho\|\mathbf{A}\|^2 + \mu)}{18}$ and $c_2 = \frac{16\rho(B^2 + M^2)}{3\|\mathbf{A}\|^2}$.

Ignoring the negative term $-\frac{1}{2}\mathbb{E}[\|\boldsymbol{\lambda}^{(T)}\|^2]$, dividing both sides by $\sum_{t=1}^{T} \rho^{(t)}$ and applying Jensen's inequality yields

$$\mathbb{E}[f(\overline{x}^T)] \leq f(\mathbf{x}^*) + \frac{1}{2\sum_{t=1}^{T} \rho^{(t)}} \left( \rho c_1 \|\mathbf{x}^* - \mathbf{y}^{(0)}\|^2 + c_2 \log(T+1) \right)$$

$$\overset{(a)}{=} f(\mathbf{x}^*) + \frac{1}{T(T+1)} \left( c_1 \|\mathbf{x}^* - \mathbf{y}^{(0)}\|^2 + \frac{c_2}{\rho} \log(T+1) \right)$$

where (a) follows because $\sum_{t=1}^{T} \rho^{(t)} = \rho \sum_{t=1}^{T} t = \frac{\rho T(T+1)}{2}$. This is (77) of our theorem.

By Lemma 4 (after taking expectations on both sides), we have

$$\mathbb{E}[\sum_{t=1}^{T} \rho^{(t)} f(\mathbf{x}^{(t)})] \geq \sum_{t=1}^{T} \mathbb{E}[\rho^{(t)} f(\mathbf{x}^*)] - \|\boldsymbol{\lambda}^*\|\mathbb{E}[\|\boldsymbol{\lambda}^{(T)}\|]$$

Combining this inequality with (75) and cancelling the common terms yields

$$\mathbb{E}[\|\boldsymbol{\lambda}^{(T)}\|^2] \leq 2\|\boldsymbol{\lambda}^*\|\mathbb{E}[\|\boldsymbol{\lambda}^{(T)}\|] + \rho c_1 \|\mathbf{x}^* - \mathbf{y}^{(0)}\|^2 + c_2 \log(T+1)$$

Since $\left(\mathbb{E}[\|\boldsymbol{\lambda}^{(T)}\|]\right)^2 \leq \mathbb{E}[\|\boldsymbol{\lambda}^{(T)}\|^2]$, we further have

$$\left(\mathbb{E}[\|\boldsymbol{\lambda}^{(T)}\|]\right)^2 \leq 2\|\boldsymbol{\lambda}^*\|\mathbb{E}[\|\boldsymbol{\lambda}^{(T)}\|] + \rho c_1 \|\mathbf{x}^* - \mathbf{y}^{(0)}\|^2 + c_2 \log(T+1)$$

This quadratic inequality can be rewritten as

$$\left(\mathbb{E}[\|\boldsymbol{\lambda}^{(T)}\|] - \|\boldsymbol{\lambda}^*\|\right)^2 \leq \|\boldsymbol{\lambda}^*\|^2 + \rho c_1 \|\mathbf{x}^* - \mathbf{y}^{(0)}\|^2 + c_2 \log(T+1)$$

Thus, we have

$$\mathbb{E}[\|\boldsymbol{\lambda}^{(T)}\|] \leq \|\boldsymbol{\lambda}^*\| + \sqrt{\|\boldsymbol{\lambda}^*\|^2 + \rho c_1 \|\mathbf{x}^* - \mathbf{y}^{(0)}\|^2 + c_2 \log(T+1)}$$

$$\leq 2\|\boldsymbol{\lambda}^*\| + \sqrt{\rho c_1}\|\mathbf{x}^* - \mathbf{y}^{(0)}\| + \sqrt{c_2 \log(T+1)} \tag{96}$$

By part (1) of Lemma 3 (with $\rho^{(t)} = \rho$), we have

$$\sum_{t=1}^{T} \rho \left(\mathbf{A}\mathbf{x}^{(t)} - \mathbf{b}\right) = \boldsymbol{\lambda}^{(T)}$$

Dividing both sides by $\sum_{t=1}^{T} \rho^{(t)} = \rho \frac{T(T+1)}{2}$, taking the vector $l_2$ norm and then taking expectations on both sides yields

$$\mathbb{E}[\|\mathbf{A}\overline{\mathbf{x}}^{(T)} - \mathbf{b}\|] = \frac{2}{T(T+1)}\mathbb{E}[\|\boldsymbol{\lambda}^{(T)}\|]$$

$$\overset{(a)}{\leq} \frac{2}{T(T+1)} \left(\frac{4\|\boldsymbol{\lambda}^*\|}{\rho} + \frac{\sqrt{c_1}}{\sqrt{\rho}}\|\mathbf{x}^* - \mathbf{y}^{(0)}\| + \frac{\sqrt{c_2 \log(T+1)}}{\rho}\right)$$

where (a) follows from (76). This is (78) of our theorem.

$\square$