[Reviews · NeurIPS 2019]

Reviewer 1



Overall the paper is well written and the proposed algorithm is well explained. Since it is variant of the existing ADMM, the authors are encouraged to spend more texts to explain the key difference, for example, what kind of advantages we can get by using this new ADMM variant. On the other hand, I find the experimental part is pretty weak. Using real and large scale data may further improve the paper. Line 36, it seems the existing ADMM can be extended to the case N \ge 3, based on certain strongly convexity condition. Is the reduction of the communication round due to the two layer-structure in Algorithm 1? Besides this part, what could be the essential advantage of this proposed ADMM method? On the other hand, the authors may want to show how much computation time or resources we can get by this communication round reduction. This paper doesn’t talk much about the fault tolerance. Does the model convergence rely on the success of the communication? If one communication fails, what will the results look like? This proposed method is still limited to the decomposable property in the variable. If it is not the case, for example, apply a L_{21} regularization term in the objective, will the Algorithm still work? In Section 4, these experimental results are all based on synthetic and small data sets. The effectiveness will be more convincing if real and large data sets are used for evaluation. Some minor comments: (1) On Line 28-29, the presentation can be be simplified as “Ax = b, x_i == x_j \forall i, j”. Apparently x_i == x_j if i == j.

Reviewer 2



This paper proposes a new communication efficient multi-block ADMM for linearly constrained stochastic optimization. The proposed method is as fast as (or faster than) existing stochastic ADMMs but the associated communication overhead is only the square root of that required by existing ADMMs. Although the paper is theoretically sound, there are still some questions need to be discussed in this paper: 1. This paper assumes that the strong duality hold in Assumption 1, but in many applications this assumption does not hold. Therefore, does the algorithm in this paper have some limitations? 2. What’s the stochastic objective function? In addition, there are many related algorithms, for example, E. Wei and A. Ozdaglar. Distributed Alternating Direction Method of Multipliers. CDC, 2012. F. Huang, S. Chen and H. Huang. Faster Stochastic Alternating Direction Method of Multipliers for Nonconvex Optimization. ICML, 2019. T. Chang, W. Liao, M. Hong and X. Wang. Asynchronous Distributed ADMM for Large-Scale Optimization-Part II: Linear Convergence Analysis and Numerical Performance. IEEE Transactions on Signal Processing, 2016. 3. The experiments in Section 4.2 of this paper lack the detailed information about the experimental environment, and the tools and hardware facilities used in the experiment should be appropriately introduced. 4. More compared algorithms (including distributed deterministic ADMM and stochastic ADMM algorithms) should be added to reflect the superiority of the proposed algorithm, and the performance of the proposed algorithm in terms of running time should be also reported.

Reviewer 3



This paper considers a communication efficient distributed optimization algorithm based on ADMM for stochastic optimization. The main idea is to perform multiple steps (can be timevarying) of stochastic gradient updates before the agents communicate, and therefore improving the communication efficiency. The proposed algorithm is shown to converge (in objective value & constraint violation) under a general non-smooth, non-strongly convex settings with O(1/eps) communication rounds and O(1/eps^2) unbiased gradient oracle calls. Other setting such as smooth+strongly convex and non-smooth+strongly convex are also analyzed and presented. Using multiple steps is however not novel. In addition, a significant drawback for the proposed algorithm is that it is not supported for time varying communication -- which is a major set back with ADMM type distributed optimization. While in overall the paper is well written with several interesting results (e.g., convergence with non-smooth+non-strongly convex problems), there are some outstanding concerns from the reviewer: -- Communication Efficiency and Comparison to Prior Work This paper counts the number of "outer-loop" as a measure of communication efficiency, especially in terms of its scaling with the desired accuracy epsilon. However, such notion should also be compared in terms of the network topology which is missing in the current Theorems (and is perhaps hidden in the constants that depends on ||A||^2). These notions are emphasized and compared in a few recent work: Nedic et al., "Network topology and communication-computation tradeoffs in decentralized optimization", Proceedings of IEEE, 2018. Scaman et al., "Optimal Algorithms for Non-Smooth Distributed Optimization in Networks", in NeurIPS 2018. Uribe et al., "Optimal Algorithms for Distributed Optimization", arXiv:1712.00232 It is also important to point out that Scaman et al. have considered using multiple step in primal update to achieve a similar performance as the current paper, i.e., requiring O(1/eps) outer loops to achieve an eps-accurate solution. In addition to the above work that tackles deterministic optimization, the following work has also considered stochastic optimization problem like the current paper: S. Pu, A. Nedic, "Distributed Stochastic Gradient Tracking Methods", arXiv:1805.11454 Without using a multiple steps update, they also achieve a similar O(1/eps) rate for strongly convex objective functions. Comparing these methods (at least) numerically is important for improving the current paper. Overall, the literature review done in this paper seems to be limited to ADMM type algorithms, while missing a lot of advances made in the gradient tracking type algorithms. Note that the latter is also actively researched. -- Minor Comments - While it is understood that the constraints "\sum_{i=1}^N A_i x_i = b" can be re-expressed to reflect on the consensus constraints, the application of the proposed algorithms to decentralized setting may still be unclear for non-expert readers. Such formulation should be described clearly in the paper as well. (e.g., in terms of the "A_i" and "b" involved in these settings). - the authors mentioned that the negative term is (7) is useful for analysis - please expatiate in terms of its intuition as it is not clear in the main paper. ---- The reviewer has read the response and I agree that the network dependence is more minor than my suggested references, which should be emphasized better in the revision. However, my comment with time varying / unreliable communication is not addressed by the authors. This concern is relevant to the authors response R2Q1, where (4),(5) are claimed to be ignorable if communication fails - there the authors seem to assume that communication failure is an "all or nothing" situation, while in reality communication failure happens only partially in the network. Second, the theory requires setting $K=T$ where $T=O(1/epsilon)$. This can be a huge number and requires knowing $\epsilon$ in advance.

[Author Response · NeurIPS 2019]

**Simulation Update:**

- **Environment:** Our experiments are performed over a machine with 27 cores (Intel Xeon Processor E5-2682 2.5GHz) in Python 3.7. Each parallel node is an independent process/core and inter-process communication uses MPI4PY.
- **per round communication time vs per round computation time**: The exact time depends on the number of nodes/processes and variable dimensions. In the experiment of Sec 4.1 , each computation round takes $0.3ms$ and each communication round takes $43.7ms$. (Communication is 110 times more expensive in this case.)
- **Large scale real data set and more baselines:** We further perform the multi-class (10 classes) classification task over MNIST data set, which contains 60000 training images and each image can be considered as a $784 + 1$ dimensional feature vector. Since the number of classes is 10, the classification is a convex optimization with a 7850 dimensional variable. Besides our method, RPDBUS ADMM, and DCS, we further test the deterministic ADMM and the stochastic ADMM in Pu&Nedic 18 (suggested by Rev5). We partition the training set into 4 disjoint subsets and solve the multi-class classification problem with 4 parallel processes. The wall-clock time (including both computation and communication) to converge to the optimal with $\|\mathbf{x}_i - \mathbf{x}_j\|_\infty \leq 10^{-4}, \forall i, j$ for each method is: our method (28.49sec), PRDBUS (1837sec), DCS (684sec), deterministic ADMM (12hour+), Pu&Nedic (3591sec). Note that our method is significantly faster than others when measured by wall-clock time.

**R2Q1:** Elaborate more and discuss tolerance on failure of communication

**A:** Our method is robust to failure of communication. If communication fails, we can skip (4-5) and let each local node continue to run its sub-procedure STO-LOCAL for one more time. Mathematically, this is equivalent to a normal Algorithm 1 implementation where one particular STO-LOCAL step runs more iterations. Our convergence analysis only requires a minimum number of iterations is executed in each STO-LOCAL sub-procedure. So the convergence is guaranteed by our theory. Both theoretical elaboration and extra experiment results will be reported in the final version.

**R2Q2:** Decomposable property and $L_{21}$ regularization.

**A:** This paper assumes the original problem has been **reformulated** into (1), which has a decomposable structure. For problems with $L_{21}$ regularization, the applicability of our method depends on whether they can be reformulated into (1). For example, consider a robust $L_{21}$ feature selection given by $\min_{\mathbf{W}} \|\mathbf{W}^T\mathbf{X} - \mathbf{Y}\|_{2,1} + \gamma\|\mathbf{W}\|_{2,1}$. It can be reformulated as $\min_{\mathbf{W},\mathbf{V}} \|\mathbf{V}\|_{2,1} + \gamma\|\mathbf{W}\|_{2,1} \ \ s.t. \ \ \mathbf{W}^T\mathbf{X} - \mathbf{Y} - \mathbf{V} = \mathbf{0}$. Since $L_{21}$ norm is separable w.r.t. each row and linear constraints are separable w.r.t. each entry, it is decomposable w.r.t. each row of $\mathbf{W}$ and $\mathbf{V}$ and can be solved in a distributed way with our method.

**R3Q1:** Strong duality in Assumption 1

**A:** Assumption 1 is mild for convex programs with linear constraints. For problems with linear constraints, Proposition 6.4.2 in "D. P. Bertsekas, A. Nedic, and A. E. Ozdaglar, Convex Analysis and Optimization." ensures Assumption 1 as long as the feasible set is non-empty and the domain of the objective function satisfies any of the following 3 conditions: (1) contains the feasible set (2) open or (3) can be convexly extended to open sets. In particular, all linear programs with non-empty feasible sets satisfy Assumption 1.

**R3Q2:** stochastic objective function and related papers

**A:** The stochastic objective fun in Sec 4.1 is a pure stochastic function where the randomness is $\mathbf{c}_i$. The stochastic fun in Sec 4.2 is a finite sum that is expectation involving uniform distribution of the samples. Stochastic opt methods for Sec 4.2 allow us to evaluate a single sample rather than all samples for each iteration and yield low complexity. All your suggested papers on ADMM are discussed and cited in the revision.

**R5:** dependence on network topology and references on "local averaging" methods.

**A:** Yes, the dependence on network topology is hidden in $\|\mathbf{A}\|$. By our Remark 3, if we choose $\rho$ to balance the dependence, both objective and constraint violations linearly depends on $\|\mathbf{A}\|$.

Compared with Nedic et al. 2018, Scaman et al. 17, Uribe et al. 17, and Pu&Nedic 18, all of which use a doubly stochastic or symmetric PSD matrix for local averaging, our ADMM method has the following advantages:
- Our inter-node communication pattern is more flexible and is not restricted to a (symmetric) pattern such as the (doubly) stochastic or symmetric PSD matrix. Of course, we can choose $\mathbf{A} = \mathbf{I} - \mathbf{W}$ where $\mathbf{W}$ is a stochastic matrix used in your suggested works since it ensures the consensus of local solutions. However, in general, we can use any $\mathbf{A}$ to ensure consistence as long as Null$\{\mathbf{A}\}$ =Span$\{\mathbf{1}\}$.
- While the dynamics of ADMM is different from mixing (local averaging) based method, our Theorem 1 and Remark 3 suggest our method can have better dependence on network topology. Our convergence only depends on $\|\mathbf{A}\|$. By choosing $\mathbf{A} = \mathbf{I} - \mathbf{W}$, we know $\|\mathbf{A}\| \leq 2$. The convergence in suggested works (using a doubly stochastic or a symmetric PSD $\mathbf{W}$ for mixing) further depends on $1/(1 - \{|\lambda_2(\mathbf{W})|, |\lambda_N(\mathbf{W})|\})$ or eigengap $\lambda_1(\mathbf{W})/\lambda_{N-1}(\mathbf{W})$, which can be much larger than constant 2 if some eigenvalues are extreme.

Nevertheless, the above suggested papers are related and complement ADMM methods. They are discussed and cited in the revision.

[Meta-Review · NeurIPS 2019]

Originally, the paper received one positive score and two negative scores 7/5/5 with acceptable confidence scores 3/4/3. The major issues include missing fault tolerance, weak experiments, insufficient literature review, etc. During discussion, Reviewer #5 was somwhat convinced by the authors' rebuttal and intended to raised his/her score. However, he/she was still concerned about the assumption that K=T=O(1/\eppsilon). The AC deemed that this is not a big issue. So the AC recommended acceptance. Nonetheless, the fault tolerance issue raised by Reviewers #2 and #5 was unaddressed. So the AC did not strongly support the acceptance.